# Quantifying the global atmospheric power budget

Anastassia M. Makarieva<sup>1,2</sup>, Victor G. Gorshkov<sup>1,2</sup>, Andrei V. Nefiodov<sup>1</sup>, Douglas Sheil<sup>3</sup>, Antonio Donato Nobre<sup>4</sup>, and Bai-Lian Li<sup>2</sup>

<sup>1</sup>Theoretical Physics Division, Petersburg Nuclear Physics Institute, 188300 Gatchina, St. Petersburg, Russia <sup>2</sup>USDA-China MOST Joint Research Center for AgroEcology and Sustainability, University of California, Riverside 92521-0124, US

<sup>3</sup>Norwegian University of Life Sciences, Ås, Norway

<sup>4</sup>Centro de Ciência do Sistema Terrestre INPE, São José dos Campos SP 12227-010, Brazil

*Correspondence to:* Anastassia Makarieva (ammakarieva@gmail.com), Victor Gorshkov (vigorshk@thd.pnpi.spb.ru), Andrei Nefiodov (anef@thd.pnpi.spb.ru), Douglas Sheil (douglas.sheil@nmbu.no), Antonio Donato Nobre (anobre27@gmail.com), Bai-Lian Li (bai-lian.li@ucr.edu)

**Abstract.** Starting from the definition of mechanical work for an ideal gas, we present a novel derivation linking global wind power to measurable atmospheric parameters. The resulting expression distinguishes three components: the kinetic power associated with horizontal motion, the kinetic power associated with vertical motion and the gravitational power of precipitation. We discuss the caveats associated with integration of material derivatives in the presence of phase transitions and how these affect published analyses of global atmospheric power. Using the MERRA database for the years 2009-2015 (three hourly data on the  $1.25^{\circ} \times 1.25^{\circ}$ grid at 42 pressure levels) we estimate total atmospheric power at  $3.1 \text{ W m}^{-2}$  and kinetic power at  $2.6 \text{ W m}^{-2}$ . The difference between the two  $(0.5 \text{ W m}^{-2})$  is about half the independently estimated gravitational power of precipitation of the database. Our analysis suggests that the total atmospheric power calculated with a spatial resolution of the order of one kilometer (thus capturing the small moist convective eddies) should be around  $5 \text{ W m}^{-2}$ . We discuss the physical constraints on global atmospheric power and how considering the dynamic effects of water vapor condensation offers new opportunities.

#### 1 Introduction

How much power does our atmosphere generate and why? These questions have long challenged theorists (Lorenz, 1967) and have gained renewed significance given how our climate is affected by changes in atmospheric circulation (e.g., Bates, 2012; Shepherd, 2014). A particular problem is the mismatch between model predictions and observed trends (e.g., Kociuba and Power, 2015). While models suggest general circulation should slow as global temperatures rise, independent observations indicate circulation is getting faster (e.g., de Boisséson et al., 2014). Global wind power appears to be rising as well (Huang and McElroy, 2015). On the other hand, global circulation models tend to overestimate wind power (Boer and Lambert, 2008). For example, using the CAM3.5 model Marvel et al. (2013) estimated the global kinetic power of the atmosphere<sup>1</sup> at 3.4 W m<sup>-2</sup>,

<sup>&</sup>lt;sup>1</sup>Note however that Boville and Bretherton (2003) cited 2 W m<sup>-2</sup> as the kinetic power for the same model. To our knowledge, the inconsistency between their estimate and that of Marvel et al. (2013) has not been addressed.

while published observational estimates range from 2 to  $2.5 \text{ W m}^{-2}$  (Kim and Kim, 2013; Schubert and Mitchell, 2013; Huang and McElroy, 2015).

Our motivation here is to clarify what is meant by *total atmospheric power* and how we can assess it. Suppose that we know at a given moment in time the air velocity and pressure gradient for each point in the atmosphere with good resolution. Would this information be sufficient to estimate the instantaneous global atmospheric power? We found that for us at least the answer to this question was not self-evident. In Section 2 we explore how the derivation of an expression for the global atmospheric power is affected by the presence of phase transitions.

In Section 3 we show how the derived relationships require a revision of the recent estimates of atmospheric wind power proposed by Laliberté et al. (2015). In Section 4 we discuss how the global wind power in a hydrostatic atmosphere can be represented as a sum of three distinct physical components. Two components dominate in a hydrostatic atmosphere: the kinetic power of the wind generated by horizontal pressure gradients and the gravitational power of precipitation generated by the ascending air. In Section 5 we illustrate our formulations by analysing the atmospheric power budget using the MERRA database (Rienecker et al., 2011). We discuss how estimates of atmospheric power are affected by resolution of the data.

In the concluding section we discuss theoretical constraints on atmospheric power. There are two problems: to explain why wind power on Earth is significantly different from zero (minimal threshold) and to find a constraint from above on this power (maximum threshold). We discuss the opportunities provided by consideration of the dynamic effects of condensation in combination with conventional thermodynamic approach.

#### 2 Atmospheric power in the presence of phase transitions

The atmosphere comprises compressible gases and small amounts of incompressible solid and liquid water. We begin with calculating work performed by the atmospheric gases. We treat them as ideal gases with the equation of state

$$p = NRT.$$
 (1)

Here T is temperature, N is air molar density (mol m<sup>-3</sup>),  $V \equiv N^{-1}$  is the atmospheric volume occupied by one mole of air, p is air pressure and R = 8.3 J mol<sup>-1</sup> K<sup>-1</sup> is the universal gas constant.

Work per unit time (power) of an air parcel containing  $\tilde{N}$  moles and occupying volume  $\tilde{V} \equiv \tilde{N}V$  (m<sup>3</sup>) is

$$p\frac{d\tilde{V}}{dt} = p\left(\tilde{N}\frac{dV}{dt} + V\frac{d\tilde{N}}{dt}\right) = \tilde{V}\left(-\frac{dp}{dt} + RN\frac{dT}{dt}\right) + RT\frac{d\tilde{N}}{dt}.$$
(2)

We now consider an atmosphere composed of n air parcels with total volume  $\mathcal{V} \equiv \sum_{i=1}^{n} \tilde{V}_i = \int_{\mathcal{V}} d\mathcal{V}$  (subscript *i* refers to the *i*-th air parcel). We define  $\mathcal{V}$  as the volume bounded by the Earth's surface and the surface corresponding to some fixed pressure level  $p_T$  at the top of the atmosphere, e.g. to  $p_T = 0.1$  hPa. This is the uppermost level in many atmospheric datasets including those in the MERRA project.

The number of molecules (moles)  $\tilde{N}$  in each air parcel can only change via an inflow (outflow) of molecules through the parcel's boundary. This change results from either diffusion of molecules between the adjacent parcels or from phase

transitions or from both. Since in the case of diffusion any molecule leaving one parcel,  $d\tilde{N}_1/dt < 0$ , arrives to some other parcel,  $d\tilde{N}_2/dt = -d\tilde{N}_1/dt > 0$ , all the diffusion terms cancel in the global sum of the last term in Eq. (2) over all parcels. What remains corresponds to phase transitions:

$$\sum_{i=1}^{n} \frac{d\tilde{N}_{i}}{dt} = \int_{\mathcal{V}} \frac{1}{\tilde{V}} \frac{d\tilde{N}}{dt} d\mathcal{V} = \int_{\mathcal{V}} \dot{N} d\mathcal{V},$$
(3)

where  $\dot{N}$  is the molar rate of phase transitions per unit volume (mol m<sup>-3</sup> s<sup>-1</sup>). Its integral over volume  $\mathcal{V}$  is equal to the total rate of phase transitions in all the *n* air parcels. By virtue of the conservation relationship (3)  $\dot{N}$  includes the inflow (outflow) into all the air parcels from all liquid or solid surfaces (droplet surface in the atmospheric interior or the Earth's surface).

Using Eqs. (2) and (3) we can write total power W of the n air parcels composing the atmosphere as

$$W \equiv \frac{1}{\mathcal{S}} \sum_{i=1}^{n} p_i \frac{d\tilde{V}_i}{dt} = \frac{1}{\mathcal{S}} \int_{\mathcal{V}} p \frac{1}{\tilde{V}} \frac{d\tilde{V}}{dt} d\mathcal{V} = \frac{1}{\mathcal{S}} \int_{\mathcal{V}} \left( -\frac{dp}{dt} + RN \frac{dT}{dt} + RT\dot{N} \right) d\mathcal{V}.$$
(4)

Here S is the Earth's surface area and W is in W m<sup>-2</sup>.

The exact differential dX/dt for  $X = \{p, T, V, ...\}$  that describes change within each air parcel in the atmospheric context corresponds to the material derivative:

$$\frac{dX}{dt} \equiv \frac{\partial X}{\partial t} + \mathbf{v} \cdot \nabla X,\tag{5}$$

where  $\mathbf{v}$  is the mean velocity of gas molecules within the parcel.

Using the continuity equation

$$\dot{N} = \frac{\partial N}{\partial t} + \nabla \cdot (N\mathbf{v}) = \frac{dN}{dt} + N(\nabla \cdot \mathbf{v})$$
(6)

and the ideal gas law (1) in the form dp/dt = RNdT/dt + RTdN/dt we obtain for W from Eq. (4):

$$W = \frac{1}{\mathcal{S}} \int_{\mathcal{V}} p \frac{1}{\tilde{V}} \frac{d\tilde{V}}{dt} d\mathcal{V} = \frac{1}{\mathcal{S}} \int_{\mathcal{V}} p(\nabla \cdot \mathbf{v}) d\mathcal{V}.$$
(7)

The magnitude of  $\nabla \cdot \mathbf{v}$  is the rate of relative volume change of a material element at a given point (e.g., Batchelor, 2000, p. 75). This change,  $(1/\tilde{V})d\tilde{V}/dt$  depends only on the difference in velocities of its bounding material surfaces which are not directly impacted by phase transitions. Therefore, W (7) does not explicitly depend on  $\dot{N}$ .

We can use the divergence theorem (Gauss-Ostrogradsky theorem) to re-write W (7) as

$$W = I_{S} - \frac{1}{S} \int_{\mathcal{V}} (\mathbf{v} \cdot \nabla p) d\mathcal{V} = I_{S} - \frac{1}{S} \int_{\mathcal{V}} (\mathbf{u} \cdot \nabla p + \mathbf{w} \cdot \nabla p) d\mathcal{V},$$

$$I_{S} \equiv \frac{1}{S} \int_{S} p(\mathbf{v} \cdot \mathbf{n}) d\mathcal{S} = \frac{p_{T}}{S} \int_{z=z(p_{T})} (\mathbf{v} \cdot \mathbf{n}) d\mathcal{S}.$$
(9)

Here **n** is the outward-pointing unit vector perpendicular to the surface area dS; the surface integral is taken over the Earth's surface and the upper boundary  $z = z(p_T)$ , where  $z(p_T)$  is the altitude of the pressure level  $p = p_T$ . Since the macroscopic

air parcels do not penetrate the Earth's surface<sup>2</sup>, for z = 0 we have  $\mathbf{v} \cdot \mathbf{n} = 0$ . At the upper boundary  $\mathbf{v}$  is zero in the steady state only, when the total volume  $\mathcal{V}$  of the air parcels does not change. Generally for  $z = z(p_T)$  we have  $\mathbf{v} \cdot \mathbf{n} \neq 0$  (a similar condition of non-zero velocity at the upper boundary (the oceanic surface) is commonly used in oceanic science, e.g. Tailleux (2015)).

# **3** Practical implications of the obtained relationships

## 3.1 The material derivative of enthalpy

In a recent effort to constrain the atmospheric power budget, Laliberté et al. (2015) used the thermodynamic identity

$$T\frac{ds}{dt} \equiv \frac{dh}{dt} - \frac{1}{\rho}\frac{dp}{dt} + \mu\frac{dq_T}{dt},\tag{10}$$

where s is entropy, h is enthalpy,  $\mu$  is chemical potential (all per unit mass of wet air),  $1/\rho$  is specific air volume and  $q_T$  is the mass fraction of total water<sup>3</sup>. Laliberté et al. (2015) neglected, as we do, the atmospheric liquid and solid water content<sup>4</sup> and approximated  $q_T = q_v$ , where  $q_v$  is the mass fraction of water vapor.

When integrating Eq. (10) over atmospheric mass, Laliberté et al. (2015) assumed that the enthalpy term vanishes,  $\int_{\mathcal{M}} (dh/dt) d\mathcal{M} = 0$ . This assumption was justified by noting that the atmosphere is approximately in a steady state. However, the correct condition for the steady atmospheric state is different. It corresponds to zero enthalpy change in all air parcels combined:

$$\frac{\partial h_{tot}}{\partial t} = \sum_{i=1}^{n} \frac{d\tilde{h}_i}{dt} = 0, \ h_{tot} \equiv \sum_{i=1}^{n} \tilde{h}_i, \ \tilde{h}_i \equiv m_i h_i.$$
(11)

where  $d\tilde{h}_i/dt$  is the change of enthalpy  $\tilde{h}_i$  of the *i*-th air parcel which has mass  $m_i$  and mass-specific enthalpy  $h_i$ .

Repeating the reasoning from the previous section we have

$$\frac{\partial h_{tot}}{\partial t} = \sum_{i=1}^{n} \frac{d\tilde{h}_i}{dt} = \sum_{i=1}^{n} \left( h_i \frac{dm_i}{dt} + m_i \frac{dh_i}{dt} \right) = \int_{\mathcal{V}} h\dot{\rho}d\mathcal{V} + \int_{\mathcal{M}} \frac{dh}{dt}d\mathcal{M},\tag{12}$$

where  $\dot{\rho}$  is the mass rate of phase transitions per unit volume (kg m<sup>-3</sup> s<sup>-1</sup>) obeying the steady-state continuity equation

$$\nabla \cdot (\rho \mathbf{v}) = \dot{\rho}. \tag{13}$$

<sup>&</sup>lt;sup>2</sup>The molecular process of evaporation from the Earth's surface can be viewed as an upward flux of gaseous molecules that are concentrated within one free path length l from the evaporating surface and have mean vertical velocity  $w_E$  of the order of mean molecular velocity. Molar density  $N_E$  of such molecules is obtained from evaporation rate E (kg m<sup>-2</sup> s<sup>-1</sup>) as  $N_E w_E = E/M_v$ , where  $M_v$  is molar mass of water. Due to the presence of these molecules, macroscopic air parcels adjacent to the surface could have a non-zero vertical velocity  $w_s$  defined as the mean vertical velocity of all molecules in the parcel. If  $l_p$  is the linear size of this parcel, we have  $w_s = w_E N_E l/(N_s l_p)$ , where  $N_s = p_s/(RT_s)$  is molar density of air at the surface,  $T_s$  and  $p_s$  are surface temperature and pressure. Then for the vertical term  $p(\mathbf{v} \cdot \mathbf{n})$  at the surface, Eq. (9), we have  $p_s w_s = (l/l_p) ERT_s/M_v$ . Since as we will see in Sections 5 and 6 global atmospheric power is of the order of  $PRT_s/M_v$ , where P = E is the global mean precipitation and evaporation, the surface term  $p_s w_s$  can be neglected in Eq. (9) if  $l/l_p \ll 1$ , i.e. on any macroscopic length scale.

<sup>&</sup>lt;sup>3</sup>The unconventional sign at the chemical potential term follows from  $\mu$  being defined in Eq. (10) relative to dry air: hence, when the relative dry air content diminishes this term is negative. For details see p. 8 in the Supplementary Materials of Laliberté et al. (2015).

<sup>&</sup>lt;sup>4</sup>This assumption corresponds to an instantaneous removal of the non-gaseous water from the atmosphere by precipitation.

As is clear from Eq. (11), when the atmosphere is in a steady state  $(\partial h_{tot}/\partial t = 0)$  the mass integral of the material derivative of enthalpy is not zero. It can be roughly estimated by assuming that all evaporation occurs at the surface z = 0, while all condensation occurs at mean condensation height  $z = \mathcal{H}_P$ :

$$\dot{\rho}(x,y,z) = E(x,y)\delta(z) - P(x,y)\delta(z - \mathcal{H}_P), \quad E = P, \quad E \equiv \frac{1}{\mathcal{S}} \int_{\mathcal{S}} E(x,y)d\mathcal{S}, \quad P \equiv \frac{1}{\mathcal{S}} \int_{\mathcal{S}} P(x,y)d\mathcal{S}, \quad (14)$$

$$\frac{1}{\mathcal{S}} \int_{\mathcal{M}} \frac{dh}{dt} d\mathcal{M} = -\frac{1}{\mathcal{S}} \int_{\mathcal{V}} h\dot{\rho} d\mathcal{V} \approx -Eh_s + Ph(\mathcal{H}_P) \equiv -P\Delta h_c, \quad \Delta h_c \equiv h_s - h(\mathcal{H}_P), \quad h = c_p T + Lq_v.$$
(15)

Here E(x,y) and P(x,y) are local evaporation and precipitation at the surface (kg m<sup>-2</sup> s<sup>-1</sup>) with global averages E and P; subscript s denotes surface values z = 0;  $\delta(z)$  is the Dirac delta function;  $c_p = 10^3 \text{ J kg}^{-1} \text{ K}^{-1}$  is heat capacity of air at constant pressure,  $L = 2.5 \times 10^6 \text{ J kg}^{-1}$  is latent heat of vaporization. For  $\mathcal{H}_P \approx 2.5 \text{ km}$  (Makarieva et al., 2013a) and  $q_v(\mathcal{H}_P) \ll q_{vs}$  we have  $-P\Delta h_c = -P(c_p\mathcal{H}_P\Gamma + Lq_{vs}) \approx -1 \text{ W m}^{-2}$ . Here  $q_{vs} = 0.0083$  corresponds to global mean surface temperature  $T_s =$ 288 K and relative humidity 80%; mean tropospheric lapse rate is  $\Gamma = 6.5 \text{ K km}^{-1}$ . Global mean precipitation P (measured in a system of units where liquid water density  $\rho_w = 10^3 \text{ kg m}^{-3}$  is set to unity) is equal to  $P \sim 1 \text{ m year}^{-1}$ , which in SI units corresponds to  $P = 3.2 \times 10^{-5} \text{ kg m}^{-2} \text{ s}^{-1}$ . A more sophisticated estimate<sup>5</sup> of integral (15) presented in Appendix A yields  $-1.6 \text{ W m}^{-2}$  with an accuracy of about 30%.

These estimates show that the enthalpy term cannot be neglected in Eq. (10) on either theoretical or quantitative grounds. By absolute magnitude the integral (15) is greater than one third of the total atmospheric power  $W \approx 4 \text{ W m}^{-2}$  estimated by Laliberté et al. (2015) for the MERRA re-analysis (3.66 W m<sup>-2</sup>) and the CESM model (4.01 W m<sup>-2</sup>).

Laliberté et al. (2015) first calculated the mass integral of Tds/dt from the right-hand side of Eq. (10), then calculated  $\mu dq_T/dt$  from atmospheric data and then used the obtained values and again Eq. 10 to estimate the total atmospheric power as  $-(1/S) \int_{\mathcal{M}} (1/\rho) (dp/dt) d\mathcal{M}$ . In such a procedure, putting  $\int_{\mathcal{M}} (dh/dt) d\mathcal{M} = 0$  should have overestimated W by about 1.6 W m<sup>-2</sup>. Since the omitted term is proportional to the global precipitation rate, it is crucial not only for a correct estimate of the mean value of W, but also for the determination of any trends related to precipitation.

Note also that even in the correct form, with the enthalpy term retained, Eq. (10) does not provide a theoretical constraint on W. This equation is an identity: it essentially defines ds/dt in terms of measurable atmospheric data. As is clear from Eq. (8), W can be estimated from the same data directly without involving entropy, see Section 5.

#### 3.2 Material derivative of pressure dp/dt and atmospheric power

Equations (2)-(8) clarify the relationship between atmospheric power and dp/dt (W m<sup>-3</sup>). Global atmospheric power W is, in the general case, not equal to the mass integral of  $-(1/\rho)dp/dt$  (or volume integral of dp/dt, which is the same since

<sup>&</sup>lt;sup>5</sup>The physical meaning of the negative sign of the enthalpy integral (15) can be understood as follows. This integral describes how much air belongs to the air parcels decreasing their enthalpy (dh/dt < 0) compared to how much air belongs to the air parcels increasing their enthalpy (dh/dt > 0). In the absence of phase transitions these masses are equal. There is as much gas going upwards, as there is going downwards. Hence,  $\int_{\mathcal{M}} (dh/dt) d\mathcal{M} = 0$ . In the presence of condensation and evaporation, these masses are not equal. Water vapor is "created" by evaporation at the planetary surface and then "destroyed" by condensation at some height  $\mathcal{H}_P > 0$ . In the result, at any height z > 0 there is, on average, more gas going upwards than downwards. In the ascending air dh/dt 

 $d\mathcal{M} = \rho d\mathcal{V}$ ). While the definition of material derivative (5) includes partial derivative over time, we have seen that atmospheric work output W does not depend on  $\partial p / \partial t$ , see Eqs. (8), (7). Indeed, from Eq. (8) and (5) we find:

$$W = -\frac{1}{\mathcal{S}} \int_{\mathcal{V}} \frac{dp}{dt} d\mathcal{V} + \frac{1}{\mathcal{S}} \int_{\mathcal{V}} \frac{\partial p}{\partial t} d\mathcal{V} + I_S.$$
(16)

In the steady state, when the volume of the atmosphere is constant, the last two terms in the right-hand side of Eq. (16) are zero. Since the distribution of pressure versus altitude in the atmosphere is approximately exponential and since  $I_S$  is proportional to  $p_T$ , by choosing a sufficiently small upper boundary pressure  $p_T$  it is possible to ensure that the instantaneous value of  $I_S$ (9) is arbitrarily small compared to the instantaneous W (see Fig. 6d in Appendix C). Everywhere below we put  $I_S = 0$ .

The long-term average of the second integral in the right-hand side of (16) is also zero. However, its instantaneous value depends on temperature tendency  $\partial T/\partial t$  and is not necessarily negligible compared to W. This term describes pressure changes that are not related to power output but reflect how internal energy of the atmosphere varies with changing atmospheric volume and/or mass (see Appendix C for details).

Locally, even in the stationary case when  $\partial p/\partial t = 0$ , material derivative of pressure  $-dp/dt \equiv -\mathbf{v} \cdot \nabla p$  is not equal to power output per unit volume of the local air parcel  $(pd\tilde{V}/dt)/\tilde{V}$  (W m<sup>-3</sup>). From Eq. (7), which holds true for an arbitrary volume  $\mathcal{V}$ , we have

$$p\frac{1}{\tilde{V}}\frac{d\tilde{V}}{dt} = p(\nabla \cdot \mathbf{v}) = -\mathbf{v} \cdot \nabla p + \nabla \cdot (p\mathbf{v}).$$
(17)

These magnitudes have different physical meaning. While  $p(\nabla \cdot \mathbf{v})$  describes power output per unit volume of the local air parcel,  $-\mathbf{v} \cdot \nabla p$ , describes power output per unit volume of the local pressure gradient. It is work performed per unit time by the force of the local pressure gradient on the air contained in the considered unit volume. Global integrals of  $-\mathbf{v} \cdot \nabla p$  and  $p(\nabla \cdot \mathbf{v})$  coincide and equal global atmospheric power W when  $I_S$  in (8) is zero.

#### 4 Revisiting the current understanding of the atmospheric power budget

We now show how total atmospheric power (8) can be decomposed into distinct terms. In Eq. (8) the expression  $-\mathbf{v} \cdot \nabla p = -\mathbf{u} \cdot \nabla p - \mathbf{w} \cdot \nabla p$  represents total work performed by the pressure gradient per unit time per unit air volume. The horizontal pressure gradient generates the kinetic energy of the horizontal wind  $\mathbf{u}$ . The vertical pressure gradient generates the kinetic energy of the vertical energy of air in the gravitational field.

In hydrostatic equilibrium we have

$$\nabla_z p = \rho \mathbf{g},\tag{18}$$

where  $\rho = NM$  is air density (kg m<sup>-3</sup>), M is air molar mass (kg mol<sup>-1</sup>). In the real atmosphere due to the presence of non-gaseous water the air distribution deviates from Eq. (18) such that we have  $\nabla_z p = (\rho + \rho_l) \mathbf{g}$  and in Eq. (8)  $-\mathbf{w} \cdot \nabla p = -\rho \mathbf{w} \cdot \mathbf{g} - \rho_l \mathbf{w} \cdot \mathbf{g}$ , where  $\rho_l$  is mass density of the non-gaseous water in the air.

Term  $-\rho \mathbf{w} \cdot \mathbf{g} = -\rho \mathbf{v} \cdot \mathbf{g}$  represents the vertical flux of air: it is positive (negative) for the ascending (descending) air. Recalling that  $\mathbf{g} = -q\nabla z$  and using the divergence theorem and the stationary continuity equation (13) we can write

$$W_P \equiv -\frac{1}{\mathcal{S}} \int\limits_{\mathcal{V}} \rho \mathbf{w} \cdot \mathbf{g} d\mathcal{V} = \frac{1}{\mathcal{S}} g \int\limits_{\mathcal{V}} \rho \mathbf{v} \cdot \nabla z d\mathcal{V} = \frac{1}{\mathcal{S}} g \int\limits_{\mathcal{S}} \mathbf{n} \cdot (\mathbf{v} \rho z) d\mathcal{S} - \frac{1}{\mathcal{S}} g \int\limits_{\mathcal{V}} z \dot{\rho} d\mathcal{V}.$$
(19)

The surface integral in (19) is taken at the Earth's surface (here it is zero because z = 0) and  $z = z(p_T)$  (here it is also zero, because  $\rho \mathbf{n} \cdot \mathbf{v} = 0$ ).

For a dry atmosphere where  $\dot{\rho} = 0$ , the last volume integral in Eq. (19) is zero and  $W_P = 0$ : indeed, in this case at any height z there is as much air going upwards as there is going downwards<sup>6</sup>. In a moist atmosphere, evaporation  $\dot{\rho} > 0$  (condensation  $\dot{\rho} < 0$ ) makes a negative (positive) contribution to  $W_P$ . This is because  $W_P$  reflects the work of water vapor as it travels from the level where evaporation occurs (where water vapor arises) to the level where condensation occurs (where water vapor disappears). When condensation takes place above where evaporation occurs, the water vapor expands as it moves upwards towards condensation, and the work is positive.

When evaporation occurs at the Earth's surface z = 0,  $W_P$  (19) is equal to  $P_g \mathcal{H}_P$ , where  $\mathcal{H}_P$  is the global mean height of condensation. It is natural to call  $W_P$  the "gravitational power of precipitation". Pauluis et al. (2000) estimated the value of  $W_P$  for the tropics. Their estimate was revised by Pauluis and Dias (2012) and by Makarieva et al. (2013a) who also estimated global  $W_P$  as 0.8 W m<sup>-2</sup>. (Here we revised this latter estimate upwards by 20%, see Appendix A).

The stationary power budget for a hydrostatic atmosphere can be written as

$$W = -\frac{1}{\mathcal{S}} \int_{\mathcal{V}} \mathbf{v} \cdot \nabla p d\mathcal{V} \equiv W_K + W_P, \tag{20}$$

$$W_K \equiv -\frac{1}{\mathcal{S}} \int_{\mathcal{V}} (\mathbf{u} \cdot \nabla p + \rho_l \mathbf{w} \cdot \mathbf{g}) d\mathcal{V} \approx -\frac{1}{\mathcal{S}} \int_{\mathcal{V}} \mathbf{u} \cdot \nabla p d\mathcal{V},$$
(21)

$$W_P \equiv -\frac{1}{\mathcal{S}} \int_{\mathcal{V}} \rho \mathbf{w} \cdot \mathbf{g} d\mathcal{V} = -\frac{1}{\mathcal{S}} \int_{\mathcal{V}} gz \dot{\rho} d\mathcal{V} = Pg\mathcal{H}_P, \quad P \equiv -\frac{1}{\mathcal{S}} \int_{z>0} \dot{\rho} d\mathcal{V}.$$
(22)

Equations Eqs. (20)-(22) and their derivation have not been previously published. These equations clarify the physical meaning of the atmospheric power budget. While we define W (4) as the power output of ideal gas parcels, this power includes  $W_P$  – the gravitational power of precipitation. The meaning is that hydrometeors perform work at the expense of their potential energy. To acquire this energy, a corresponding amount of water vapor must be raised by air parcels. We can also see that  $W_P$ does not depend on the interaction between the air and the falling hydrometeors. This term would be present in the atmospheric power budget even if hydrometeors were experiencing free fall and did not interact with the air at all (such that no frictional dissipation on hydrometeors occurred).

The term  $-\rho_l \mathbf{w} \cdot \mathbf{g}$  in Eq. (21) is not related to the gravitational power of precipitation. It describes kinetic energy generation on the vertical scale of the order of the atmospheric scale height  $\mathcal{H} \equiv -p/(\partial p/\partial z) = RT/(Mg) \sim 10$  km. This energy is generated because the vertical air distribution deviates from the hydrostatic equilibrium (18). Hydrometeors act as

<sup>&</sup>lt;sup>6</sup>Note that using the continuity equations for dry air  $\nabla \cdot (\rho_d \mathbf{v}) = 0$  and water vapor  $\nabla \cdot (\rho_v \mathbf{v}) = \dot{\rho}$ , where  $\rho_d$  and  $\rho_v$  are densities of dry air and water vapor, we find from Eq. (19) that  $W_P = -(1/S) \int_{\mathcal{V}} \rho_v \mathbf{w} \cdot \mathbf{g} d\mathcal{V}$ .

resistance preventing the pressure difference  $\Delta p \sim \rho_l g \mathcal{H}$  from converting to the kinetic energy of a vertical wind. In the atmosphere on average  $\rho_l/\rho \sim 10^{-5}$  (Makarieva et al., 2013a). Without hydrometeors, the non-equilibrium pressure difference  $\Delta p \sim 10^{-5}\rho g \mathcal{H} \sim 1$  Pa would produce maximum vertical velocity of about  $w_m \sim 1$  m s<sup>-1</sup> ( $\rho w_m^2/2 = \Delta p$ ). This is two orders of magnitude larger than the characteristic vertical velocities  $w \sim 10^{-2}$  m s<sup>-1</sup> of large-scale air motions. (Hydrometeors thus inhibit vertical motion in a similar way turbulent friction at the surface inhibit horizontal air motion. For example, the observed meridional surface pressure differences of the order of  $\Delta p_h \sim 10$  hPa in the tropics, if friction were absent, could have produced maximum horizontal air velocities of about  $u_m \sim 40$  m s<sup>-1</sup> ( $\rho u_m^2/2 = \Delta p_h$ ).) Quantitatively,  $-\rho_l \mathbf{w} \cdot \mathbf{g}$  is less than 1% of W and can be neglected: its volume integral taken per unit surface area is less than  $\rho_l g \mathcal{H} w \sim 10^{-5} p w \sim 10^{-2}$  W m<sup>-2</sup>, where  $p = \rho q \mathcal{H} = 10^5$  Pa is air pressure at the surface.

In some publications kinetic power is assumed equal to the volume integral of dp/dt (e.g., Robertson et al., 2011, their Eq. 1). Here we have seen that the long-term mean of this integral is equal to total power W, Eq. (16). However, the instantaneous value of this integral is not equal to the instantantaneous value of W, see also Appendix (C). In other studies kinetic power is correctly defined and estimated from horizontal velocities as  $W_K$  (21) (see, e.g., Boville and Bretherton, 2003; Huang and McElroy, 2015). In such studies  $W_K$  is sometimes confused with total atmospheric power: i.e. in the total power budget the gravitational power of precipitation,  $W_P$ , is overlooked (e.g., Huang and McElroy, 2015, their Fig. 10). We also note that the gravitational power of precipitation  $W_P$  has not been explicitly identified in past studies assessing the conversion rates between available potential and kinetic energies in the framework of the Lorenz energy cycle (see, e.g., Kim and Kim, 2013, and references therein).

The fact that kinetic power  $W_K$  (21) depends on horizontal and not vertical velocities is essential for comparing theory and observations. Horizontal pressure gradients and wind velocities are observed directly, while vertical velocities are inferred only indirectly and with significant uncertainty (see Appendix C).

## 5 Observation-based estimates of global atmospheric power

## 5.1 Total power W

In meteorological databases including the MERRA dataset MAI3CPASM that we used (see Appendix B for details), dp/dt is often represented as a separate variable named *pressure velocity* (omega). We estimated W as

$$W = \langle \Omega \rangle, \ \ \Omega \equiv -\frac{1}{\mathcal{S}} \int_{\mathcal{V}} \omega d\mathcal{V}, \ \ \omega \equiv \frac{dp}{dt} \equiv \frac{\partial p}{\partial t} + \mathbf{v} \cdot \nabla p.$$
<sup>(23)</sup>

Time averaging denoted as  $\langle \rangle$  was made over seven years, 2009-2015. Using instantaneous omega values provided each three hours at 42 pressure levels on the  $1.25^{\circ} \times 1.25^{\circ}$  grid we obtained two estimates,  $W_1 = 2.97$  W m<sup>-2</sup> and  $W_2 = 3.24$  W m<sup>-2</sup> depending on how the local omega value at the Earth's surface was estimated. The value of  $W_1 \equiv \langle \Omega_1 \rangle$  was obtained by linearly extrapolating the omega value from the two nearest pressure levels, see Eq. (B3), while  $W_2 \equiv \langle \Omega_2 \rangle$  was obtained assuming

**Table 1.** MERRA-based and theory-derived estimates of the global atmospheric power budget  $W = W_K + W_P$  (W m<sup>-2</sup>) and the enthalpy term  $-P\Delta h_c$  (15) (W m<sup>-2</sup>). Subscripts 1 and 2 refer to the two means of estimating W and  $W_K$ , see Section 5 and Appendix B, Eqs. (B3), (B12) for 1 and (B4), (B13) for 2.

| # | Source of estimate                                             | W          | $W_K$      | $W_P$            | $-P\Delta h_c$    |
|---|----------------------------------------------------------------|------------|------------|------------------|-------------------|
| 1 | MERRA 2009-2015 (W <sub>1</sub> , W <sub>K1</sub> )            | 2.97       | 2.70       | $0.27^{*}$       | $-0.69^{\dagger}$ |
| 2 | MERRA 2009-2015 (W <sub>2</sub> , W <sub>K2</sub> )            | 3.24       | 2.53       | $0.71^{*}$       | $-0.42^{\dagger}$ |
| 3 | MERRA 2009-2015 (mean)                                         | 3.1        | 2.6        | $0.5^*$          | $-0.6^{\dagger}$  |
| 4 | Theoretical estimates based on $P = 0.96$ m year <sup>-1</sup> | $4.8^{\P}$ | $3.8^{\$}$ | $1.0^{\ddagger}$ | $-1.6^{\ddagger}$ |
| 5 | Relative difference $(#4 - #3)/#3$                             | 0.4        | 0.3        | 0.5              | 0.6               |

\* estimated as the difference between MERRA-based W and  $W_K$ ;<sup>†</sup> estimated as the difference between MERRA-based W and  $W_L = 3.66$  W m<sup>-2</sup> of Laliberté et al. (2015);<sup>‡</sup> estimated in Appendix A;<sup>§</sup> estimated from Eq. (25);<sup>¶</sup> estimated as the sum of the theoretical  $W_K$  and  $W_P$ .

that omega at the surface is equal to the surface pressure tendency  $\partial p_s/\partial t$ , see Eq. (B4). The results are shown in Fig. 1a and Table 1.

Laliberté et al. (2015) using their Eq. (10) estimated total atmospheric power from the MERRA database as  $W_L = 3.66 \text{ W m}^{-2}$ . Both  $W_1$  and  $W_2$  are significantly smaller. We have  $W_1 - W_L = -0.69 \text{ W m}^{-2}$  (19%) and  $W_2 - W_L = -0.42 \text{ W m}^{-2}$  (12%). As discussed in Section 3, the difference between W and  $W_L$ , caused by the omission of the enthalpy term in Eq. (10), should be equal to the enthalpy term (15) of about  $-1.6 \text{ W m}^{-2}$ . The actual difference is the same order of magnitude with our estimate (15) but is about 60% smaller, Table 1.

This discrepancy may be a consequence of Laliberté et al. (2015) using different MERRA datasets (MAI3NECHM, MAI6NVANA, MAT3NVCHM). These data have a  $0.67^{\circ} \times 0.5^{\circ}$  lon-lat resolution and are provided on 72 vertical levels. However, pressure velocity is not provided. It was calculated by Laliberté et al. (2015) from horizontal velocities using a correction procedure to ensure mass conservation (see the Supplementary Materials of Laliberté et al., 2015, for details). The impact of this procedure on the obtained results was not investigated. We employed the "ready-to-use" omega values from the MERRA dataset providers. Another reason for the discrepancy is the resolution of the data – this is a topic we examine further below.

#### 5.2 Kinetic power $W_K$

Two estimates of kinetic power were obtained (Fig. 1a and Table 1):  $W_{K1} = 2.70 \text{ W m}^{-2}$  by extrapolating  $\mathbf{u} \cdot \nabla p$  linearly to the surface from the two nearest pressure levels, see Eq. (B12), and  $W_{K2} = 2.53 \text{ W m}^{-2}$  assuming  $\mathbf{v} = 0$  and  $\mathbf{u} \cdot \nabla p = 0$  at the surface, see Eq. (B13). The second estimate is close to the long-term mean kinetic power 2.46 W m<sup>-2</sup> obtained by Huang and McElroy (2015) for 1979-2010.

The difference between the estimates  $W_1$  and  $W_2$  and  $W_{K1}$  and  $W_{K2}$  stems from different assumptions about the value of the integrated quantity ( $\omega$  or  $\mathbf{u} \cdot \nabla p$ ) at the surface. The surface layer averages about 13 hPa higher pressure than  $p_1 = 1000$  hPa,