# Peer review of "Quantifying the global atmospheric power budget"

_Atmospheric Chemistry and Physics, 2016_

## Referee Comment (RC1) · Anonymous Referee #1 · 4 May 2016

Summary: In this study, the authors attempt to examine the gap between the gravitational power of precipitation, which is estimated as the total atmospheric power - kinetic power from the MERRA dataset, and another independent gravitational power of precipitation estimated from the surface precipitation data. While I can see a good merit of this work, I found the paper contains several loopholes that need to be clarified before the manuscript can be accepted for publication. Also, the presentation of this work is somewhat confusing, and can be simplified substantially to make it clearer. My concerns are given as bellow:

Major concerns:

1. The evaluation of the gravitational power of precipitation (GPP) as presented in Appendix A, which is used to verify the GPP estimated from the MERRA data, contains a significant source of uncertainties as it depends so much on different input parameters as listed in Appendix A. Likewise, the GPP estimated from MERRA also depends strongly on the data resolution, the number of vertical levels, or the numerical approximations. Before trying to explain the discrepancies between GPP obtained from GPCP data and the GPP obtained from the MERRA data, the authors should at least quantify the errors in all of your numbers. While the authors claim that the uncertainty of your estimated GPP from GPCP is 30%, there is no guarantee that the difference between the two GPP estimations will be statistical significance. Afterall, 30% of 1 W m-2 is 0.3, and so it could be anything from 0.7-1.3 W m-2, which may be comparable to the GPP computed from the MERRA data;

2. Estimations of the total atmospheric power $W$ and $W_K$ are subject to similar uncertainties as mentioned in my comment # 1 above. At resolution of 1.25 degree and 42 vertical levels, any global estimation of the total integrated energy and kinetic energy contains large variation, let alone the difference between two. Have the authors tried the NCEP reanalysis or ECMWF dataset at different resolutions to see how sensitive your estimations are? As long as we don't have reliable estimation of $W, W_K$, and GPP, explanation for the difference would provide little scientific value.

3. The derivation of the total atmospheric power given by Eq. (7)-(8) is unnecessarily complicated. I can directly obtain Eq. (7) from Eq .(2) by noting simply that $\int p\ dV/dt=\int p\ d(\delta x\ \delta y\ \delta z)/dt=\int p(\nabla \cdot v)dV$. Not sure why the authors present their argument in such a lengthy and confusing way.

4. The authors criticize Laliberte et al. (2015)'s estimation of the integral of $dh/dt$, as they believe that it is not $dh/dt=0$ but should be $\partial h/ \partial t=0$ for a stationary budget. However, my understanding of Laliberte et al.'s study is that the total derivative that Laliberte et al used is in the context of global integration. So if you define $H=\int h\ dV$, then $dH/dt=\int \partial h/\partial t\ dV$, since the total volume

is fixed in time. As such, Laliberte et al.'s global stationary approximation is consistent with your local stationary approximation.

Minor concern:

The practice of putting a dot on a variable to represent sources/sinks is too confusing, as the dot often denotes time derivation. Equation such $\dot N =dN/dt+N(\nabla \cdot v)$ is perplexing. The authors should replace all such dotted source/sink by different symbols to avoid the confusion.

---

## Referee Comment (RC2) · Anonymous Referee #2 · 10 May 2016

This manuscript looks at the power budget in the MERRA reanalysis over the last 7 years. It is generally poorly written and way too long for the arguments being made. In its current state, it does not stand up as a contribution worthy of the high standards of publication for ACP. Based on my comments (to be found below), I do not recommend this manuscript for publication at ACP.

**Key Comments:**

1. Section 2 is both way too complicated and appears to be wrong. Following Vallis' (2006) notation:

$$W = \int_{\mathcal{V}} p \frac{d\alpha}{dt} \rho d\mathcal{V} = \int_{\mathcal{V}} p \left( \partial_t(\rho\alpha) + \nabla \cdot (\rho\alpha\mathbf{v}) - \alpha S_\rho \right) d\mathcal{V},$$

where $S_\rho = \partial_t(\rho) + \nabla \cdot (\rho \mathbf{v})$ is the local sources and sinks of mass. Now, $\alpha\rho = 1$ so

$$W = \int_{\mathcal{V}} p \left(\nabla \cdot (\mathbf{v}) - \alpha S_\rho\right) d\mathcal{V} = \int_{\mathcal{V}} \left(\nabla \cdot (p\mathbf{v}) - \mathbf{v} \cdot \nabla p - \alpha p S_\rho\right) d\mathcal{V}.$$

This is the same form as in equation (8). But it depends explicitly on $S_\rho$, contrary to the authors' claim. Why this contradiction? The problem in the authors' derivation comes in part from equation (3). While it is true that $\sum_i d\tilde{N}_i/dt = 0$, it is not true that $\sum_i T_i d\tilde{N}_i/dt = 0$, unless the atmosphere is isothermal. But it is exactly what's used to convert the last term in equation (2) to the last term in equation (4). $\tilde{V} = \tilde{N}/N$ has units of m$^3$ (parcel$^{-1}$). To compute work, however, we need the specific volume with units of m$^3$ (kg$^{-1}$). So we have to introduce a new quantity, the mass per parcel $\tilde{m}$ so that the specific volume is $\tilde{V}/\tilde{m}$. Then the expression for work (equation 4) with the same units as in Vallis (2006) reads:

$$W = \frac{1}{\mathcal{S}} \int_{\mathcal{V}} p \frac{\tilde{m}}{\tilde{V}} \frac{d(\tilde{V}/\tilde{m})}{dt} d\mathcal{V}.$$

But the continuity equation (6) also requires fixing. Since, $N$ has units of mol m$^{-3}$ then equation (6) is an equation for mass conservation only if the molar mass $\tilde{m}/\tilde{N}$ is constant. But here the authors are, among other things, concerned about the effect of moisture on the work and moist air, unlike dry air, has an inhomogeneous in molar mass. The continuity equation (6) should then read:

$$\partial_t((\tilde{m}/\tilde{V})) + \nabla \cdot (\mathbf{v}(\tilde{m}/\tilde{V})) = \dot{\tilde{m}}(N/\tilde{N}) + \tilde{m}/\tilde{N}\dot{N} - (\tilde{m}N/\tilde{N}^2)\dot{\tilde{N}}$$

where the right hand side is the local sources and sinks of mass. With these fixes, the expression for work will look exactly like in Vallis (2006) and will depend on the sources and sinks of mass.

2. Section 3.1. This section is also way too complicated. After the first paragraph, one can jump directly to the top of page 5. Now equation (15) is not wrong

per se. However, the Makarieva et al. (2013) analytical derivation is somewhat meaningless when applied to reanalysed data: this can be evaluated directly. This is exactly what I have done for the purpose of this review. Using the 1 hourly vertically integrated budgets provided from the data archive, one can compute the integral $\int_S \bar{h}\bar{\rho}d\mathcal{S}$, where the overline indicates vertically integrated fields. In the reanalysis, $\bar{\rho} \neq 0$ because of the analysis step. In MERRA, this is provided directly. In the MERRA documentation it is indicated that this $\bar{\rho}$ includes both the effect of $E - P$ and adjustments needed to represent the observed surface pressure field accurately. It therefore includes the effect described by the authors. This quantity for 1980-1985 is 0.2 W/m$^2$. Adding the vertical dependence would likely be a second order effect since $E - P$ is mostly driven by horizontal and temporal variability. This simple analysis performed using the output from the MERRA product seems to show that Appendix A is likely to be inaccurate (0.2 is not within 30% of 1.6). In any case, this issue was discussed at length by Trenberth (see his papers in the 1990's) and the proposed solution is to modify the winds so that the continuity equation does not have a source term. I had a hard time finding this but you mention that Laliberte et al (2015) might have done something like this. In this case, I do believe that $\int_{\mathcal{M}} dh/dt d\mathcal{M} = 0$ makes sense since it is an exact derivative.

3. Computing the work from MERRA data. As mentioned before, the MERRA product has many vertically integrated budget variables that allow one to quantify each one of the term in the energy equation. For this review, I've looked at the kinetic energy generation 1980-1985 and the yearly average gives 3.40-3.48 W/m$^2$ for the integral of $\omega\alpha$ and 3.6-3.8 W/m$^2$ when including the kinetic energy generation from the analysis step. The kinetic energy generation is balanced by damping from the numerical dissipation, the dynamical remapping and the physically parametrized frictional dissipation. This means that the estimates provided in section 5.1 are substantial underestimates.

4. In section 5.1, I do not see the use for $W_1$. And why not use $\omega_s = \partial_t p_s + \mathbf{v}_s \cdot \nabla_H p_s$, with $\nabla_H$ being the horizontal gradient? The $p_s$ and $\mathbf{v}_s$ are both available and this is the right expression. Maybe that could fix their underestimate of $W$.

5. The way I see it, there are approximately three manuscripts in this study. The first one, sections 2 and 3 as well as Appendix A, consist mostly of derivations that are either flawed or mostly useless for this study. The second paper is more akin to a white paper and comprises sections 6.1 and 6.2. Now, sections 1, 4, 5 and the very beginning of section 6 as well as Appendix B and C are self-contained and describe an original treatment of reanalysis data. Appendix C could be moved up after section 4. If they wish to submit their results to another publication, I would recommend that the authors focus on these sections and perform their analysis on the whole of MERRA (1979-2015).

6. Finally, I'm not sure the following sentence is logically true: "The fact that $W_{Kc}$ is likewise higher than our MERRA-derived kinetic power, testifies in favor of the theoretical estimate". All it means is that $W_{Kc}$ is potentially a right upper bound. The only way to check whether it is the right upper bound would be to either verify if it holds on other Earth-like planets or using simulations with increasing resolution and seeing that it describes the scaling. As I said before, the last two sections of this manuscript are really too speculative in their current form and they are dragging down the original results described in sections 5.

---

## Referee Comment (RC3) · Anonymous Referee #3 · 13 May 2016

The manuscript is poorly written and requires substantial improvement before publication. The authors misrepresent part of their results as a new analysis, while they have been previously discussed in the literature.

Main comments:

**1. Appropriation in the main result:**

The manuscript states pretty explicitly that the main contribution here is

"Starting from the definition of mechanical work for an ideal gas, we present a novel derivation linking global wind power to measurable atmospheric parameters. The resulting expression distinguishes three components: the kinetic power associated with horizontal motion, the kinetic power associated with vertical motion and the gravita-

**tional power of precipitation."**

as it is stated in the abstract. This claim is repeated on multiple occasions. I assume that this specifically refer to the equation (20-22), which the authors claim that "*Equations Eqs. (20)-(22) and their derivation have not been previously published.*"

These equations are presented in Pauluis etal. (2000) (See equations (2), (4), (8) and (10). See also equations (4) and equation (6) of Pauluis and Held (JAS, 2002)). It is very troublesome that the authors fail to mention that equations (20-22) are presented in Pauluis etal. (2000) despite the fact that this pa

The appropriation is not limited to the equations, but extends to some of the arguments presented. For instance, the authors relate the claim

"The meaning is that hydrometeors perform work at the expense of their potential energy. To acquire this energy, a corresponding amount of water vapor must be raised by air parcels. We can also see that WP does not depend on the interaction between the air and the falling hydrometeors. This term would be present in the atmospheric power budget even if hydrometeors were experiencing free fall and did not interact with the air at all (such that no frictional dissipation on hydrometeors occurred). "

This points is made previously ( and more clearly) in Pauluis etal. JAS (2000, p. 991):

"The dissipation by precipitation can be thought as proceeding in two steps. First, water is lifted by the atmospheric circulation, increasing its potential energy. Then, during precipitation, the potential energy of condensed water is transferred to the ambient air where it is dissipated by molecular viscosity in the microscopic shear zone around the hydrometeors."

To put it bluntly, the authors are presenting as their own an analysis that was done by others, and in doing so, are misleading their reader.

**2. Discussion of Laliberte etal. (2015)|**

The discussion of Laliberte etal. (2015) is very esoteric and does not pertain much to the rest of the discussion. Section 3.2 is a very minor point. It is fairly well-known that the integral of dp/dt is only equal to the work performed for a steady system, an assumption that is clearly stated in Laliberte etal. As for section 3.1, there are several problems with the authors analysis. First, it should be clearly stated that the global integral of dh/dt is indeed zero in the absence of mass source and sink in the continuity equation. This is the assumption made in Laliberte etal. It is also the continuity equation used in the MERRA Reanalysis. Hence, the authors should explicitly acknowledge that the claim that the integral of dh/dt is indeed correct within the assumptions made in the MERRA Reanalysis.

Second, it is perfectly valid to question the impact of mass source and sink on the framework of Laliberte etal., but this should be done clearly. In particular, The Bernoulli equation is an equality with 4 different terms. Changing the mass conservation does not only affect the global integral of dh/dt, but also that of ds/dt and dq/dt. The authors here assume -without proof- that the change in the enthalpy integral would be reflected solely in the work output.

The broader issue here is that the discussion of section 3.1. and 3.2. is presented without context and incomplete. It could only be understood by very few potential readers. It makes the paper unnecessarily confusing and should be removed.

**3. Overal structure:**

The paper is poorly constructed. It is mainly three separate studies. Sections 2-4 attempt a theoretical discussion of the issues that mostly reprise previous work. It is unnecessarily confusing. Section 5 is the main 'new' result. The computation done are fairly routine, and the result in line with what we know. The inability of the authors to produce a consistent figure for Wp is distressing and should be better addressed in the revision. Section 6 is a lengthy disgression which is mostly a repeat of the authors previous work.

My recommendation here would be to simplify section 2 and 4, drop section3 and expand on section 5. Section 6 could be clarified as well.

---

## Referee Comment (RC4) · Anonymous Referee #4 · 20 May 2016

**Summary and recommendation**

The main aim of this paper is to clarify the atmospheric power budget by seeking to exploit the divergent character of the gaseous mass flux in order to identify those terms in the power budget that can be related explicitly to the condensation/evaporation rates. The paper makes some valid point (Sections 2 and 3), such as pointing out that a term neglected in a recent study by Laliberte et al. (2015) is not only different from zero but too large to be really negligible, but the solution proposed does not seem valid. As to section 4, which claims to revisit the current understanding of the atmospheric power budget, it merely consists in some manipulation of the equations for a hydrostatic atmosphere that arguably sheds no light on the problem. The final section is too speculative. I don't think the paper makes a meaningful contribution to the understanding of atmospheric energetics, and I therefore cannot recommend publication.

**Main comments**

- 1. Abstract and elsewhere. I believe that the authors abuse the word *power*, which is used generically for all terms that enter the energy budget, such as in: *Kinetic power associated with horizontal motion, the kinetic power associated with vertical motion, and the gravitational power of precipitation*. In discussions of ocean and atmospheric energetics, it is more usual to restrict the term 'power' to the particular energy conversion responsible for supplying external energy to the system considered, and to be explicit as to what kind of energy conversions the other term represent. For instance, the term  $u \cdot \nabla p$  is as far as a I can judge a conversion between available potential energy and kinetic energy, which is considerably more informative that 'kinetic power', and the authors should similarly clarify the physical meaning of the other terms.
- 2. The water cycle is generally regarded as making the atmospheric heat engine less efficient as the result of part of the solar forcing being expanded in lifting water vapour against the gravity field, part of which is then removed through precipitation, leaving only the residual to power the atmospheric circulation, an idea proposed by Pauluis and reprised in Laliberte et al. (2015). It seems that this should be discussed.
- 3. Remarks on the methodology. Physically, the atmospheric energy budget is best understood by introducing some kind of available enthalpy  $ape = h(\eta, q_t, p) - h_r(\eta, q_t)$ , where *h* is the moist specific enthalpy,  $\eta$  is some suitable definition of moist specific entropy, and  $q_t$  the total specific humidity, *p* is pressure, where  $h_r(\eta, q_t)$  representing the part of the total enthalpy that is not available for adiabatic conversions into kinetic energy, so that

$$dh = (T - T_r)d\eta + (\mu - \mu_r)dq_t + \alpha dp$$

As a result, it is possible to express the total power term as

$$\int_{V} p \frac{D\alpha}{Dt} \rho dV = \underbrace{\int_{V} \frac{D(p\alpha)}{Dt} \rho dV}_{=0} - \int_{V} \alpha \frac{Dp}{Dt} \rho dV = \int_{V} \frac{T - T_{r}}{T} \dot{q} dm + \int_{V} (\mu - \mu_{r}) \frac{Dq_{t}}{Dt} dm$$

where  $\dot{q}$  represents diabatic heating terms by all manner of conduction of radiation. This neglects the integral of dh/dt, but this term could be retained if desired. The passage from the first term to the second term requires  $\nabla(\rho v) = 0$ , and  $\rho v$  to the total mass flux, in order to be able to claim that the integral of  $D(p\alpha)/Dt$  vanishes, so the authors should clarify this point, as well as boundary conditions assumed by the different velocities entering the definition of v. In any case, the above formalism is usually what constitutes the starting point for linking the atmospheric power budget to a Carnot-like theory and for constraining the atmospheric power budget to solar heating, sensible heat fluxes, and condensation/evaporation process. The approach proposed by the authors seem to be guite unrelated to this standard view.

4. Sections 2 and 3. The whole point of the exercise of this exercise seems to establish that the term  $\int_V dh/dt \rho dV$  assumed to be zero in Laliberte et al. is actually not zero, and that it is too large to be neglected. I agree with this statement, but the result obtained by the authors seems unphysical. The simplest way to show that the above term is not zero is through using using standard integration by parts

$$\int_{V} \frac{dh}{dt} \rho \mathrm{d}V = \int_{V} \nabla \cdot (\rho hv) \, \mathrm{d}V - \int_{V} h \nabla \cdot (\rho v) \, \mathrm{d}V = \int_{\partial V} \rho hv \cdot ndS - \int_{V} h \nabla \cdot (\rho v) \mathrm{d}V$$

How to estimate this term depends on how the velocity v, the density  $\rho$  and enthalpy h are defined. If v is the fully barycentric velocity, and  $\rho$  the full density, then mass conservation imposes  $\nabla \cdot (\rho v) = 0$ , and the term is controlled by

boundary fluxes of enthalpy and is equal to the difference between the enthalpy evaporated minus the enthalpy precipitated. If  $\rho v$  is the mass flux of the gaseous component of moist air, then how to estimate this term is more complicated, since  $\nabla \cdot (\rho v) \neq 0$ . Physically, the term  $h \nabla (\rho v)$  is unphysical, since condensation or evaporation converts water vapour enthalpy  $h_v$  into liquid water enthalpy  $h_l$  and conversely, so should only involve the difference  $h_v - h_l = L$ , where L is latent heat, it should not involve the dry air enthalpy; the formula  $h\nabla(\rho v)$  involves the dry air enthalpy, however, which is part of the definition of h.

Physically, the result should not involve the dry air enthalpy, and should also be independent of the different constants entering the definition of the three forms of enthalpy, which the authors have not shown.

5. Section 4. I don't really understand why this decomposition is useful. Indeed, a well known consequence of making the hydrostatic approximation is to filter out the contribution of the vertical velocity to the kinetic energy. As a result, the evolution equation for the kinetic energy becomes

$$\rho \frac{D}{Dt} \frac{u^2}{2} + u \cdot \nabla p = \rho F \cdot u \tag{1}$$

so that in equilibrium

$$\int_{V} u \cdot \nabla p \, \mathrm{d}V = Friction,\tag{2}$$

which shows that only what the authors call the kinetic energy power (the conversion between kinetic energy and available potential energy) becomes relevant to understand how the atmospheric circulation is powered. As is also well known, even without the hydrostatic approximation, the budget of gravitational potential energy is zero

$$\int_{V} \rho g w \, \mathrm{d}V = 0 \tag{3}$$

where  $\rho w$  is the total mass flux, and hence decoupled from the kinetic energy budget. One may if one so desires to separate the total mass flux into gaseous and liquid components, and restrict attention to the former, for which the GPE budget becomes

$$\frac{d(GPE)}{dt}\Big|_{gas} = \underbrace{\int_{V} \rho gw \, \mathrm{d}V}_{-SW_P} + GAS \ DESTRUCTION = 0, \tag{4}$$

where  $\rho w$  is now the gaseous mass flux only, GAS DESTRUCTION means GPE sink due to destruction of water vapour mass by condensation, but that does not make it less decoupled from the horizontal kinetic energy budget, where the underlined term is what the authors call the *power of precipitation*, whatever that means. Physically, this term represents primarily a conversion with internal energy, and is not directly related to the kinetic energy of the system, making its usefulness for clarifying the atmospheric power budget dubious. Moreover, it is also well known that for a hydrostatic fluid, it is the total potential energy of the system (i.e., the enthalpy) that matters, given that large variations in gravitational potential energy are compensated by large variations in internal energy, is at odds with the common wisdom that GPE is not useful to consider on its own. The claim that GPE variations are somehow connected with kinetic energy production is odd, given that the hydrostatic approximation is unconnected to the vertical velocity field.

6. On a last note, I have a hard time accepting that the term *u* · ∇*p* is something *observable*, given that the only way to estimate this term can only be done by means of a numerical model; likewise for the internal condensation/precipitation terms.

---

## Author Comment (AC1) · 1 Jul 2016

We thank our referees for their excellent comments and apologize for the delay in addressing them. The first two authors had been in a field trip completely disconnected from the electronic world from April 24th (at which time no comments were yet available) until yesterday. We have just asked the Editor to re-open and if possible extend the discussion for one month giving the referees an opportunity to react to our replies if needed. We aim to respond to all the comments within the nearest days.

Before addressing the specific comments of the referees, below is an addition to Section 1 that is to replace the second paragraph on p. 2 in the revised manuscript. We believe that it sets the scene for our study more specifically and provides a unified perspective on several key propositions of the referees.

"Our motivation here is to clarify what is meant by *atmospheric power* and how we can assess it from observations in a moist atmosphere. We found that the available literature provides no clear answer. One approach to define atmospheric power is to consider the rate at which the kinetic energy of winds dissipates to heat. This can be done by assessing the friction term in the equations of motion. In a steady state the rate at which kinetic energy dissipates should be equal to the rate at which it is generated. Then *atmospheric power* can be defined as the *rate at which new kinetic energy must be produced to offset the dissipative effects of friction* (Lorenz, 1967, p. 97).

Following Lorenz (1967, Eq. 102), the atmospheric power (W  $m^{-2})$  in a steady state should be defined as

$$W_{I} \equiv -\frac{1}{\mathcal{S}} \int_{\mathcal{M}} \mathbf{v} \cdot \nabla p \alpha d\mathcal{M} = -\frac{1}{\mathcal{S}} \int_{\mathcal{V}} \mathbf{v} \cdot \nabla p d\mathcal{V}, \qquad (1)$$

Here *p* is air pressure, **v** is air velocity,  $\alpha \equiv 1/\rho$ ,  $\rho$  is air density, S, M and V are, respectively, Earth's surface area, total mass and total volume of the atmosphere,  $dM = \rho dV$ . Laliberté et al. (2015) termed  $W_I$  atmospheric *work output*. Sometimes atmospheric power is also referred to as *global wind power* (e.g., Marvel et al., 2013) or *kinetic energy dissipation* (Boville and Bretherton, 2003).

Lorenz (1967, Eq. 102) proposed an additional formulation for kinetic energy production, which was adopted by many researchers:

$$W_{II} \equiv -\frac{1}{S} \int_{\mathcal{V}} \mathbf{u} \cdot \nabla p d\mathcal{V}, \qquad (2)$$

where **u** is horizontal velocity of air. For a recent use of this formulation, see, e.g., Huang and McElroy (2015) and references therein. According to Lorenz (1967),  $W_I = W_{II}$ .

On the other hand, it is possible to formulate atmospheric power from a thermodynamic viewpoint as work per unit time, while work in the atmosphere is performed by the air
parcels that change their volume. With this reasoning according to Pauluis and Held (2002, Eq. 4) the atmospheric power should be defined as

$$W_{III} \equiv \frac{1}{S} \int_{\mathcal{V}} p \nabla \cdot \mathbf{v} d\mathcal{V}.$$
(3)

Pauluis and Held (2002) referred to  $W_{III}$  as to *mechanical work* (per unit time).

Meanwhile, according to Vallis (2006, Eq. 1.65), work done per unit mass is  $pd\alpha$ . Then total work performed by the atmosphere per unit time is1

$$W_{IV} \equiv \frac{1}{S} \int_{\mathcal{M}} p \frac{d\alpha}{dt} d\mathcal{M}.$$
 (4)

Here

$$\frac{dX}{dt} \equiv \frac{\partial X}{\partial t} + \mathbf{v} \cdot \nabla X \tag{5}$$

is the material derivative of the considered property X.

As we discuss below, for a dry hydrostatic atmosphere obeying the continuity equation

$$\frac{\partial \rho}{\partial t} + \nabla \cdot (\rho \mathbf{v}) = \dot{\rho} \tag{6}$$

with  $\dot{\rho} = 0$ , all four definitions of atmospheric power are equal,  $W_I = W_{II} = W_{III} = W_{IV}$ . In an atmosphere with a water cycle, the source term  $\dot{\rho}$  is not zero2. Here gas (water vapor) is created by evaporation and destroyed by condensation with a local rate  $\dot{\rho} \neq 0$  (kg m-3 s-1). As we will show, in this case each of the four candidate expressions  $W_I$ ,  $W_{II}$ ,  $W_{III}$  and  $W_{IV}$  are distinct.
<sup>1Definition (4) for atmospheric power was endorsed by two referees of this work, see doi 10.5194/acp-2016-203-RC2 and 10.5194/acp-2016-203-RC4.

 $^{2}$ Referee 1 rightfully pointed out that using the dot for the source term is potentially confusing, since in some texts the dot indicates partial derivative over time. We have however retained this dotted notation for the sources to ensure consistency in notations with the study of Laliberté et al. (2015) which we examine in detail.

To our knowledge, these distinctions have not been previously highlighted and examined. Which definition, if any, represents the "true power" of a moist atmosphere and is consistent with the thermodynamic interpretation of work? We show that confusion between definitions results in inconsistencies and errors in estimating atmospheric power and contributes to misunderstandings of the atmospheric power budget.

In Section 2 we explore how the derivation of an expression for global atmospheric power is affected by phase transitions etc."

**References**

- B. A. Boville and C. S. Bretherton. Heating and kinetic energy dissipation in the NCAR Community Atmosphere Model. J. Climate, 16:3877–3887, 2003. doi: 10.1175/1520-0442(2003) 016<3877:HAKEDI>2.0.CO;2.
- J. Huang and M. B. McElroy. A 32-year perspective on the origin of wind energy in a warming climate. *Renewable Energy*, 77:482–492, 2015. doi: 10.1016/j.renene.2014.12.045.
- F. Laliberté, J. Zika, L. Mudryk, P. J. Kushner, J. Kjellsson, and K. Döös. Constrained work output of the moist atmospheric heat engine in a warming climate. *Science*, 347:540–543, 2015. doi: 10.1126/science.1257103.
- E. N. Lorenz. *The Nature and Theory of the General Circulation of the Atmosphere*. World Meteorological Organization, 1967.
- K. Marvel, B. Kravitz, and K. Caldeira. Geophysical limits to global wind power. *Nature Climate Change*, 3:118–121, 2013. doi: 10.1038/nclimate1683.
- O. Pauluis and I. M. Held. Entropy budget of an atmosphere in radiative–convective equilibrium. Part I: Maximum work and frictional dissipation. *J. Atmos. Sci.*, 59:125–139, 2002. doi: 10.1175/1520-0469(2002)059<0125:EBOAAI>2.0.CO;2.
- G. K. Vallis. Atmospheric and Oceanic Fluid Dynamics: Fundamentals and Large-Scale Circulation. Cambridge University Press, 2006.

---

## Author Comment (AC2) · 4 Jul 2016

Here we reply to Comment 2 of Referee 2, who shows how, in their view,  $I_h \equiv \int_{\mathcal{M}} dh/dt d\mathcal{M}$  neglected by Laliberté et al. 2015 can be directly estimated from MERRA – rendering useless our theoretical estimate of  $I_h$  in Appendix A. The referee performs such an estimate, showing that it is significantly smaller than ours, and states that  $I_h$  is zero when, as possibly assumed by Laliberté et al. 2015,  $\int \dot{\rho} dz = 0$ .

We thank Referee 2 for their effort to numerically check our results. However, as we show below, the estimate obtained by Referee 2 appears to result from a misunderstanding (a confusion of the mean of a product  $\overline{f_1 \cdot f_2}$  for the product of means  $\overline{f_1} \cdot \overline{f_2}$ , which is fatal when  $\overline{f_1} = 0$ ). As such, this estimate neither disproves our theoretical result nor justifies the omission of  $I_h$  by Laliberté et al. 2015. As we clarify below, we

have demonstrated in our work that  $I_h$  is not proportional to the vertical integral of the source term  $\int \dot{\rho} dz$  and does not vanish when the latter is zero.

Comment 2 of Referee 2 [doi:10.5194/acp-2016-203-RC2] in full reads:

"2. Section 3.1. This section is also way too complicated. After the first paragraph, one can jump directly to the top of page 5. Now equation (15) is not wrong per se. However, the Makarieva et al. (2013) analytical derivation is somewhat meaningless when applied to reanalysed data: this can be evaluated directly. This is exactly what I have done for the purpose of this review. Using the 1 hourly vertically integrated budgets provided from the data archive, one can compute the integral  $\int_{S} h \bar{\rho} dS$ , where the overline indicates vertically integrated fields. In the reanalysis,  $\dot{\rho} \neq 0$  because of the analysis step. In MERRA, this is provided directly. In the MERRA documentation it is indicated that this  $\overline{\rho}$  includes both the effect of E - P and adjustments needed to represent the observed surface pressure field accurately. It therefore includes the effect described by the authors. This quantity for 1980-1985 is  $0.2 \text{ W/m}^2$ . Adding the vertical dependence would likely be a second order effect since E - P is mostly driven by horizontal and temporal variability. This simple analysis performed using the output from the MERRA product seems to show that Appendix A is likely to be inaccurate (0.2 is not within 30% of 1.6). In any case, this issue was discussed at length by Trenberth (see his papers in the 1990's) and the proposed solution is to modify the winds so that the continuity equation does not have a source term. I had a hard time finding this but you mention that Laliberte et al (2015) might have done something like this. In this case, I do believe that  $\int_{\mathcal{M}} dh/dt d\mathcal{M} = 0$  makes sense since it is an exact derivative."

We presume that the referee's agreement with our Eq. (15) pertains to the equality

$$I_h \equiv \int_{\mathcal{M}} \frac{dh}{dt} d\mathcal{M} = -\int_{\mathcal{V}} h\dot{\rho} d\mathcal{V} \equiv -A.$$
 (c1)
$$A \approx B \equiv \int_{\mathcal{S}} \hat{h} \hat{\rho} d\mathcal{S}.$$
 (c2)

suggesting that  $\hat{h}$  and  $\hat{\rho}$  are available from the MERRA dataset (we replaced the overline by  $\hat{\rho}$  in *B* to preserve the overline for the averages to appear below).

We need first to resolve an inconsistency between the units of our *A* and the referee's *B*. First, we note that the dot over enthalpy *h* in *B* may be a misprint since an *enthalpy* source  $\dot{h}$  appears to be an unspecified variable out of context. Next, if following the referee's indication that  $\hat{\phantom{a}}$  in *B* denotes *vertically integrated fields* we assume that  $\hat{h} \equiv \int h dz$  and  $\hat{\rho} \equiv \int \dot{\rho} dz$ , then *B* has the units of [J s-1m], while *A* has the units of [J s-1]. So expression *B* needs some "fix" before it could be compared with *A*.

Keeping  $\hat{\rho} \equiv \int \dot{\rho} dz$ , the only way we can see to remedy *B* is to assume that  $\hat{h} \equiv \int h dz / \int dz$ , units [J kg-1] is the mean enthalpy in the air column (not the vertically integrated enthalpy [J kg-1 m]). In this case the units of *A* and *B* coincide and what the referee proposes reads

$$A \approx \int_{\mathcal{S}} \left( \frac{\int h dz}{\int dz} \int \dot{\rho} dz \right) d\mathcal{S}.$$
 (c3)

Noting that  $d\mathcal{V} = dz d\mathcal{S}$ , this implies the following replacement in A

$$\int h\dot{\rho}dz \approx \frac{\int hdz}{\int dz} \int \dot{\rho}dz.$$
 (c4)

By dividing both parts of (c4) by  $\int dz$  we find that (c4) relates the columnar mean of  $h\dot{\rho}$  to the product of columnar means of h and  $\dot{\rho}$ . The two expressions are not equivalent, since, as is well-known:

$$\overline{h\dot{\rho}} = \overline{h} \cdot \overline{\dot{\rho}} + \overline{(h - \overline{h})(\dot{\rho} - \overline{\dot{\rho}})},$$

(c5)
where  $\overline{X} \equiv \int X dz / \int dz$ . The second term in the right-hand part of (c5) represents the covariance of the two variables. Indeed, we know that the enthalpy and the rate of phase transitions in the atmosphere are spatially correlated: h is higher at the surface where evaporation occurs and  $\dot{\rho} > 0$  and lower in the upper atmosphere where condensation occurs and  $\dot{\rho}

---

## Author Comment (AC3) · 6 Jul 2016

Here we reply to Comments 3, 1 and 3 of, respectively, Referees 1, 2 and 4 addressing how the correct expression for atmospheric power W should look like. We note that the referees disagree on that matter. Referee 1 (and implicitly Referee 3) agree with our Eq. (7), which shows that W does not explicitly depend on the rate of phase transitions.

Meanwhile, Referees 2 and 4 opine, respectively, that our results either *appear to be* wrong or are unrelated to the standard view suggesting two derivations of their own. As shown below, both derivations assume that work per unit mass is equal to  $pd\alpha$ , which is not a valid assumption in the presence of phase transitions. The resulting expressions contradict not only our Eq. (7) but also the identical Eq. (4) of Pauluis and Held (2002) endorsed by Referee 3.

The revised Section 2 below accounts for the above comments of the referees. It is followed by the comments themselves and our specific replies to them. Following the suggestion of Referee 1, we unburdened Section 2 of the longer derivation of Eq. (7) from the continuity equation and the ideal gas law and derived the same result immediately from the consideration of the relative change of the air parcel's volume. However, this simpler derivation contains an implicit assumption, which, as discussed in the revised text, necessitates our original longer derivation (in the revised text it is moved to the Appendix).1

**2 Atmospheric power in the presence of phase transitions**

When going from a dry to a moist atmosphere, where, besides dry air, there is also water vapor and the non-gaseous water (condensate) present, we need to accurately

$$W_{I} \equiv -\frac{1}{S} \int_{\mathcal{M}} \mathbf{v} \cdot \nabla p \alpha d\mathcal{M} = -\frac{1}{S} \int_{\mathcal{V}} \mathbf{v} \cdot \nabla p d\mathcal{V}, \qquad (1)$$

$$W_{II} \equiv -\frac{1}{S} \int_{\mathcal{V}} \mathbf{u} \cdot \nabla p d\mathcal{V}, \qquad (2)$$

$$W_{III} \equiv \frac{1}{S} \int_{\mathcal{V}} p \nabla \cdot \mathbf{v} d\mathcal{V}, \tag{3}$$

$$W_{IV} \equiv \frac{1}{S} \int_{\mathcal{M}} p \frac{d\alpha}{dt} d\mathcal{M}, \tag{4}$$

$$\frac{dX}{dt} \equiv \frac{\partial X}{\partial t} + \mathbf{v} \cdot \nabla X. \tag{5}$$

$$\frac{\partial \rho}{\partial t} + \nabla \cdot (\rho \mathbf{v}) = \dot{\rho},\tag{6}$$

<sup>1This section follows Section 1, the relevant revised portion of which can be found in our first comment doi:10.5194/acp-2016-203-AC1. The relevant equations from the revised first section are

define velocity and density (Pelkowski and Frisius, 2011). We consider an atmosphere of total volume  $\mathcal{V}$  as composed of n macroscopic air parcels each of volume  $\tilde{V}_i$  (m3) such that  $\mathcal{V} \equiv \sum_{i=1}^{n} \tilde{V}_i = \int_{\mathcal{V}} d\mathcal{V}$ . Here  $\mathcal{V}$  can be defined as the volume bounded by the Earth's surface and the surface corresponding to some fixed pressure level  $p_T$  at the top of the atmosphere, e.g. to  $p_T = 0.1$  hPa. This is the uppermost level in many atmospheric datasets including those in the MERRA dataset. With  $\tilde{m}_d$ ,  $\tilde{m}_v$  and  $\tilde{m}_c$  being mass of, respectively, dry air, water vapor and condensate in a considered parcel, we define  $\rho \equiv \tilde{m}/\tilde{V}$  to be the air density,  $\tilde{m} \equiv \tilde{m}_d + \tilde{m}_v$ ,  $\rho_d \equiv \tilde{m}_d/\tilde{V}$ ,  $\rho_v \equiv \tilde{m}_v/\tilde{V}$ , and  $\rho_c \equiv \tilde{m}_c/\tilde{V}$  to be the condensate density.

Assuming the thermodynamic notion that work is performed due to expansion of a macroscopic body and is the product of pressure and volume change, the work of an air parcel per unit time per unit volume is

$$W_p \equiv \frac{p}{\tilde{V}} \frac{d\tilde{V}}{dt}.$$
(7)

At this point, as suggested by our Referee 1, it is tempting to note, following, for example, Batchelor (2000, p. 74), that the volume  $\tilde{V}$  of the air parcel changes as a result of movement of each element of the bounding material surface with velocity v, such that  $W_p$  becomes

$$W_p = \frac{p}{\tilde{V}} \int_{\tilde{S}} \mathbf{v} \cdot \mathbf{n} d\tilde{S} = \frac{p}{\tilde{V}} \int_{\tilde{V}} \nabla \cdot \mathbf{v} d\tilde{V} = p \nabla \cdot \mathbf{v}, \tag{8}$$

where n is the outward normal vector. The latter equality is valid in the limit of sufficiently small  $\tilde{V}$ . Then, the global atmospheric power per unit surface area can be defined and evaluated from the observed pressure and velocity of air as

$$W \equiv \frac{1}{\mathcal{S}} \sum_{i=1}^{n} W_{pi} \tilde{V}_{i} = \frac{1}{\mathcal{S}} \int_{\mathcal{V}} W_{p} d\mathcal{V} = \frac{1}{\mathcal{S}} \int_{\mathcal{V}} p \nabla \cdot \mathbf{v} d\mathcal{V}.$$
(9)

Eq. (9) is equivalent to Eq. (4) of Pauluis and Held (2002),  $W = W_{III}$ , see Eq. (3) above. Several essential comments are in order concerning the above derivation of W (9) (note that Pauluis and Held (2002) listed this equation without a derivation or reference). First, the above derivation considers the work of the expanding *air* that have pressure p. Hence, v in (9) is the velocity of the gaseous constituents of the atmosphere and not the mean velocity of gas and condensate.

Second, the derivation of Eq. (9) assumes the continuity of velocity at the parcels' boundaries, but does not assume the continuity of pressure. Indeed, in agreement with the thermodynamic definition of work (7), pressure is assumed to be the same everywhere within the air parcel although it can vary from parcel to parcel. (If one additionally requires the continuity of pressure (i.e. considers that pressure, too, varies across the parcel as velocity does) and defines work of a parcel not as  $p \int_{\tilde{S}} \mathbf{v} \cdot \mathbf{n} d\tilde{S}$  but as  $\int_{\tilde{S}} p\mathbf{v} \cdot \mathbf{n} d\tilde{S}$ , then the resulting expression for total work W would be  $(1/S) \int_{\mathcal{V}} \nabla \cdot (p\mathbf{v}) d\mathcal{V}$ , which is always zero.) For this reason, W (9) can be considered as the *definition* of macroscopic mechanical work per unit time (atmospheric power) that is consistent with the thermodynamic definition of work (7). Therefore, W (9) is a function of the temporal and spatial scale at which the macroscopic velocity  $\mathbf{v}$  is defined.

Third, quite remarkably, the above derivation of (9) appears to be invariant with respect to the presence or absence of phase transitions that change the amount of gas. That is, when deriving Eq. (9) we did not use the equality

$$W_p = \frac{p}{\tilde{V}} \left( \tilde{m} \frac{d\alpha}{dt} + \alpha \frac{d\tilde{m}}{dt} \right), \tag{10}$$

where  $\alpha \equiv 1/\rho = \tilde{V}/\tilde{m}$  is the volume occupied by unit air mass.

This is because Eq. (8) is based on an implicit assumption that the volume of the air parcel can change *only* when  $\nabla \cdot \mathbf{v} \neq 0$ , i.e. when the parcel boundaries move at different velocities *at the considered scale*.

Indeed, the standard thermodynamic interpretation of Eq. (9) is that if a certain parcel expands (positive work), the rest of the atmosphere contracts by the same amount (negative work), since the total atmospheric volume  $\mathcal{V}$  is constant. Thus, the expanding air parcels perform work on the compressing air parcels. When expansion and compression occur at different pressures, the resulting difference can be converted to mechanical work producing kinetic energy of wind which then dissipates to heat.

However, in the presence of phase transitions the situation is different. Consider for simplicity a parcel of water vapor in a still atmosphere composed of pure water vapor. Let it suddenly condense into a droplet. Despite the gas parcel has compressed into the negligible volume of the droplet, it did so *not because* some other parcel had expanded. In fact, no other air parcel performed any work on the condensing gas that suddenly reduced its volume. Instead, this reduction in volume  $d\tilde{V}/dt < 0$  occurred at the expense of the work of the intermolecular forces that ensured condensation. As condensation may occur nearly instantaneously (governed by velocities of the order of sound velocity), this type of volume change is generally *not accounted for* by the velocity divergence  $\nabla \cdot \mathbf{v}$  defined at an arbitrary scale. The question therefore arises whether the above derivation and the resulting expression  $W = W_{III}$  (9) can be generally reconciled with Eq. (10) for  $W_p$  in the presence of phase transitions.

We show in Appendix A that if we use the continuity equation and the ideal gas equation of state, the integration of  $W_p$  (10) yields Eq. (9). This is because the requirement of continuity postulates that any void space produced by condensation must be immediately filled by the expanding adjacent air parcels. This nearly instantaneous non-equilibrium expansion and the associated positive work of the air parcels precisely cancels the negative work of the intermolecular forces that make the air compress because of condensation. The requirement of continuity at a given spatial and temporal scale masks the two processes that occur at a different scale specified by the condensation process.

As we discuss in Section 6, since condensation is not spatially uniform, it is during

such condensation-induced instantaneous expansion of the neigboring air parcels that the macroscopic pressure gradients may form to ultimately drive the atmospheric circulation and *determine* the value of atmospheric power W (9). In the conventional view the circulation arises when some air parcels aquire an opportunity to expand as they are receiving more heat (and thus warming more) than the others. The condensation-driven circulation arises when some air parcels acquire an opportunity to expand as the adjacent space is suddenly freed from the condensed gas. Notably, Eq. (9) does not carry information about the causes of circulation. It just gives the definition of macroscopic mechanical work per unit time (power) that is compatible with the thermodynamic definition (7).

Forth, Eq. (10) makes it clear that in the presence of phase transitions work done per unit mass is not equal to  $pd\alpha/dt$  (cf. Vallis, 2006, Eq. 1.65) but to

$$W_p \frac{\tilde{V}}{\tilde{m}} = W_p \alpha = p \left( \frac{d\alpha}{dt} + \frac{\alpha}{\tilde{m}} \frac{d\tilde{m}}{dt} \right) = p \left( \frac{d\alpha}{dt} + \alpha^2 \dot{\rho} \right), \tag{11}$$

where  $\dot{\rho} \equiv (d\tilde{m}/dt)/\tilde{V}$  (kg m-3 s-1) is the source term from the continuity equation (6). It describes the local rate of phase transitions. The global integral of this additional term is not zero,  $\int_{\mathcal{M}} p\alpha^2 \dot{\rho} d\mathcal{M} = \int_{\mathcal{V}} p\alpha \dot{\rho} d\mathcal{V} \neq 0$ . Therefore, expression  $W_{IV}$  (4) that neglects this term is incorrect,  $W_{IV} \neq W_{III} = W$ . It cannot be used for evaluation of atmospheric power when the atmosphere has a water cycle.

Finally, we note that Eq. (9) does not assume stationarity. In the next section we consider how W can be decomposed into several terms with different physical meaning. This will clarify how  $W = W_{III}$  relates to  $W_I$  (1) and  $W_{II}$  (2).

**Appendix A. Deriving W (9) from $W_p$ (10) for ideal gas**

The equation of state for ideal gas is

$$p = NRT. \tag{A1}$$

Here T is temperature, N is air molar density (mol m-3),  $V \equiv N^{-1}$  is the atmospheric volume occupied by one mole of air, p is air pressure and R = 8.3 J mol-1 K-1 is the universal gas constant.

Using (A1) we can now write  $W_p$  (10) as  $W_p = \frac{p}{\tilde{V}} \frac{d\tilde{V}}{dt} = \frac{p}{\tilde{V}} \left( \tilde{N} \frac{dV}{dt} + V \frac{d\tilde{N}}{dt} \right) = -\frac{dp}{dt} + RN \frac{dT}{dt} + \frac{RT}{\tilde{V}} \frac{d\tilde{N}}{dt}$ =  $RT \left( -\frac{dN}{dt} + \frac{1}{\tilde{V}} \frac{d\tilde{N}}{dt} \right)$ . Here  $\tilde{N}$  is the number of moles of gas within volume  $\tilde{V}, \tilde{V} = \tilde{N}V$ .

The number of molecules (moles)  $\tilde{N}$  in each air parcel can only change via an inflow (outflow) of molecules through the parcel's boundary. This change results from either diffusion of molecules between the adjacent parcels or from phase transitions or from both. Since in the case of diffusion any molecule leaving one parcel,  $d\tilde{N}_1/dt < 0$ , arrives to some other parcel,  $d\tilde{N}_2/dt = -d\tilde{N}_1/dt > 0$ , all the diffusion terms cancel in the global sum of the last term in Eq. (2) over all parcels. What remains corresponds to phase transitions:

$$\sum_{i=1}^{n} \frac{d\tilde{N}_{i}}{dt} = \int_{\mathcal{V}} \frac{1}{\tilde{V}} \frac{d\tilde{N}}{dt} d\mathcal{V} = \int_{\mathcal{V}} \dot{N} d\mathcal{V}, \tag{A2}$$

where  $\dot{N}$  is the molar rate of phase transitions per unit volume (mol m-3 s-1). Its integral over volume  $\mathcal{V}$  is equal to the total rate of phase transitions in all the *n* air parcels. By virtue of the conservation relationship (A2)  $\dot{N}$  includes the inflow (outflow) into all the air parcels from all liquid or solid surfaces (droplet surface in the atmospheric interior or the Earth's surface).

Using Eqs. (2) and (A2) we can write total power W of the n air parcels composing the atmosphere as

$$W \equiv \frac{1}{S} \int_{\mathcal{V}} W_p d\mathcal{V} = \frac{1}{S} \int_{\mathcal{V}} RT \left( \dot{N} - \frac{dN}{dt} \right) d\mathcal{V}.$$
 (A3)

Here  $dN/dt \equiv \partial N/\partial t + \mathbf{v} \cdot \nabla N$  is the material derivative of N with  $\mathbf{v}$  being the gas velocity.

On the other hand, the continuity equation can be written as

$$\dot{N} - \frac{dN}{dt} \equiv \dot{N} - \frac{\partial N}{\partial t} - \mathbf{v} \cdot \nabla N = N \nabla \cdot \mathbf{v}.$$
(A4)

Note that the left-hand side of the continuity equation is identical to the term in braces in Eq. (A3), the latter based on the equation of state (A1). Multiplying Eq. (A4) by RT and noting that p = NRT, we find that Eq. (A3) turns into Eq. (9).

The physical meaning of Eq. (A3) becomes clear from consideration of an atmosphere that is motionless on a large scale, such that  $\mathbf{v} = 0$  and  $\nabla \cdot \mathbf{v} = 0$ . Then condensation that occurs instantaneously on a smaller scale is described by the source term  $\dot{N} < 0$  that represents the large-scale mean. The compensatory expansion of the adjacent air is described by the large-scale mean  $\partial N/\partial t < 0$  showing that the molar concentration of air diminishes. As is clear from Eqs. (A3) and (A4), since  $\dot{N} - \partial N/\partial t = 0$ , no resulting work is performed on the considered scale: W = 0.

**Reply to the referees**

We thank our referees for their comments.

Comment 3 of Referee 1 [doi:10.5194/acp-2016-203-RC1]: 3. The derivation of the total atmospheric power given by Eq. (7)-(8) is unnecessarily complicated. I can directly obtain Eq. (7) from Eq. (2) by noting simply that  $\int pdV/dt = \int pd(\delta x \delta y \delta z)/dt =$

 $\int p(\nabla \cdot v)dV$ . Not sure why the authors present their argument in such a lengthy and confusing way. The referee also noted in his general comments that the presentation of this work is somewhat confusing, and can be simplified substantially to make it clearer.

We followed the referee's suggestion to directly derive W from the definition of relative volume change. As we discuss in the revised Section 2, this derivation contains an implicit assumption that any volume change occurs at the expense of the divergence of velocity  $\nabla \cdot \mathbf{v}$  defined at an arbitrary scale. Since phase transitions involve gas velocities that are scale-specific, the plausibility of this assumption for this case requires a discussion, which is presented in the revised text.

Comment 2 of Referee 2 [doi:10.5194/acp-2016-203-RC2]: 1. Section 2 is both way too complicated and appears to be wrong. Following Vallis' (2006) notation:

$$W = \int_{\mathcal{V}} p \frac{d\alpha}{dt} \rho d\mathcal{V} = \int_{\mathcal{V}} p(\partial_t(\rho\alpha) + \nabla \cdot (\rho\alpha \mathbf{v}) - \alpha S_\rho) d\mathcal{V},$$

where  $S_{\rho} = \partial_t(\rho) + \nabla \cdot (\rho \mathbf{v})$  is the local sources and sinks of mass. Now,  $\alpha \rho = 1$  so

$$W = \int_{\mathcal{V}} p(\nabla \cdot (\mathbf{v}) - \alpha S_{\rho}) d\mathcal{V} = \int_{\mathcal{V}} (\nabla \cdot (p\mathbf{v}) - \mathbf{v} \cdot \nabla p - \alpha p S_{\rho}) d\mathcal{V}.$$

This is the same form as in equation (8). But it depends explicitly on  $S_{\rho}$ , contrary to the authors' claim. Why this contradiction? The problem in the authors' derivation comes in part from equation (3). While it is true that  $\sum_i d\tilde{N}_i/dt = 0$ , it is not true that  $\sum_i T_i d\tilde{N}_i/dt = 0$ , unless the atmosphere is isothermal. But it is exactly what's used to convert the last term in equation (2) to the last term in equation (4).  $\tilde{V} = \tilde{N}/N$  has units of  $m^3$  (parcel-1). To compute work, however, we need the specific volume with units of  $m^3$  (kg-1). So we have to introduce a new quantity, the mass per parcel  $\tilde{m}$  so that the specific volume is  $\tilde{V}/\tilde{m}$ . Then the expression for work (equation 4) with the same units as in Vallis (2006) reads:

$$W = \frac{1}{S} \int_{\mathcal{V}} p \frac{\tilde{m}}{\tilde{V}} \frac{d(\tilde{V}/\tilde{m})}{dt} d\mathcal{V}.$$

But the continuity equation (6) also requires fixing. Since, *N* has units of mol  $m^{-3}$  then equation (6) is an equation for mass conservation only if the molar mass  $\tilde{m}/\tilde{N}$  is constant. But here the authors are, among other things, concerned about the effect of moisture on the work and moist air, unlike dry air, has an inhomogeneous in molar mass. The continuity equation (6) should then read:

$$\partial_t((\tilde{m}/\tilde{V})) + \nabla \cdot (\mathbf{v}(\tilde{m}/\tilde{V})) = \dot{\tilde{m}}(N/\tilde{N}) + \tilde{m}/\tilde{N}\dot{N} - (\tilde{m}N/\tilde{N}^2)\tilde{N}$$

where the right hand side is the local sources and sinks of mass. With these fixes, the expression for work will look exactly like in Vallis (2006) and will depend on the sources and sinks of mass.

The discrepancy between our Eq. (7) and the referee's derivation results from the incorrect definition of work. It is not  $pd\alpha$  as clarified in our revised text, see Eq. (11) above. Therefore,  $W \neq W_{IV} \equiv (1/S) \int_{\mathcal{V}} p(d\alpha/dt)\rho d\mathcal{V}$ .

We also note that in our derivation we did not assume either  $\sum_i d\tilde{N}_i/dt = 0$  or  $\sum_i RT d\tilde{N}_i/dt = 0$ . This misunderstanding might have arisen because the derivation was presented in a very compact form. The revised more detailed text (new Appendix A) makes it clear that the resulting expression for work does not depend on the temperature term discussed by the referee.

Comment 3 of Referee 4 [doi:10.5194/acp-2016-203-RC4]: Remarks on the methodology. Physically, the atmospheric energy budget is best understood by introducing some kind of available enthalpy  $ape = h(\eta, q_t, p) - h_r(\eta, q_t)$ , where h is the moist specific enthalpy,  $\eta$  is some suitable definition of moist specific entropy, and  $q_t$  the total specific humidity, p is pressure, where  $h_r(\eta, q_t)$  representing the part of the total enthalpy that is not available for adiabatic conversions into kinetic energy, so that

$$dh = (T - T_r)d\eta + (\mu - \mu_r)dq_t + \alpha dp$$

As a result, it is possible to express the total power term as

$$\int_{V} p \frac{D\alpha}{Dt} \rho dV = \underbrace{\int_{V} \frac{D(p\alpha)}{Dt} \rho dV}_{=0} - \int_{V} \alpha \frac{Dp}{Dt} \rho dV = \int_{V} \frac{T - T_{r}}{T} \dot{q} dm + \int_{V} (\mu - \mu_{r}) \frac{Dq_{t}}{Dt} dm$$

where  $\dot{q}$  represents diabatic heating terms by all manner of conduction of radiation. This neglects the integral of dh/dt, but this term could be retained if desired. The passage from the first term to the second term requires  $\nabla(\rho v) = 0$ , and  $\rho v$  to the total mass flux, in order to be able to claim that the integral of  $D(p\alpha)/Dt$  vanishes, so the authors should clarify this point, as well as boundary conditions assumed by the different velocities entering the definition of v. In any case, the above formalism is usually what constitutes the starting point for linking the atmospheric power budget to a Carnot-like theory and for constraining the atmospheric power budget to solar heating, sensible heat fluxes, and condensation/evaporation process. The approach proposed by the authors seem to be quite unrelated to this standard view.

The referee uses the same incorrect expression for work as Referee 2 above, with the same resulting discrepancies from our derivation. Total power is not equal to  $W_{IV} \equiv (1/S) \int_{\mathcal{V}} p(d\alpha/dt)\rho d\mathcal{V}$ , see Eq. (11) above. Moreover, since  $\nabla \cdot (\rho \mathbf{v}) = \dot{\rho} \neq 0$ , the second equation of the referee contradicts the first one.

**References**

- G. K. Batchelor. An Introduction to Fluid Dynamics. Cambridge University Press, 2000. doi: 10.1017/CBO9780511800955.
- O. Pauluis and I. M. Held. Entropy budget of an atmosphere in radiative–convective equilibrium. Part I: Maximum work and frictional dissipation. J. Atmos. Sci., 59:125–139, 2002. doi: 10.1175/1520-0469(2002)059<0125:EBOAAI>2.0.CO;2.

Joachim Pelkowski and Thomas Frisius. The theoretician's clouds-heavier or lighter than air?

On densities in atmospheric thermodynamics. J. Atmos. Sci., 68:2430–2437, 2011. doi: 10.1175/JAS-D-11-085.1.

G. K. Vallis. Atmospheric and Oceanic Fluid Dynamics: Fundamentals and Large-Scale Circulation. Cambridge University Press, 2006.

---

## Author Comment (AC4) · 11 Jul 2016

Here we reply to Comment 1 of Referee 3 who suggests that, contrary to our claims, our main results and specifically Eqs. 20-22 are not original. Presumably there is some misunderstanding involved so we have revised the text clarifying how our results relate to previous work. In particular, we now show that Eqs. 20-22 could not *in principle* be formulated by Pauluis et al. 2000, because their basic assumptions are not consistent with either Eqs. 20-22 or with Eq. 4 of Pauluis and Held 2002. We acknowledge the value in making this claim clear and explicit as it is precisely because Eqs. 20-22 were not published previously that the global gravitational power of precipitation  $W_P$  could also not be estimated from re-analyses until now. Eqs. 20-22 are distinct in showing that  $W_P$  can be estimated directly from air velocity and pressure gradient without any knowledge of the atmospheric moisture content or precipitation rates.

The revised Section 4 (now Section 3) of our manuscript can be found below1, where subsection 3.3 is devoted to comparing our results with those of Pauluis et al. This is followed by the referee's comment and our reply to it.

1In the revised manuscript this section follows the revised Section 2 published in our previous comment [doi:10.5194/acp-2016-203-AC3]. The relevant equations from the previous sections are as follows:

$$W_{I} \equiv -\frac{1}{S} \int_{\mathcal{M}} \mathbf{v} \cdot \nabla p \alpha d\mathcal{M} = -\frac{1}{S} \int_{\mathcal{V}} \mathbf{v} \cdot \nabla p d\mathcal{V}, \qquad (1)$$

$$W_{II} \equiv -\frac{1}{S} \int_{\mathcal{V}} \mathbf{u} \cdot \nabla p d\mathcal{V}, \qquad (2)$$

$$W_{III} \equiv \frac{1}{S} \int_{\mathcal{V}} p \nabla \cdot \mathbf{v} d\mathcal{V}, \tag{3}$$

$$W_{IV} \equiv \frac{1}{S} \int_{\mathcal{M}} p \frac{d\alpha}{dt} d\mathcal{M}, \tag{4}$$

$$\frac{dX}{dt} \equiv \frac{\partial X}{\partial t} + \mathbf{v} \cdot \nabla X,\tag{5}$$

$$\frac{\partial \rho}{\partial t} + \nabla \cdot (\rho \mathbf{v}) = \dot{\rho},\tag{6}$$

$$V_p \equiv \frac{p}{\tilde{V}} \frac{dV}{dt},\tag{7}$$

$$W \equiv \frac{1}{\mathcal{S}} \sum_{i=1}^{n} W_{pi} \tilde{V}_{i} = \frac{1}{\mathcal{S}} \int_{\mathcal{V}} W_{p} d\mathcal{V} = \frac{1}{\mathcal{S}} \int_{\mathcal{V}} p(\nabla \cdot \mathbf{v}) d\mathcal{V}.$$
(9)

V

**3 Revisiting the current understanding of the atmospheric power budget**

**3.1 The boundary condition for vertical velocity at the surface**

Noting that  $p(\nabla \cdot \mathbf{v}) = \nabla \cdot (p\mathbf{v}) - \mathbf{v} \cdot \nabla p$  and using the divergence theorem (Gauss-Ostrogradsky theorem) we can see that  $W = W_{III}$  (9) coincides with  $W_I$  (1),

$$W = W_{III} = \frac{1}{\mathcal{S}} \int_{\mathcal{V}} p(\nabla \cdot \mathbf{v}) d\mathcal{V} = -\frac{1}{\mathcal{S}} \int_{\mathcal{V}} (\mathbf{v} \cdot \nabla p) d\mathcal{V} + I_T + I_s \equiv W_I + I_T + I_s, \quad (12)$$

if the following integrals are zero:

$$I_T \equiv \frac{p_T}{S} \int_{z=z(p_T)} (\mathbf{v} \cdot \mathbf{n}) dS = 0, \qquad (13)$$

$$I_s \equiv \frac{1}{S} \int_{S} p_s(\mathbf{v} \cdot \mathbf{n}) dS = 0.$$
(14)

Integral (13) is taken over the upper boundary  $z = z(p_T)$ , where  $z(p_T)$  is the altitude of the pressure level  $p = p_T$  defining the top of the atmosphere. Since the distribution of pressure versus altitude is exponential and  $I_T$  is proportional to  $p_T$ , by choosing a sufficiently small  $p_T$  it is possible to ensure that  $I_T$  (13) is arbitrarily small compared to W. For  $p_T = 0.1$  hPa we estimate  $I_T \sim 10^{-4}W$  (see Fig. 6d in Appendix C). So it is safe to assume that  $I_T = 0$ .

Integral (14) is taken over the Earth's surface ( $p_s$  is surface pressure). In a dry atmosphere we have

$$\mathbf{v} \cdot \mathbf{n}|_{z=0} = w_s = 0. \tag{15}$$

Here  $w_s$  is the surface value of the vertical velocity of air w. As we discuss below, for a moist atmosphere with surface evaporation Eq. (15) also holds, such that  $W = W_{III} = W_I$ , see Eqs. (9), (3), (1), (12).

СЗ

Evaporation from the Earth's surface represents a flux of water vapor molecules. As far as the relative humidity at the surface is always less than or equal to unity, the local value of this flux at z = 0 is never negative. The evaporating water molecules may have a mean vertical velocity  $w_E > 0$  of the order of sound velocity only until they collide with the other air molecules at a distance of about one free path length  $l_f \sim 10^{-7}$  m from the surface. Molecular collisions ensure that the mean velocity of evaporated molecules rapidly approaches the parcel's mean due to molecular collisions. Molar density  $N_E$  of evaporating molecules is obtained from evaporation rate E (kg m-2 s-1) as  $N_E w_E = E/M_v$ , where  $M_v$  is molar mass of water. If  $l_p$  is the linear size of this parcel, we have for the mean vertical velocity of all molecules in the parcel  $w_s = w_E N_E l_f / (N_s l_p)$ , where  $N_s = p_s / (RT_s)$  is molar density of air at the surface,  $T_s$  is surface temperature. Then in Eq. (14) we have  $p_s w_s = (l_f / l_p) ERT_s / M_v$ .

Since, as we will see in Sections 5 and 6, global atmospheric power is of the order of  $PRT_s/M_v$ , where P = E is global mean precipitation and evaporation, the surface term  $p_sw_s = (l_f/l_p)ERT_s/M_v$  can be neglected in Eq. (12) if  $l_f/l_p \ll 1$ , i.e. on any macroscopic scale. This reflects the fact that the atmosphere *does not* circulate because of being "pushed upwards" by surface evaporation. Notably, Eq. (15) is not in contradiction with the existence of an inflow of water vapor into the atmosphere at z = 0. It just means that water vapor should be considered as arising by evaporation within the surface air parcels, the latter having zero vertical velocity at their lower boundary.

We emphasize that the boundary condition (15) is vital for the equality between  $W = W_{III}$  (9), (3) derived from the thermodynamic definition of work and  $W_I$  (1) of Lorenz (1967). In the last decades  $W_I$  was used by various researchers to evaluate the atmospheric energy cycle. For example, Laliberté et al. (2015) used  $W_I$  as their thermodynamic definition of atmospheric work output thus assuming that Eq. (15) holds. If one ignored Eq. (15) and defined  $w_s > 0$  for z = 0 from the upward flux of water vapor as  $\rho_{vs}w_s = E$ , where  $\rho_{vs}$  is water vapor density at the surface (see, e.g., Eq. 3 of Pauluis et al., 2000, to be discussed below), one would have obtained unphysical results. With  $E \sim 10^3$  kg m-2 year-1 and  $\rho_{vs} \sim 10^{-2}$  kg m-3 we would have  $w_s \sim 3 \times 10^{-3}$  m s-1 and  $I_s = p_s w_s = 3 \times 10^2$  W m-2. Then from Eq. (12) we would conclude that total power of atmospheric circulation  $W = W_I + I_s > I_s$  exceeds the incoming flux of solar radiation.

**3.2 Kinetic power and the gravitational power of precipitation**

We now show how total atmospheric power (12) is comprised of three distinct terms. In hydrostatic equilibrium we have

$$\nabla_z p = \rho \mathbf{g}.\tag{16}$$

In the real atmosphere due to the presence of non-gaseous water the distribution of air deviates from Eq. (16) such that we instead have

$$\nabla_z p = (\rho + \rho_c) \mathbf{g}.\tag{17}$$

Using (17) we have in (12)  $-\mathbf{v} \cdot \nabla p = -\mathbf{u} \cdot \nabla p - \mathbf{w} \cdot \nabla p = -\mathbf{u} \cdot \nabla p - (\rho + \rho_c) \mathbf{w} \cdot \mathbf{g}$ . Here  $-\rho \mathbf{w} \cdot \mathbf{g}$  represents the vertical flux of air: it is positive (negative) for the ascending (descending) air. Recalling that

$$\mathbf{g} = -g\nabla z \tag{18}$$

and using the divergence theorem and the stationary continuity equation (6) we have

$$W_P \equiv -\frac{1}{\mathcal{S}} \int_{\mathcal{V}} \rho(\mathbf{w} \cdot \mathbf{g}) d\mathcal{V} = \frac{g}{\mathcal{S}} \int_{\mathcal{V}} \rho(\mathbf{v} \cdot \nabla z) d\mathcal{V} = \frac{g}{\mathcal{S}} \int_{\mathcal{S}} \mathbf{n} \cdot (\mathbf{v}\rho z) d\mathcal{S} - \frac{1}{\mathcal{S}} \int_{\mathcal{V}} gz\dot{\rho}d\mathcal{V}.$$
 (19)

The surface integral in (19) is taken at the Earth's surface (here it is zero because z = 0) and  $z = z(p_T)$  (here it is also zero, because  $\rho \mathbf{n} \cdot \mathbf{v} = 0$ ).

It is natural to call  $W_P$  the "gravitational power of precipitation". Indeed, in the last integral in (19) gz represents potential energy of a unit mass in the Earth's gravitational C5

field. Therefore, when evaporation occurs at the surface z = 0 and the condensate falls from height z where it originated, it is clear from Eq. (19) that  $W_P$  (19) is equal to  $Pg\mathcal{H}_P$ , where  $\mathcal{H}_P$  is the global mean height of condensation.

The stationary power budget for a hydrostatic atmosphere can be written as

$$W = -\frac{1}{S} \int_{\mathcal{V}} \mathbf{v} \cdot \nabla p d\mathcal{V} \equiv W_K + W_P, \qquad (20)$$

$$W_{K} \equiv -\frac{1}{\mathcal{S}} \int_{\mathcal{V}} (\mathbf{u} \cdot \nabla p) d\mathcal{V} + W_{c} \approx -\frac{1}{\mathcal{S}} \int_{\mathcal{V}} \mathbf{u} \cdot \nabla p d\mathcal{V}, \quad W_{c} \equiv -\frac{1}{\mathcal{S}} \int_{\mathcal{V}} \rho_{c} (\mathbf{w} \cdot \mathbf{g}) d\mathcal{V} (21)$$

$$W_P \equiv -\frac{1}{\mathcal{S}} \int_{\mathcal{V}} \rho \mathbf{w} \cdot \mathbf{g} d\mathcal{V} = -\frac{1}{\mathcal{S}} \int_{\mathcal{V}} g z \dot{\rho} d\mathcal{V} = P g \mathcal{H}_P, \quad P \equiv -\frac{1}{\mathcal{S}} \int_{z>0} \dot{\rho} d\mathcal{V}.$$
(22)

Equations (20)-(22) and their derivation have not been previously published (see the next section). These equations clarify the physical meaning of the atmospheric power budget.

Term  $W_c$  in Eq. (21) describes the impact of condensate loading. It represents kinetic energy generation on the vertical scale of the order of the atmospheric scale height  $\mathcal{H} \equiv -p/(\partial p/\partial z) = RT/(Mg) \sim 10$  km. This energy is generated because the vertical air distribution deviates from the hydrostatic equilibrium (16). Hydrometeors falling at terminal velocity exert a force on the air equal to their weight. The condensate thus acts as resistance preventing the pressure difference  $\Delta p \sim \rho_{cg} \mathcal{H}$  from converting to the kinetic energy of a vertical wind. In the atmosphere on average  $\rho_c/\rho \sim 10^{-5}$  (Makarieva et al., 2013). Without hydrometeors, the non-equilibrium pressure difference  $\Delta p \sim$  $10^{-5}\rho q \mathcal{H} \sim 1$  Pa would produce maximum vertical velocity of about  $w_m \sim 1$  m s-1  $(\rho w_m^2/2 = \Delta p)$ . This is two orders of magnitude larger than the characteristic vertical velocities  $w \sim 10^{-2}$  m s-1 of large-scale air motions. (Hydrometeors thus inhibit vertical motion in a similar way to how turbulent friction at the surface inhibits horizontal air motion. For example, the observed meridional surface pressure differences of the order of  $\Delta p_h \sim 10$  hPa in the tropics, if friction were absent, could have produced maximum horizontal air velocities of about  $u_m \sim 40 \text{ m s}^{-1}$  ( $\rho u_m^2/2 = \Delta p_h$ ).) Quantitatively,  $-\rho_c \mathbf{w} \cdot \mathbf{g}$  is less than 1% of W and can be neglected: its volume integral taken per unit surface area is less than  $\rho_c g \mathcal{H} w \sim 10^{-5} pw \sim 10^{-2} \text{ W m}^{-2}$ , where  $p = \rho g \mathcal{H} = 10^5$  Pa is air pressure at the surface.

In contrast, the gravitational power of precipitation  $W_P$  does not depend on aircondensate interactions. (For example, this term would be present in the atmospheric power budget even if the condensate disappeared immediately upon condensation or experienced free fall not interacting with the air at all.) This is because  $W_P$  (19) reflects the net work of water vapor as it travels from the level where evaporation occurs (where water vapor arises) to the level where condensation occurs (where water vapor disappears). When condensation occurs above where evaporation occurs, the water vapor expands as it moves upwards towards condensation, and the work is positive irrespective of what happens to the condensate. For a dry atmosphere where  $\dot{\rho} = 0$ , the last volume integral in Eq. (19) is zero and  $W_P = 0$ : indeed, in this case at any height *z* there is as much air going upwards as there is going downwards.

Next, Eqs. (20)-(22) clarify the relationship between the two formulations of atmospheric power  $W_I$  (1) and  $W_{II}$  (2):  $W_{II} \approx W_K$  coincides with  $W = W_I$  in the absence of phase transitions only, i.e. when  $W_P = 0$ . This resolves some confusion in the literature, whereby in some publications it is total atmospheric power  $W = W_I$  that is referred to as *generation of kinetic energy* (e.g., Robertson et al., 2011, their Eq. 1), while in others the same term is applied to  $W_K$ , which is estimated from horizontal velocities (see, e.g., Boville and Bretherton, 2003; Huang and McElroy, 2015). At the same time, in such studies  $W_K$  is confused with the total atmospheric power W: i.e. in the total power budget the gravitational power of precipitation,  $W_P$ , is overlooked (e.g., Huang and McElroy, 2015, their Fig. 10). We also note that the gravitational power of precipitation  $W_P$  has not been explicitly identified in past studies assessing the Lorenz energy cycle (see, e.g., Kim and Kim, 2013, and references therein).

Finally, Eqs. (20)-(22) show that the gravitational power of precipitation can be esti-

**C7**

mated from air velocity and pressure gradient alone as  $W_P = W - W_K$  without any knowledge of the atmospheric moisture content or precipitation rates. This allows global  $W_P$  to be estimated from re-analyses data, see Section 6. So far, the only global estimate of  $W_P$  was that of Makarieva et al. (2013) based on  $W_P = Pg\mathcal{H}_P$ . Pauluis et al. (2000) used  $Pg\mathcal{H}_P$  to estimate precipitation-related dissipation in the tropics.

**3.3 Our results compared to Pauluis et al. 2000**

Our assessment of the atmospheric power budget started from the thermodynamic definition of work (7). Integrated over atmospheric volume Eq. (7) yielded total atmospheric power  $W = W_{III}$  (9), (3). The boundary condition (15) turned  $W_{III}$  into the commonly used  $W_I$  (12). Then we used the continuity equation (6) and hydrostatic equilibrium (17) to separate the kinetic energy generation  $W_K$  from the gravitational power of precipitation  $W_P$ ,  $W = W_K + W_P$ , in Eqs. (20)-(22).

Here we compare our results with those of Pauluis et al. (2000) who likewise identified two distinct terms in the atmospheric power budget. Pauluis et al. (2000) were presumably aware of the fact that total atmospheric power is equal to  $W_{III}$ , since Eq. (3) was listed by Pauluis and Held (2002). However, as we show below, several inconsistencies in their basic assumptions did not permit obtaining results equivalent to Eqs. (20)-(22).

Noting that condensate is falling at terminal velocity  $v_T$  experiencing resistance force  $\rho_{cg}$ , Pauluis et al. (2000) defined the precipitation-related frictional dissipation as follows (we added factor 1/S to enable comparison with our results):

$$W_P^* \equiv \frac{1}{S} \int_{\mathcal{V}} \rho_c g v_T d\mathcal{V}.$$
 (23)

Here  $v_T \equiv w - w_c$  is the difference between the vertical velocities air and condensate. Assuming that *at any level*  $z = z_0$  in the atmosphere the upward flux of water wapor is balanced by the downward flux of condensate,

$$\int_{z=z_0} \rho_c w_c d\mathcal{S} + \int_{z=z_0} \rho_v w d\mathcal{S} = 0, \qquad (24)$$

Pauluis et al. (2000, see their Eq. 3) obtained2

$$W_P^* = \frac{1}{S} \int_{\mathcal{V}} (\rho_c + \rho_v) wg d\mathcal{V} \equiv W_P + W_c, \qquad (25)$$

where  $W_P$  and  $W_c$  are defined in Eqs. (22) and (21). Thus,  $W_P^*$  lumps together two terms with distinct meaning, with  $W_c$  depending on the interaction between the condensate and air and  $W_P$  independent of it.

Rather than using Eq. (18) and the continuity equation for water vapor similar to what was done in Eq. (19), Pauluis et al. (2000) further assumed that  $W_P^*$  is proportional to the precipitation rate P at the surface, which is given by the surface integral  $(1/S) \int_{z=0} \rho_c v_T dS$ . This formulation of surface precipitation via  $v_T$  yielded

$$W_P^* = Pg\mathcal{H}_P \tag{26}$$

for the case when no re-evaporation occurs in the atmosphere. In reality, however,  $P = -(1/S) \int_{z=0} \rho_c w_c dS$ , so Eq. (26) is not consistent with Eq. (25).

The two integrals coincide,  $-\int_{z=0} \rho_c w_c dS = \int_{z=0} \rho_c v_T dS$ , only if  $w|_{z=0} \equiv w_s = 0$ . But this is inconsistent with the key assumption (24), since for  $w_s = 0$  and  $w_{cs} \neq 0$  Eq. (24) does not hold for z = 0. Indeed, for z = 0 Eq. (24) contradicts the boundary condition (15)  $w_s = 0$ . In particular, for the case when local evaporation equals local precipitation, Eq. (24) gives  $w_s = -\rho_{cs}w_{cs}/\rho_{vs} > 0$ .

This inconsistency between Eq. (24) for z = 0 and the equality  $W_I = W_{III}$ , see Eq. (12), precludes a straightforward derivation of W (9) from  $W_{III}$  (3) of Pauluis and Held (2002). Instead, Pauluis et al. (2000) assumed that *total mechanical work by resolved eddies*  $W_{tot}$  is equal to the sum of the *frictional dissipation associated with convective and boundary-layer turbulence*  $W_D$  and the *total dissipation rate due to precipitation*  $W_P^*$ :

$$W_{tot} = W_D + W_P^*. \tag{27}$$

Since no general specification for turbulent processes exists, this formulation *per se*, unlike Eqs. (20)-(22), cannot guide an assessment of  $W_{tot}$  from observations. However,  $W_D$  can be retrieved from the equation of motion as the volume integral of  $-\mathbf{F} \cdot \mathbf{v}$ , where **F** is turbulent friction force (cf. Lorenz, 1967, Eq. 101).

For the moist air (gas) moving under the action of a pressure gradient force, gravity, condensate loading and turbulent friction force, the scalar product of the equation of motion with velocity v reads (see, e.g., Huang and McElroy, 2015, for more details):

$$\rho \frac{dK}{dt} = -\mathbf{v} \cdot \nabla p + \rho \mathbf{w} \cdot \mathbf{g} + \rho \mathbf{w} \cdot \mathbf{g} + \mathbf{F} \cdot \mathbf{v}.$$
(28)

Here  $K \equiv v^2/2$  is the kinetic energy of air per unit air mass. By virtue of relationship (17) the sum of the first three terms in the right-hand side of (28) is equal to  $-\mathbf{u} \cdot \nabla p$ . So, integrating (28) over volume, using the divergence theorem, continuity equation (6), boundary condition (19) and Eq. (21), we obtain

$$W_D \equiv -\frac{1}{\mathcal{S}} \int_{\mathcal{V}} (\mathbf{F} \cdot \mathbf{v}) d\mathcal{V} = W_K - W_c - \overline{K}, \quad \overline{K} \equiv \frac{1}{\mathcal{S}} \int_{\mathcal{V}} K \dot{\rho} d\mathcal{V}.$$
(29)

Now, if we assume that  $W_{tot} = W_I$  as in Eq. (20) and use  $W_D$  (29), the correct formulation for the atmospheric power budget in terms of Pauluis et al. (2000) would be

$$W_{tot} = W_D + W_P^* + \frac{\mathbf{\vec{k}}}{\mathbf{\vec{k}}}, \quad W_P^* = Pg\mathcal{H}_P + \underline{W_c}.$$

(30)

<sup>2Using the continuity equations for dry air  $\nabla \cdot (\rho_d \mathbf{v}) = 0$  and water vapor  $\nabla \cdot (\rho_v \mathbf{v}) = \dot{\rho}$ , where  $\rho_d$  and  $\rho_v$  are densities of dry air and water vapor, we find from Eq. (19) that  $W_P = -(1/S) \int_{\mathcal{V}} \rho_v \mathbf{w} \cdot \mathbf{g} d\mathcal{V}$ .

The underlined terms differentiate the correct equations (30) from the formulations (26) and (27) of Pauluis et al. (2000). Term  $\vec{K}$  describes the sink of the kinetic energy of air caused by condensation. For example, if condensation occurs in the middle troposphere where air velocity is of the order of 25 m s-1, the magnitude of this term estimates as  $Pv^2/2 \sim 10^{-2}$  W m-2. This is of the same order as the condensate loading term  $W_c$ , which was retained by Pauluis et al. (2000) in the definition of  $W_P^*$  (25).

If the condensate does not interact with the air but experiences free fall, then, despite  $\rho_c \neq 0$ , the condensate loading (i.e. terms proportional to  $\rho_c$ ) is absent from the equation of motion as well as from Eqs. (17), (21) and (28). However, since  $W_c$  is present in the definition of  $W_P^*$  (23), in this case the correct equation for  $W_{tot}$  becomes  $W_{tot} = W_D + W_P^* - \underline{W_c} + \underline{K}$ , i.e. Eqs. (23) and (27) additionally overestimate the actual  $W_{tot}$  by the term equal to condensate loading. Thus, the formulations of Pauluis et al. (2000) are generally consistent with Eqs. (20)-(22) if only the condensate loading term and  $\overline{K}$  are neglected. Nevertheless, even in this case there remains a discrepancy between  $W_{tot} = W_I$  of Pauluis et al. (2000) and  $W_{tot} = W_{III}$  (3) of Pauluis and Held (2002), both representing the total atmospheric power, see Eq. (12). Since Eq. (24) of Pauluis et al. (2000) assumes  $w_s > 0$  and  $I_s > 0$ , it follows from the divergence theorem that  $W_{tot} = W_I + I_s \neq W_{III}$ . In other words, Pauluis et al. (2000) could not demonstrate that their  $W_{tot}$  defined by Eq. (27) is equivalent to the thermodynamic definition of work  $W_{III}$  (3) (see discussion in the end of Section 3.1). The derivation of Eqs. (20)-(22) is free from this contradiction.

**Reply to the Referee**

Comment 1 of Referee 3 [doi:10.5194/acp-2016-203-RC3]:

**1. Appropriation in the main result:**

The manuscript states pretty explicitly that the main contribution here is

"Starting from the definition of mechanical work for an ideal gas, we present a novel derivation linking global wind power to measurable atmospheric parameters. The resulting expression distinguishes three components: the kinetic power associated with horizontal motion, the kinetic power associated with vertical motion and the gravitational power of precipitation."

as it is stated in the abstract. This claim is repeated on multiple occasions. I assume that this specifically refer to the equation (20-22), which the authors claim that "Equations Eqs. (20)-(22) and their derivation have not been previously published."

These equations are presented in Pauluis etal. (2000) (See equations (2), (4), (8) and (10). See also equations (4) and equation (6) of Pauluis and Held (JAS, 2002)). It is very troublesome that the authors fail to mention that equations (20-22) are presented in Pauluis etal. (2000) despite the fact that this pa

[the referee's comment continues below after our reply]

We revised the text having added a separate "Section 3.3 Our results compared to Pauluis et al. 2000". Right below Eqs. (20)-(22) we explain why in our view these equations are original. Furthermore, we also explicitly refer the readers to Section 3.3 where these results are compared with Pauluis et al. 2000 by noting: "Equations (20)-(22) and their derivation have not been previously published (see the next section)." Readers can judge our claims for themselves.

As a separate point, we note that Eqs. (20)-(22) make it clear that  $W_P$  can be estimated from the data on air velocity and pressure gradient with no information required about moist processes. As can easily be verified by examining the texts in question, this message is absent from the works cited by the referee (or indeed in any previous publications of which we are aware). To facilitate this comparison we list the equations mentioned by the referee below together with our Eqs. 20-22 from the submitted manuscript.

Pauluis et al (2000), Eqs. (2), (4), (8) and (10), respectively:

$$W_p = \int_{\Omega} g\rho_c v_{\mathsf{T}}, \qquad (c1)$$

$$W_p = \int_{\Omega} g\rho_t w, \qquad (c2)$$

$$W_D = \int \overline{\rho} gw \left[ \frac{\Theta'}{\overline{\Theta}} + \left( \frac{R_v}{R_d} - 1 \right) \frac{\rho_v}{\overline{\rho}} - \frac{\rho_c}{\overline{\rho}} \right], \tag{c3}$$

$$V_{\text{tot}} = \int wg \left[ \overline{\rho} \frac{\Theta'}{\overline{\Theta}} + \rho_v \frac{R_v}{R_d} \right], \qquad (c4)$$

where  $\rho_t = \rho_c + \rho_v$ .

Pauluis and Held (2002), Eqs. (4) and (6), respectively:

t

$$W = \int_{\Omega} p \partial_i V_i, \tag{c5}$$

$$D_p = \int_{\Omega} g\rho_c V_T = \int_{\Omega} \rho q_t gw, \qquad (c6)$$

where  $V_i$  is the *i*th component of the velocity,  $\partial_i = \partial/\partial x_i$  is the partial derivative in the *i* direction,  $\rho_c$  is the mass of falling hydrometeors per unit volume,  $q_t$  is mass of total water per unit mass of moist air,  $V_T$  is the terminal velocity of the falling hydrometeors, and *w* is the vertical velocity of the air.

Equations (20)-(22):

$$W = -\frac{1}{S} \int_{\mathcal{V}} \mathbf{v} \cdot \nabla p d\mathcal{V} \equiv W_K + W_P,$$
(c7)

$$\begin{split} W_{K} &\equiv -\frac{1}{\mathcal{S}} \int_{\mathcal{V}} (\mathbf{u} \cdot \nabla p) d\mathcal{V} + W_{c} \approx -\frac{1}{\mathcal{S}} \int_{\mathcal{V}} \mathbf{u} \cdot \nabla p d\mathcal{V}, \quad W_{c} \equiv -\frac{1}{\mathcal{S}} \int_{\mathcal{V}} \rho_{c}(\mathbf{w} \cdot \mathbf{g}) d\mathcal{V}, \\ W_{P} &\equiv -\frac{1}{\mathcal{S}} \int_{\mathcal{V}} \rho \mathbf{w} \cdot \mathbf{g} d\mathcal{V} = -\frac{1}{\mathcal{S}} \int_{\mathcal{V}} g z \dot{\rho} d\mathcal{V} = P g \mathcal{H}_{P}, \quad P \equiv -\frac{1}{\mathcal{S}} \int_{z>0} \dot{\rho} d\mathcal{V}. \end{split}$$
(c9)

Note that  $\rho = \rho_d + \rho_v \neq \rho q_t$ ;  $\mathbf{v} = \mathbf{u} + \mathbf{w}$  is air velocity (horizontal and vertical).

The referee continues:

The appropriation is not limited to the equations, but extends to some of the arguments presented. For instance, the authors relate the claim

"The meaning is that hydrometeors perform work at the expense of their potential energy. To acquire this energy, a corresponding amount of water vapor must be raised by air parcels. We can also see that WP does not depend on the interaction between the air and the falling hydrometeors. This term would be present in the atmospheric power budget even if hydrometeors were experiencing free fall and did not interact with the air at all (such that no frictional dissipation on hydrometeors occurred)."

**This points is made previously ( and more clearly) in Pauluis etal. JAS (2000, p. 991):**

"The dissipation by precipitation can be thought as proceeding in two steps. First, water is lifted by the atmospheric circulation, increasing its potential energy. Then, during precipitation, the potential energy of condensed water is transferred to the ambient air where it is dissipated by molecular viscosity in the microscopic shear zone around the hydrometeors."

To put it bluntly, the authors are presenting as their own an analysis that was done by others, and in doing so, are misleading their reader.

We show in the revised text (see the last paragraph in Section 3.3) that the above statement of Pauluis et al. (2000, p. 991) is not consistent with their own analysis, because their definition of precipitation-related dissipation hinges on the interaction between the condensate and the air. This definition, besides the gravitational power of precipitation to which the above comment correctly refers, also includes the condensate loading term  $W_c$  which has a different meaning. As demonstrated by our Eqs. 20-22 and the text below them, this term (but not  $W_P$ ) does depend on the interaction between the condensate and the air3.

**References**

- B. A. Boville and C. S. Bretherton. Heating and kinetic energy dissipation in the NCAR Community Atmosphere Model. *J. Climate*, 16:3877–3887, 2003. doi: 10.1175/1520-0442(2003) 016<3877:HAKEDI>2.0.CO;2.
- V. G. Gorshkov. Energetics of the biosphere. Leningrad Politechnical Institute, 1982.
- V. G. Gorshkov. *Physical and biological bases of life stability. Man, Biota, Environment.* Springer, 1995.
- J. Huang and M. B. McElroy. A 32-year perspective on the origin of wind energy in a warming climate. *Renewable Energy*, 77:482–492, 2015. doi: 10.1016/j.renene.2014.12.045.
- Y.-H. Kim and M.-K. Kim. Examination of the global lorenz energy cycle using MERRA and NCEP-reanalysis 2. *Climate Dynamics*, 40:1499–1513, 2013. doi: 10.1007/ s00382-012-1358-4.
- F. Laliberté, J. Zika, L. Mudryk, P. J. Kushner, J. Kjellsson, and K. Döös. Constrained work output of the moist atmospheric heat engine in a warming climate. *Science*, 347:540–543, 2015. doi: 10.1126/science.1257103.

E. N. Lorenz. *The Nature and Theory of the General Circulation of the Atmosphere*. World Meteorological Organization, 1967.

M. I. L'vovitch. World water resources and their future. American Geophysical Union, 1979.

A. M. Makarieva, V. G. Gorshkov, A. V. Nefiodov, D. Sheil, A. D. Nobre, P. Bunyard, and B.-L. Li.

The key physical parameters governing frictional dissipation in a precipitating atmosphere. *J. Atmos. Sci.*, 70:2916–2929, 2013. doi: 10.1175/JAS-D-12-0231.1.

- O. Pauluis and I. M. Held. Entropy budget of an atmosphere in radiative-convective equilibrium. Part I: Maximum work and frictional dissipation. *J. Atmos. Sci.*, 59:125–139, 2002. doi: 10.1175/1520-0469(2002)059<0125:EBOAAI>2.0.CO;2.
- O. Pauluis, V. Balaji, and I. M. Held. Frictional dissipation in a precipitating atmosphere. *J. Atmos. Sci.*, 57:989–994, 2000. doi: 10.1175/1520-0469(2000)057<0989:FDIAPA>2.0.CO;2.
- Franklin R. Robertson, Michael G. Bosilovich, Junye Chen, and Timothy L. Miller. The effect of satellite observing system changes on MERRA water and energy fluxes. *J. Climate*, 24: 5197–5217, 2011. doi: 10.1175/2011JCLI4227.1.

<sup>3For the record, the first estimate of the gravitational power of precipitation of which we are aware is Gorshkov (1982, p. 6). Considering possible sources of renewable energy to be harnessed on land, Gorshkov (1982) estimated the gravitational power of terrestrial precipitation by analogy with hydropower, for which the formulation  $Pg\mathcal{H}_P$  is commonly used. He used global mean precipitation on land of 0.5 m year-1 as given by L'vovitch (1979) and assumed that it rains on average from somewhere in the middle of the troposphere to obtain 1014 W (for an English citation, see Gorshkov (1995, p. 30)).

---

## Author Comment (AC5) · 13 Jul 2016

Here we reply to Comments 4, 2, 2 and 4 of, respectively, Referees 1, 2, 3 and 4 that pertain to our analysis of Laliberte et al. 2015. Specifically, Referee 3 suggested that Laliberte et al. assumed no sources and sinks in the continuity equation. We thus show that this suggestion contradicts what Laliberte et al. themselves state.

Referee 3 suggests we should omit our discussion of Laliberte et al. But given that Laliberte et al. analyzed the same database and reached different conclusions – the discrepancy requires clarification. We found that Laliberte et al. omitted a crucial term, the global integral of the material derivative of enthalpy. Our analysis clarifies the scale of this omission and can, we hope, reduce future confusion. Thus we believe this contribution is constructive and is essential to placing our work in context.

Below the referees' comments are listed each followed by our reply. Then there is revised and significantly shortened Section 3 (now Section 4), which contains our analysis of Laliberté et al. (2015). We correct two misprints in our previous comment [doi:10.5194/acp-2016-203-AC4, p. C10]: the third term in the right-hand side of Eq. (28) is  $\rho_c \mathbf{w} \cdot \mathbf{g}$  and the definition of  $\vec{K}$  in Eq. (29) misses the minus sign  $(\vec{K} \equiv -(1/S) \int_V K \dot{\rho} dV)$ .

Comment 4 of Referee 1 [doi:10.5194/acp-2016-203-RC1]: 4. The authors criticize Laliberte et al. (2015)'s estimation of the integral of dh/dt, as they believe that it is not dh/dt = 0 but should be  $\partial h/\partial t = 0$  for a stationary budget. However, my understanding of Laliberte et al.'s study is that the total derivative that Laliberte et al used is in the context of global integration. So if you define  $H = \int h dV$ , then  $dH/dt = \int \partial h/\partial t dV$ , since the total volume is fixed in time. As such, Laliberte et al.'s global stationary approximation.

Laliberté et al. (2015) aim to estimate the global mean value of atmospheric power  $-(1/\alpha)(dp/dt)$ . They cannot therefore follow the above described procedure integrating the first law of thermodynamics first over mass  $\mathcal{M}$ ,  $d\mathcal{M} = \rho d\mathcal{V}$ , and then taking its derivative over time. This procedure for  $-(1/\alpha)(dp/dt)$  would yield  $-\int_{\mathcal{V}} \partial p/\partial t d\mathcal{V} = 0$ .

Indeed, Laliberté et al. (2015) explicitly define dh/dt as the material derivative of enthalpy [see p. 540, middle column, 7th line from top], not the partial derivative over time. They state that they average the first law of thermodynamics taking the massweighted annual and spatial mean of all the terms in the equation, including dh/dt [p. 540, middle column, 7th line from bottom]. They denoted this mean as  $\{\cdot\}$ . The massweighted spatial mean of the material derivative of h, which is enthalpy per unit wet air mass, consists in taking its integral over total atmospheric mass and then dividing by the planet surface area. This means that stating that  $\{dh/dt\} = 0$  Laliberté et al. (2015) meant  $I_h \equiv (1/S) \int_{\mathcal{M}} dh/dt d\mathcal{M} = 0$  and not  $\partial (\int_{\mathcal{M}} h d\mathcal{M})/\partial t = 0$ .

We also note that, to support their statement that the expression for total atmospheric

power does not contain the enthalpy term, Laliberté et al. (2015) refer to Eq. 4 of Pauluis (2011) [p. 540, right column, 12th line from top]. This link does not recognize that Eq. 4 of Pauluis (2011) [ref. 10 of Laliberté et al. (2015)] refers to atmospheric power defined *per unit dry air mass*. As we note in the revised text, the material derivative of any variable integrated over *total mass of atmospheric dry air* is zero (because of zero sources or sinks of dry air). In contrast, the material derivative of any variable integrated over *total atmospheric mass* is in the general case not zero, because of the non-zero sources and sinks in the continuity equation. This point, which follows from the previous derivations in the paper, is essential for understanding the atmospheric power budget and also for estimating it.

Comment 2 of Referee 2 [doi:10.5194/acp-2016-203-RC2]:

2. Section 3.1. This section is also way too complicated. After the first paragraph, one can jump directly to the top of page 5. Now equation (15) is not wrong per se. ...

Following the referee's suggestion, we revised the text to immediately obtain equation (15) for  $I_h$  (currently Eq. 32) after the first paragraph. The remaining part of the referee's comment was addressed separately in our second Author Comment [doi:10.5194/acp-2016-203-AC2].

Comment 2 of Referee 3 [doi:10.5194/acp-2016-203-RC3] (note it comes in several parts):

2. Discussion of Laliberte etal. (2015)

The discussion of Laliberte etal. (2015) is very esoteric and does not pertain much to the rest of the discussion. Section 3.2 is a very minor point. It is fairly well-known that the integral of dp/dt is only equal to the work performed for a steady system, an assumption that is clearly stated in Laliberte etal.

Section 3.2 did not mention Laliberte et al. and did not question their steady state assumption. This section drew attention to the  $\partial p/\partial t$  term and made a reference to

СЗ

Appendix C where it is shown that this term may be considerable on a seasonal scale, see Fig. 6a. As discussed later in the paper, this fact can account for the discrepancy between the seasonal changes of global mean precipitation P and  $W_P$  derived from mean atmospheric dp/dt. We removed Section 3.2 from the revised paper but extended the discussion of this matter in Section 5.

The referee continues: As for section 3.1, there are several problems with the authors analysis. First, it should be clearly stated that the global integral of dh/dt is indeed zero in the absence of mass source and sink in the continuity equation.

We see no problem here, as this statement immediately follows from the obtained expression for  $I_h$ . We have included the suggested statement in the revised text.

The referee continues: First, it should be clearly stated that the global integral of dh/dt is indeed zero in the absence of mass source and sink in the continuity equation. This is the assumption made in Laliberte etal. It is also the continuity equation used in the MERRA Reanalysis. Hence, the authors should explicitly acknowledge that the claim that the integral of dh/dt is indeed correct within the assumptions made in the MERRA Reanalysis.

The absence of mass source and sink in the continuity equation is equivalent to the absence of a water cycle. Laliberté et al. (2015) focus was on thermodynamic aspects of the atmospheric water cycle. They could not and did not assume the absence of mass source and sink in the continuity equation.

Specifically, on p. 2 in their Supplementary Materials, Laliberté et al. (2015) state: "In the atmosphere, the moist entropy *s* and the specific humidity  $q_T$  satisfy  $\partial_t s + v \cdot \nabla s = \dot{s}$  and  $\partial_t q_T + v \cdot \nabla q_T = \dot{q_T}$ , where  $\dot{s}$  and  $\dot{q_T}$  are their respective sources and sinks." (Note that the latter equation is equivalent to  $dq_T/dt = \dot{q_T}$ .)

To make it clear that this statement is incompatible with the assumption of "absent sources and sinks in the continuity equation", we consider the continuity equation for air as a whole

$$\frac{\partial \rho}{\partial t} + \nabla \cdot (\rho \mathbf{v}) = \dot{\rho} \tag{c1}$$

together with the continuity equation for water vapor

$$\frac{\partial \rho_v}{\partial t} + \nabla \cdot (\rho_v \mathbf{v}) = \dot{\rho}.$$
(c2)

Noting that  $q_T \equiv \rho_v / \rho$  (Laliberté et al. (2015) neglect the tiny condensate content) we find from Eqs. (c1) and (c2) that

$$\dot{q_T} = \frac{\dot{\rho}}{\rho} \left( 1 - \frac{\rho_v}{\rho} \right). \tag{c3}$$

Thus, if Laliberté et al. (2015) had assumed  $\dot{\rho} = 0$ , they would have omitted not only the enthalpy term in their first law of thermodynamics but also the term proportional to  $dq_T/dt = \dot{q}_T$ . The latter term was the focus of their analysis though. Thus, the referee's suggestion that Laliberté et al. (2015) assumed  $\dot{\rho} = 0$  is not valid.

Neither is this assumption made in the MERRA database. What *is* assumed in the MERRA database and could also be assumed by Laliberté et al. (2015), as explained by Referee 2, see also Bosilovich et al. (2011), is that the vertically integrated continuity equation has no sources or sinks, that is  $\int \dot{\rho} dz = 0$ . However, as we discussed in detail in a previous comment [doi:10.5194/acp-2016-203-AC2], this relationship does make  $I_h$  equal to zero.

The referee continues: Second, it is perfectly valid to question the impact of mass source and sink on the framework of Laliberte et al., but this should be done clearly. In particular, The Bernoulli equation is an equality with 4 different terms. Changing the mass conservation does not only affect the global integral of dh/dt, but also that of ds/dt and dq/dt. The authors here assume -without proof- that the change in the enthalpy integral would be reflected solely in the work output.

If the referee's assumption about absent sources and sinks in the analysis of Laliberté et al. (2015) were correct, we would agree with this statement. For example, if Laliberte et al. defined *h* as enthalpy per unit dry air mass, then, as shown in our revised manuscript, the integral of dh/dt over *total dry air mass* would be zero. The other terms in the first law of thermodynamics would look different, too, if taken per dry air mass.

However, Laliberté et al. (2015) defined h as enthalpy per unit wet mass and, as is clear from their approach, integrated it over the entire mass of the atmosphere in the presence of mass sources and sinks. In this case the integral of dh/dt is not zero and its omission is not justified.

The referee continues: The broader issue here is that the discussion of section 3.1. and 3.2. is presented without context and incomplete. It could only be understood by very few potential readers. It makes the paper unnecessarily confusing and should be removed.

The work of Laliberte et al. 2015 is published in a journal aimed at a broad readership. Their account is clear: the authors present the first law of thermodynamics and set out to integrate it over atmospheric mass. All the terms in the corresponding equation are explicitly defined. Then they state that the global integral of one of the terms is zero [p. 540, right column, 3rd line from top]. We evaluate this integral and show that it is not zero and that its omission significantly impacts the paper's quantitative conclusions.

If we submitted our present manuscript without discussing Laliberte et al., a referee would rightly advise us to acquaint ourselves with the current literature and address the discrepancy between our results and those of Laliberté et al. (2015) (who analyzed the same MERRA database). We thus believe that our analysis of Laliberte et al. 2015 is an essential part of our study and have striven to present it as clearly as possible in the revised manuscript.

Comment 4 of Referee 4 [doi:10.5194/acp-2016-203-RC4]

4. Sections 2 and 3. The whole point of the exercise of this exercise seems to establish that the term  $\int_V dh/dt\rho dV$  assumed to be zero in Laliberte et al. is actually not zero, and that it is too large to be neglected. I agree with this statement, but the result obtained by the authors seems unphysical. The simplest way to show that the above term is not zero is through using using standard integration by parts

$$\int_{V} \frac{dh}{dt} \rho dV = \int_{V} \nabla \cdot (\rho hv) dV - \int_{V} h \nabla \cdot (\rho v) dV = \int_{\partial V} \rho hv \cdot n dS - \int_{V} h \nabla \cdot (\rho v) dV$$

How to estimate this term depends on how the velocity v, the density  $\rho$  and enthalpy h are defined. If v is the fully barycentric velocity, and  $\rho$  the full density, then mass conservation imposes  $\nabla \cdot (\rho v) = 0$ , and the term is controlled by boundary fluxes of enthalpy and is equal to the difference between the enthalpy evaporated minus the enthalpy precipitated. If  $\rho v$  is the mass flux of the gaseous component of moist air, then how to estimate this term is more complicated, since  $\nabla \cdot (\rho v) \neq 0$ . Physically, the term  $h\nabla(\rho v)$  is unphysical, since condensation or evaporation converts water vapour enthalpy  $h_v$  into liquid water enthalpy  $h_l$  and conversely, so should only involve the difference  $h_v - h_l = L$ , where L is latent heat, it should not involve the dry air enthalpy; the formula  $h\nabla(\rho v)$  involves the dry air enthalpy, however, which is part of the definition of h.

As was stated in our manuscript (see Eq. 5 on p. 3) and is perhaps better emphasized in our revision (p. C4, first paragraph in Author Comment 3 doi:10.5194/acp-2016-203-AC3), velocity v is the velocity of the gaseous component of moist air (i.e. of the substance that actually performs work). Enthalpy h is defined per unit mass of wet air (i.e. dry air mass plus water vapor mass). There is thus nothing unphysical in the resulting expression for the integral of dh/dt over total mass of dry air and water vapor depending on parameters of *both* dry air and water vapor.

Revised Section 4 (former Section 3) follows1.

**4 Practical implications of the obtained relationships**

In a recent effort to constrain the atmospheric power budget, Laliberté et al. (2015) used the thermodynamic identity

$$T\frac{ds}{dt} \equiv \frac{dh}{dt} - \alpha \frac{dp}{dt} + \mu \frac{dq_T}{dt},$$
(31)

where *s* is entropy, *h* is enthalpy,  $\mu$  is chemical potential (all per unit mass of wet air),  $\alpha$  is specific air volume and  $q_T$  is the mass fraction of total water2. Laliberté et al. (2015) neglected, as we do, the atmospheric liquid and solid water content3 and approximated  $q_T = q_v$ , where  $q_v$  is the mass fraction of water vapor.

vant equations from the previous sections are as follows:

$$\frac{dX}{dt} \equiv \frac{\partial X}{\partial t} + \mathbf{v} \cdot \nabla X \tag{5}$$

$$\frac{\partial \rho}{\partial t} + \nabla \cdot (\rho \mathbf{v}) = \dot{\rho}$$
(6)

$$W_p \frac{\tilde{V}}{\tilde{m}} = W_p \alpha = p \left( \frac{d\alpha}{dt} + \frac{\alpha}{\tilde{m}} \frac{d\tilde{m}}{dt} \right) = p \left( \frac{d\alpha}{dt} + \alpha^2 \dot{\rho} \right), \tag{11}$$

$$W = W_{III} = \frac{1}{\mathcal{S}} \int_{\mathcal{V}} p(\nabla \cdot \mathbf{v}) d\mathcal{V} = -\frac{1}{\mathcal{S}} \int_{\mathcal{V}} (\mathbf{v} \cdot \nabla p) d\mathcal{V} + I_T + I_s \equiv W_I + I_T + I_s,$$
(12)

$$I_T \equiv \frac{p_T}{S} \int_{z=z(p_T)} (\mathbf{v} \cdot \mathbf{n}) dS = 0,$$
(13)

$$I_s \equiv \frac{1}{S} \int_{S} p_s(\mathbf{v} \cdot \mathbf{n}) dS = 0.$$
(14)

<sup>1This section follows revised Section 3 from our previous comment doi:10.5194/acp-2016-203-AC4. The rele-C7

<sup>2The unconventional sign at the chemical potential term follows from  $\mu$  being defined in Eq. (31) relative to dry air: hence, when the relative dry air content diminishes this term is negative. For details see p. 8 in the Supplementary Materials of Laliberté et al. (2015).

<sup>3This assumption corresponds to an instantaneous removal of the non-gaseous water from the atmosphere by precipitation.

When integrating Eq. (31) over atmospheric mass, Laliberté et al. (2015) assumed that the enthalpy term vanishes,  $\int_{\mathcal{M}} (dh/dt) d\mathcal{M} = 0$ . This assumption was justified by noting that the atmosphere is approximately in a steady state. However, using the definition of material derivative (5), the steady-state continuity equation (6), the divergence theorem and the boundary conditions (13), (14) and noting that  $\rho \mathbf{v} \cdot \nabla h = \nabla \cdot (h\rho \mathbf{v}) - h\nabla \cdot (\rho \mathbf{v})$  and  $d\mathcal{M} = \rho d\mathcal{V}$ , we obtain

$$I_h \equiv \frac{1}{S} \int_{\mathcal{M}} \frac{dh}{dt} d\mathcal{M} = -\frac{1}{S} \int_{\mathcal{V}} h\dot{\rho} d\mathcal{V} \neq 0.$$
(32)

We can see that  $I_h$  is zero if only there are no sources and sinks of water vapor in the atmosphere, i.e. when  $\dot{\rho} = 0$ .

The physical meaning of this result is as follows. Enthalpy change per unit time in all air parcels (material elements) in a steady-state atmosphere is indeed zero. However, dh/dt is not equal to enthalpy change per unit mass of a given air parcel. (Likewise  $pd\alpha/dt$  is not equal to work per unit time per unit mass, see Eq. (11) above.) Indeed, for an air parcel of mass  $\tilde{m}$  total enthalpy of the parcel is  $h_p \equiv h\tilde{m}$ ; its change per unit mass is  $(dh_p/dt)/\tilde{m} = dh/dt + (h/\tilde{m})d\tilde{m}/dt \neq dh/dt$ . Therefore, the integral of dh/dt over total atmospheric mass is not zero. The same reasoning shows that if enthalpy were defined per unit dry air mass, such that  $h_p \equiv h\tilde{m}_d$ , then the integral of dh/dt per total dry air mass would be zero. Thus, we note the following general relationship. For any scalar quantity X in the view of the steady-state continuity equation (6) and Eqs. (13), (14) we have

$$\int_{\mathcal{V}} \frac{dX}{dt} \rho d\mathcal{V} = -\int_{\mathcal{V}} X \dot{\rho} d\mathcal{V}.$$
(33)

$$\int_{\mathcal{V}} \frac{dX}{dt} \rho_d d\mathcal{V} = 0. \tag{34}$$

Equation (34) follows from the continuity equation for dry air  $\nabla \cdot (\rho_d \mathbf{v}) = 0$ .

The magnitude of  $I_h$  (32) can be roughly estimated assuming that evaporation and condensation are localized at, respectively, the surface z = 0 and the mean condensation height  $z = \mathcal{H}_P$ . This approximation allows one to explicitly specify  $\dot{\rho}$  in (32) via the Dirac delta function  $\delta(z)$ :

$$\dot{\rho} = E(x,y)\delta(z) - P(x,y)\delta(z - \mathcal{H}_P), \quad \int \dot{\rho}dz = E(x,y) - P(x,y). \tag{35}$$

Here E(x,y) and P(x,y) are local evaporation and precipitation at the surface (kg m-2 s-1) with global averages E = P.

From (35) we have

$$I_h \approx -Eh_s + Ph(\mathcal{H}_P) \equiv -P\Delta h_c, \quad \Delta h_c \equiv h_s - h(\mathcal{H}_P), \quad h = c_p T + Lq_v.$$
(36)

Here  $c_p = 10^3 \text{ J kg}^{-1} \text{ K}^{-1}$  is heat capacity of air at constant pressure,  $L = 2.5 \times 10^6 \text{ J kg}^{-1}$  is latent heat of vaporization. We can see that  $I_h$  is proportional not to the difference between evaporation and precipitation (which can be locally arbitrarily small), but to the intensity of the water cycle E = P multiplied by the difference in air enthalpy between z = 0 and  $z = \mathcal{H}_P$ .

For  $\mathcal{H}_P \approx 2.5$  km (Makarieva et al., 2013) and  $q_v(\mathcal{H}_P) \ll q_{vs}$  we have  $-P\Delta h_c = -P(c_p\mathcal{H}_P\Gamma + Lq_{vs}) \approx -1$  W m-2. Here  $q_{vs} = 0.0083$  corresponds to global mean surface temperature  $T_s = 288$  K and relative humidity 80%; mean tropospheric lapse rate is  $\Gamma = 6.5$  K km-1. Global mean precipitation P (measured in a system of units where liquid water density  $\rho_w = 10^3$  kg m-3 is set to unity) is equal to  $P \sim 1$  m year-1, which in SI units corresponds to  $P = 3.2 \times 10^{-5}$  kg m-2 s-1. A more sophisticated estimate of  $I_h$  (36) presented in Appendix B yields -1.6 W m-2 with an accuracy of about 30%.

These estimates show that the enthalpy term cannot be neglected in Eq. (31) on either theoretical or quantitative grounds. By absolute magnitude the integral (36) is greater

than one third of the total atmospheric power  $W \approx 4 \text{ W m}^{-2}$  estimated by Laliberté et al. (2015) for the MERRA re-analysis (3.66 W m-2) and the CESM model (4.01 W m-2).

Laliberté et al. (2015) appear to have first calculated the mass integral of Tds/dt from the right-hand side of Eq. (31), then calculated  $\mu dq_T/dt$  from atmospheric data and then used the obtained values and again Eq. (31) to estimate the total atmospheric power as  $-(1/S) \int_{\mathcal{M}} \alpha (dp/dt) d\mathcal{M}$ . In such a procedure, putting  $\int_{\mathcal{M}} (dh/dt) d\mathcal{M} = 0$  should have overestimated W by about 1.6 W m-2. Since the omitted term is proportional to the global precipitation rate, it is crucial not only for a correct estimate of the mean value of W, but also for the determination of any trends related to precipitation.

Note also that even in the correct form, with the enthalpy term retained, Eq. (31) does not provide a theoretical constraint on W. This equation is an identity: it essentially defines ds/dt in terms of measurable atmospheric data. As is clear from Eq. (12), W can be estimated from the same data directly without involving entropy, as we discuss in the next section.

**References**

- Michael G. Bosilovich, Franklin R. Robertson, and Junye Chen. Global energy and water budgets in MERRA. *J. Climate*, 24:5721–5739, 2011. doi: 10.1175/2011JCLI4175.1.
- F. Laliberté, J. Zika, L. Mudryk, P. J. Kushner, J. Kjellsson, and K. Döös. Constrained work output of the moist atmospheric heat engine in a warming climate. *Science*, 347:540–543, 2015. doi: 10.1126/science.1257103.
- A. M. Makarieva, V. G. Gorshkov, A. V. Nefiodov, D. Sheil, A. D. Nobre, P. Bunyard, and B.-L. Li. The key physical parameters governing frictional dissipation in a precipitating atmosphere. *J. Atmos. Sci.*, 70:2916–2929, 2013. doi: 10.1175/JAS-D-12-0231.1.
- O. Pauluis. Water vapor and mechanical work: A comparison of Carnot and steam cycles. *J. Atmos. Sci.*, 68:91–102, 2011. doi: 10.1175/2010JAS3530.1.

---

## Author Comment (AC6) · 25 Jul 2016

Here we reply to Comments 1 and 2 of Referee 1, Comments 3, 4 and 6 of Referee 2, Comment 3 of Referee 3 and Comments 4, 5 and 6 of Referee 4 addressing the challenge of obtaining reliable estimates of the distinct terms in the atmospheric power budget. Specifically, Referee 1 suggested that we should address the uncertainties surrounding the atmospheric power estimates, which we do in the revised text.

Following the recommendations of Referees 1, 2 and 3 we have extended our analysis in Section 5. We now analyze the 3-hourly MERRA dataset for the entire period 1979-2015. To illustrate the impact of temporal resolution on our results, we additionally analyze daily and monthly mean MERRA data for the same period. Furthermore, we assess NCAR/NCEP daily and monthly data for the last thirty five years.

Comment 1 of Referee 1 [doi:10.5194/acp-2016-203-RC1]:

*" 1. The evaluation of the gravitational power of precipitation (GPP) as presented in Appendix A, which is used to verify the GPP estimated from the MERRA data, contains a significant source of uncertainties as it depends so much on different input parameters as listed in Appendix A. Likewise, the GPP estimated from MERRA also depends strongly on the data resolution, the number of vertical levels, or the numerical approximations. Before trying to explain the discrepancies between GPP obtained from GPCP data and the GPP obtained from the MERRA data, the authors should at least quantify the errors in all of your numbers. While the authors claim that the uncertainty of your estimated GPP from GPCP is 30%, there is no guarantee that the difference between the two GPP estimations will be statistical significance. Afterall, 30% of 1 W m$^{-2}$ is 0.3, and so it could be anything from 0.7-1.3 W m$^{-2}$, which may be comparable to the GPP computed from the MERRA data;"*

We agree with the above points and have included a discussion of uncertainties in a separate subsection in the revised Section 5. Specifically, we make two notes regarding our conclusion that $W_P$ in MERRA is underestimated. First, as illustrated by the derivation of Eqs. (20)-(22) (see the footnote[1]), $W_P$ *must* depend on data resolution. Indeed, $W_P$ derives from the vertical air velocity and thus describes rainfall associated with air motions at the considered scale.

Meanwhile the theoretical estimate of $W_P$ is based on the total observed rainfall and
* * *
[1]

$$W = -\frac{1}{S}\int_{\mathcal{V}} \mathbf{v}\cdot\nabla p \, d\mathcal{V} \equiv W_K + W_P, \tag{20}$$

$$W_K \equiv -\frac{1}{S}\int_{\mathcal{V}}(\mathbf{u}\cdot\nabla p)d\mathcal{V} + W_c \approx -\frac{1}{S}\int_{\mathcal{V}}\mathbf{u}\cdot\nabla p \, d\mathcal{V}, \quad W_c \equiv -\frac{1}{S}\int_{\mathcal{V}}\rho_c(\mathbf{w}\cdot\mathbf{g})d\mathcal{V}, \tag{21}$$

$$W_P \equiv -\frac{1}{S}\int_{\mathcal{V}}\rho\mathbf{w}\cdot\mathbf{g}\,d\mathcal{V} = -\frac{1}{S}\int_{\mathcal{V}}gz\dot{\rho}\,d\mathcal{V} = Pg\mathcal{H}_P, \quad P \equiv -\frac{1}{S}\int_{z>0}\dot{\rho}\,d\mathcal{V}. \tag{22}$$

thus assesses cumulative gravitational power of precipitation at all scales. If $W_P$ derived from MERRA coincided with theoretical $W_P$, that would mean that no rainfall is associated with the air motions at a scale finer than 100 km and six hours. Since the scale of convection is of the order of a few kilometers or less, apparently some rain must remain unresolved by the larger-scale motions. Therefore, the fact that $W_P$ in MERRA is lower than its independent theoretical estimate does not indicate inconsistencies in the database.

Second, the theoretical estimate in Appendix A (now B) illustrates how the various parameters entering the value of $W_P$ impact its magnitude. The bottomline however is provided by the TRMM-derived estimate of Pauluis and Dias (2012), which is $1.5$ W m$^{-2}$ for the area between 30° N and 30° S. So, global $W_P$ cannot be lower than $0.75$ W m$^{-2}$. If it is $0.75$ W m$^{-2}$, this means that there is no precipitation at all in the extratropics. However, since extratropical precipitation is significant (2.2 mm day$^{-1}$ versus 3.1 mm day$^{-1}$ in the tropics, see Fig. 5 in our manuscript), it will contribute to the global value of $W_P$. Even we assume that all extratropical rainfall precipitates from $\mathcal{H}_P = 1$ km (which is clearly an underestimate), global $W_P$ will constitute $0.87$ W m$^{-2}$. Therefore, the uncertainty of the lower limit of our estimate $W_P = 1$ W m$^{-2}$ is about 10%.

We note that while formally the analyzed MERRA data have a 3-hourly resolution, they represent an analysis of 6-hourly data with the intermediate values provided for assessing partial derivatives over time of the corresponding variables. To illustrate the impact of temporal resolution on the atmospheric power budget we compared $W$, $W_P$ and $W_K$ calculated from 6-hourly, daily and monthly mean MERRA data. These results are shown in Fig. 1 attached to this response and present in the revised Section 5.

With temporal resolution changing from 1 month to 6 hours $W$, $W_K$ and $W_P$ rise, respectively, from 1.02, 0.33 and 0.69 W m$^{-2}$ to 3.27, 2.46 and 0.81 W m$^{-2}$. This supports our conclusion that with growing resolution of the available observations the kinetic power $W_K$ will increase (presumably until the resolution of the smaller-scale

convective motions is reached). Assuming a power law for the scaling of $W_K$ with temporal resolution $r$

$$\frac{W_K(r_1)}{W_K(r_2)} = \left(\frac{r_1}{r_2}\right)^k, \quad k = \frac{\log[(W_K(r_1)/W_K(r_2)]}{\log[r_1/r_2]}, \tag{c1}$$

where $r$ is temporal resolution in hours, from $W_K(24) = 1.78$ W m$^{-2}$ (daily) and $W_K(6)$=2.46 W m$^{-2}$ (six hours) from Eq. (c1) we find $k = -0.23$. Using this value and $W_K(6)$=2.46 W m$^{-2}$ we find $W_K(1)$=3.7 W m$^{-2}$, i.e. kinetic power of convective air motions having temporal scale of 1 hour should be about 4 W m$^{-2}$. This is consistent with the theoretical estimate for condensation-induced air circulation.

Comment 2 of Referee 1 [doi:10.5194/acp-2016-203-RC1]: *"2. Estimations of the total atmospheric power $W$ and $W_K$ are subject to similar uncertainties as mentioned in my comment # 1 above. At resolution of 1.25 degree and 42 vertical levels, any global estimation of the total integrated energy and kinetic energy contains large variation, let alone the difference between two. Have the authors tried the NCEP reanalysis or ECMWF dataset at different resolutions to see how sensitive your estimations are? As long as we don't have reliable estimation of $W, W_K$, and GPP, explanation for the difference would provide little scientific value."*

We agree with the above comments and extended our analysis to include the NCAR/NCEP daily data for the same period 1979-2015. This yielded instructive results.

As we note in our manuscript (p. 8) and emphasize in the revision, kinetic power $W_K$ is derived from observations of wind velocities and should be associated with much less uncertainty than the vertical velocity. This is confirmed by comparison of $W_K$ across the MERRA and NCAR/NCEP databases, Fig. 2. The profiles of $W_K$ are close at most latitudes and the global mean values are also similar: 1.75 W m$^{-2}$ for NCAR/NCEP and 1.79 W m$^{-2}$ for MERRA[2].
* * *
[2] We note that the spatial resolution of a particular re-analysis is not necessarily the same as the spatial resolution

The situation is different for total power $W$, which depends on the vertical velocity. The global value of $W$ appears as a near-zero sum of large terms of different signs that describe the ascending and descending air motions. This is the reason for its high uncertainty: in order to yield a global $W$ of the same accuracy as $W_K$, these vertical air flows must be deduced from the continuity equation with an accuracy exceeding that of the horizontal air flows (that define $W_K$) by two orders of magnitude. However, this cannot be readily achieved, since the only source of information about the vertical air flow is the continuity equation and the observations of the horizontal air flow. As a result of this high uncertainty, $W$ appears inconsistent across the databases.

In Fig. 2b we show the dependence of the columnar mean $\Omega$ (Eq. 23 in our manuscript)

$$W = \langle\Omega\rangle, \quad \Omega \equiv -\frac{1}{\mathcal{S}}\int_{\mathcal{V}}\omega d\mathcal{V}, \quad \omega \equiv \frac{dp}{dt} \equiv \frac{\partial p}{\partial t} + \mathbf{v}\cdot\nabla p, \tag{c2}$$

on latitude in NCAR/NCEP versus MERRA database. One can see that, similar to $W_K$ in Fig. 2a, the differences between the derived zonal distributions are relatively small. However, as far as the local magnitudes of $\Omega$ exceed its global mean value by about two orders of magnitude, it turns out that these small local differences translate into profound differences in the global atmospheric power $W$. Our analysis suggests that global $W$ estimated from Eq. (20) in the NCAR/NCEP daily data is *negative* and constitutes $-6.06\,\mathrm{W\,m^{-2}}$ versus $2.45\,\mathrm{W\,m^{-2}}$ in MERRA. Unless there is some technical error involved (which is always possible but appears unlikely since our estimates of $W_K$ are consistent across the databases and since taking the integral of pressure velocity over volume is straightforward), the obtained results suggest that the global estimate of $W$ and, hence, $W_P$ in a given dataset is significantly impacted by the particular procedures involved to calculate pressure velocity $\omega$ from the continuity equation.
* * *
of the experimental data the re-analysis presents. While using numerical modelling it is possible to rescale the observed data to a finer resolution, the results will not necessarily reflect the processes in the real atmosphere. The similarity between daily data in MERRA (spatial resolution 1.25x1.25 degrees) and NCAR/NCEP (2.5x2.5 degrees) may thus reflect the fact that the raw experimental data can have an average resolution coarser than in either dataset.

Specifically, as also pointed out by Referee 2 [doi:10.5194/acp-2016-203-RC2, Comment 2], in the MERRA database pressure velocity is calculated involving information on the local water cycle in such a manner that the vertically integrated continuity equation has a zero source/sink. This procedure takes some information about local precipitation into account (see Eq. 15 in our manuscript) and, as a result, can yield a reasonable value for total atmosperic power, for $W_P$ and for other terms depending on $\dot\rho$. This procedure should also be responsible for the fact that the MERRA-derived $W_P$ has a relatively minor dependence on temporal resolution compared to $W_K$. Indeed, with transition from monthly to 6-hourly resolution $W_P$ barely increases by 30%, Fig. 1c, while $W_K$ rises almost eight-fold. This is because the long-term mean local rainfall rate does not depend on temporal resolution being a cumulative representation of precipitation events at all scales.

To our knowledge, atmospheric power has not been systematically assessed in re-analyses in the straightforward way outlined by Eq. (c2) – i.e. as the integral of pressure velocity over atmospheric volume. Thus we cannot compare our NCAR/NCEP results with any published estimate. Rather, atmospheric power was commonly assessed as the total dissipation rate in the atmospheric energy cycle, i.e. as work per unit time of the turbulent friction force (see, e.g., Eq. (A3) of Boer and Lambert (2008)). In particular, Boer and Lambert (2008), when comparing atmospheric power across the re-analyses and global circulation models, quoted a figure of $2$ W m$^{-2}$ for the 6-hourly NCAR/NCEP data (see Table 3 of Boer and Lambert (2008)). Our results for the daily NCAR/NCEP data for $W_K$ is $1.75$ W m$^{-2}$, which is consistent with the above estimate taking into account the dependence of $W_K$ on temporal resolution as shown in Fig. 1b. Therefore, the estimates reported by Boer and Lambert (2008) do not represent total atmospheric power, which thus remains unstudied across the models and re-analyses datasets.

Our comparison of $W$ between NCAR/NCEP and MERRA highlights the high uncertainty in the calculation of vertical velocities. The estimates of total atmospheric power

$W$ and the gravitational power of precipitation $W_P$ made from re-analyses according to our Eqs. (20)-(22) should be used to constrain the calculation of vertical velocities in re-analyses thus improving their consistency in representing the atmospheric energetics.

In particular, while the MERRA database, which does account for precipitation when calculating $\omega$, produces a reasonable estimate of total atmospheric power $W$ and gravitational power $W_P$, Fig. 1c shows that this $W_P$ has a pronounced seasonal cycle that appears unreasonable. In July (when the global temperature is at its maximum, see Fig. 6b in our manuscript), global $W_P$ is nearly twice lower than it is in January. This seasonal variation does not correlate with the seasonal global rainfall (see Fig. 1b in our manuscript) and may be an artefact of the procedures involved to calculate pressure velocity in the MERRA database. Our results call for a systematic study of the atmospheric power budget across the re-analyses and also across global circulation models on the basis of Eqs. (20)-(22).

Comment 3 of Referee 3 [doi:10.5194/acp-2016-203-RC3]:

*"3. The paper is poorly constructed. It is mainly three separate studies. Sections 2-4 attempt a theoretical discussion of the issues that mostly reprise previous work. It is unnecessarily confusing. Section 5 is the main 'new' result. The computation done are fairly routine, and the result in line with what we know. The inability of the authors to produce a consistent figure for Wp is distressing and should be better addressed in the revision. Section 6 is a lengthy disgression which is mostly a repeat of the authors previous work."*

We believe that the revised paper is now much clearer and presents a coherent theme. We underline that the entire literature on this subject is somewhat confusing and it is the need to identify and examine the inconsistencies in other studies that leads to difficulties. Our revision is attentive to these difficulties (e.g. the different formulations of $W$).

We note that published approaches to $W_P$ suffer important inconsistencies. Pauluis et al. (2000) estimated, on theoretical grounds, that tropical $W_P$ (between 30N and 30S) should be between 2 and 4 W m$^{-2}$. Pauluis and Dias (2012) analyzed TRMM data to conclude that tropical $W_P$ is, rather, 1.8 W m$^{-2}$. Makarieva et al. (2013) likewise on theoretical grounds, suggested that Pauluis et al. (2000) overestimated tropical $W_P$ by around one-hundred percent. Their results led Pauluis and Dias to revise their calculations and publish a revised TRMM-based estimate of 1.5 W m$^{-2}$ for the tropics as a corrigendum to their 2012 work. On the other hand, Makarieva et al. (2013) suggested that global $W_P$ should be around 0.8 W m$^{-2}$; in our present work we show that the true value is around 1 W m$^{-2}$ and we address the associated uncertainties.

As we discussed in our reply to Comment 2 of Referee 1, we show in the revision that the inconsistency in the estimates of $W_P$ as well as of total power $W$ is an inherent property of the re-analyses. We disagree that this is already known, since we find no estimates of global $W_P$ from re-analyses or otherwise. This is indeed surprising given recent emphasis on the thermodynamic aspects of the water cycle (see, e.g., Pauluis, 2015). We hope that our revised work brings greater clarity to this matter.

In particular, our results suggest that if Laliberté et al. (2015) used NCAR/NCEP rather than MERRA data for their analysis, they would have obtained a negative value for total atmospheric power (and, hence, $W_P$). Key to their result is the procedure of zeroing pressure velocity at the surface between the modelling steps; unfortunately, its details were not reported. If a different procedure were used, the results would be different as well.

Comment 3 of Referee 2 [doi:10.5194/acp-2016-203-RC2]: *3. Computing the work from MERRA data. As mentioned before, the MERRA product has many vertically integrated budget variables that allow one to quantify each one of the term in the energy equation. For this review, I've looked at the kinetic energy generation 1980-1985 and the yearly average gives 3.40-3.48 W/m$^2$ for the integral of $\omega\alpha$ and 3.6-3.8 W/m$^2$ when including the kinetic energy generation from the analysis step. The kinetic en-*

*ergy generation is balanced by damping from the numerical dissipation, the dynamical remapping and the physically parametrized frictional dissipation. This means that the estimates provided in section 5.1 are substantial underestimates.*

We agree with the referee that our estimates of $W$ are substantial underestimates of the real atmospheric power; indeed, it is our major point. We do not consider the kinetic energy generation from the analysis step. (Neither did Laliberté et al. (2015)). We explicitly address how the global atmospheric power can be estimated using the re-analysis pressure and velocity at their face value. In our revised manuscript we present an analysis of the entire period 1979-2015. We note that for $W_K$ our results practically coincide with those of Huang and McElroy (2015), who reported $W_K = 2.46$ W m$^{-2}$ for 1979-2010. Our calculations for the same period give $W_K = 2.45$ W m$^{-2}$.

Our annual estimates of $W$ for 1980-1985 range from 3.20 to 3.28 W m$^{-2}$. This is 6% smaller than the referee's. The discrepancy may stem from two sources. First, the dataset with the vertically integrated $\omega\alpha$ derives from the 1-hourly surface dataset (presumably MAT1NXINT [tavg1_2d_int_Nx]), while our estimate derives from the 3-hourly dataset (MAI3CPASM [inst3_3d_asm_Cp]). As we have shown that $W$ increases with finer temporal resolution, see Fig. 1a, this may explain the 6% discrepancy. Second, the discrepancy may stem from a difference in the boundary condition for $\omega$ at the surface.

It is not explicitly indicated in the dataset how the integration was performed. We have explicitly stated what boundary condition we are using to make our analysis tractable and comparable to other studies. Furthermore, we investigated the impact of the surface boundary condition on our analysis for each variable and showed that the associated uncertainty is about 6%.

Specifically, for $W$ and $W_K$ we estimated the value of $\omega$ and $\mathbf{u} \cdot \nabla p$ at the surface in two ways (see Appendix B in our manuscript). One is to assume that air velocity at the surface is zero, $\mathbf{v} = 0$, another is to linearly extrapolate $\omega$ and $\mathbf{u} \cdot \nabla p$ from the nearest

pressure level to the surface. Our increased attention to the boundary layer is justified by the fact that horizontal velocity experiences significant non-uniform changes along the vertical. In the limit of an infinitely precise vertical resolution the two approaches should give the same value. In the real atmosphere they produce different results.

Specifically, the extrapolated $W_K$ ($W_{K1}$ in our manuscript) turns out to be higher than $W_K$ calculated assuming $\mathbf{v} = 0$. This has to do with the vertical profile of $W_K$ shown in Fig. 3. Kinetic energy generation grows with increasing pressure in the lower atmosphere. Extrapolation of this dependence to the surface yields a positive surface value for kinetic energy generation. Thus, $W_K$ obtained from this extrapolation is higher than when we assume that $\mathbf{v} = 0$, such that no kinetic energy is generated at $z = 0$

In contrast, the estimate of total power $W$ is smaller when extrapolated than when assuming zero velocity at the surface. This has to do with a different distribution of pressure velocity over pressure levels, Fig. 3. Here the lowest layer between 975 hPa and the surface makes a large negative contribution to the total $W$. This is because the air predominantly descends in the regions of higher surface pressure. Therefore with one and the same $\omega$ at $975$ hPa, the layer where the air descends and surface pressure is about, say, 1020 hPa is thicker than where the air ascends and surface pressure is about 1000 hPa. Since $W$ is proportional to $-\omega$, in the result the net contribution of the lower layer to global $W$ is negative.

The difference between the two estimates for $W$ and for $W_K$ is about 10%. The difference between $W_P$ values obtained by the two means is greater. $W_P$ obtained by interpolation is considerably smaller than $W_P$ obtained assuming that zero velocity at the surface. This suggests that our conclusion about $W_P$ being underestimated in MERRA is robust.

Comment 4 of Referee 2 [doi:10.5194/acp-2016-203-RC2]: *4. In section 5.1, I do not see the use for $W_1$. And why not use $\omega_s = \partial_t p_s + \mathbf{v}_s \cdot \nabla_H p_s$, with $\nabla_H$ being the horizontal gradient? The $p_s$ and $\mathbf{v}_s$ are both available and this is the right expression. Maybe that*

*could fix their underestimate of $W$.*

The use of $W_1$ is discussed in our reply to Comment 3 of Referee 2. $W_1$ and $W_{K1}$ were used to investigate the uncertainty associated with insufficient data resolution in the boundary layer.

One cannot use $\omega_s = \partial_t p_s + \mathbf{v}_s \cdot \nabla_H p_s$, because the horizontal gradient of surface pressure $\nabla_H p_s$ only exists if the surface is horizontal (i.e. has invariant geopotential height). Since the geopotential height of the real surface varies, surface pressure is much more affected by this variability in the vertical plane than by any effects in the horizontal plane, which prohibits the use of $p_s$ for a reliable determination of $\omega_s$.

Moreover, since the term $\mathbf{v}_s \cdot \nabla_H p_s$ is present in the surface values of both $\omega$ and $\mathbf{u} \cdot \nabla p$, even if this term were added, this would not change the difference between the global $W$ and $W_K$.

To simplify the presentation, in the revised text we everywhere use $W_2$ and $W_{K2}$ and discuss $W_1$ and $W_{K1}$ only in the section devoted to the uncertainties.

Comment 6 of Referee 2 [doi:10.5194/acp-2016-203-RC2]: *6. Finally, I'm not sure the following sentence is logically true: "The fact that $W_{Kc}$ is likewise higher than our MERRA-derived kinetic power, testifies in favor of the theoretical estimate". All it means is that $W_{Kc}$ is potentially a right upper bound. The only way to check whether it is the right upper bound would be to either verify if it holds on other Earth-like planets or using simulations with increasing resolution and seeing that it describes the scaling. As I said before, the last two sections of this manuscript are really too speculative in their current form and they are dragging down the original results described in sections 5.*

In our manuscript we explained what we mean by *testifying in favor of the theoretical estimate*. The phrase quoted by the referee is immediately followed by an explicit clarification (p. 16): "To explain this point in greater detail: Eq. (15) and Eq. (22),

which estimate, respectively, the mass integral of $dh/dt$ and the gravitational power of precipitation $W_P$, are not dependent on the assumption that air circulation on Earth is condensation-driven. These equations describe how the corresponding variables can be estimated from observations. Both variables are approximately proportional to the volume integral of net condensation rate in the atmospheric interior $-\int_{z>0} \dot{\rho} d\mathcal{V}$. We notice that both variables estimated from the MERRA database are by 50-70% smaller than when estimated independently from the observed global precipitation $P$. We attribute this to the insufficient spatial resolution of the air motions associated with condensation. Now, we predict that if atmospheric circulation is condensation-driven, kinetic power generation is also proportional to $P$, as described by Eq. (25). Since we already know that not all condensation is resolved in the MERRA dataset, we can expect that kinetic energy generation estimated from MERRA using Eq. (21) will be smaller by a comparable magnitude than its theoretical estimate (25). ***This is what we find. If kinetic power was unrelated to precipitation, we could not expect that its value would be smaller than the precipitation-based theoretical estimate (25). If, on the other hand, our theoretical estimate turned out to be smaller than the MERRA-derived estimate, $W_{Kc} < W_K$, this would testify against condensation-induced dynamics.***"

In the revised manuscript we show that kinetic power $W_K$ does grow with finer resolution, Fig. 1, and that the theoretically predicted 4 W m$^{-2}$ is the plausible limit observed at convective scale.

Comment 2 of Referee 4 [doi:10.5194/acp-2016-203-RC4]:

*2. The water cycle is generally regarded as making the atmospheric heat engine less efficient as the result of part of the solar forcing being expanded in lifting water vapour against the gravity field, part of which is then removed through precipitation, leaving only the residual to power the atmospheric circulation, an idea proposed by Pauluis and reprised in Laliberte et al. (2015). It seems that this should be discussed.*

The referee's account of the work of Laliberté et al. (2015) appears to be a misunderstanding. There are three relevant quantities: the power of a Carnot cycle $W_C$, the kinetic atmospheric power $W_K$ and the total atmospheric power $W$. The focus of Pauluis et al. (2000) was indeed to show that $W_K$ is lower than $W$ because, using the referee's words, solar power is "lifting water vapour against the gravity field, part of which is then removed through precipitation, leaving only the residual to power the atmospheric circulation". However, Pauluis (2011) advanced a different statement: that total power $W$ is lower than Carnot power $W_C$ because of the irreversible processes like water vapor diffusion. Laliberté et al. (2015) were likewise concerned about why $W$ is smaller than $W_C$ and did not assess the gravitational power of precipitation.

This misunderstanding might have stemmed from the comment of Pauluis (2015) on the work of Laliberté et al. (2015), where the two statements, $W_K < W$ and $W < W_C$, became mixed. To provide some context, an ideal atmospheric Carnot cycle consuming heat flux $F = 100$ W m$^{-2}$ at surface temperature $T_{in} = 300$ K and releasing heat at $T_{out} = T_{in} - \Delta T$ with $\Delta T_C = 30$ K, would generate kinetic energy at a rate of $W_C = F(\Delta T_C/T_{in}) = 10$ W m$^{-2}$. Laliberté *et al.* (2015) estimated total atmospheric power $W$ at around 4 W m$^{-2}$. Comparing their result with $W_C$, Pauluis (2015) noted that "estimates for the rate of kinetic energy production by atmospheric motions are about half this figure". Here confusion has apparently arisen between total atmospheric power $W$ and kinetic power $W_K$ (because Laliberté et al. (2015) assessed only $W$ but not $W_K$, the latter being about 2.5 W m$^{-2}$, i.e. a quarter rather than half of $W_C$). Indeed, Pauluis (2015) continued that "the difference is very likely due to Earth's hydrological cycle, which reduces the production of kinetic energy in two ways", one of which is the gravitational power of precipitation $W_P$ and the other is the irreversible diffusion processes. However, from our Eqs. (20)-(22), $W_P$ reduces $W_K$ compared to $W$ but it does not reduce $W$ compared to $W_C$, since $W_K + W_P = W < W_C$.

Comment 6 of Referee 4 [doi:10.5194/acp-2016-203-RC4]: *6. On a last note, I have a hard time accepting that the term* $u \cdot \nabla p$ *is something observable, given that the only*

*way to estimate this term can only be done by means of a numerical model; likewise for the internal condensation/precipitation terms.*

In the meteorological literature it is common to refer to the re-analyses data as to *observations* using which models outputs could be verified – see, for example, the study of Boer and Lambert (2008) devoted to the atmospheric energy cycle. This is because the re-analyses aim to systematize available observations of air pressure, velocity, temperature, humidity etc. in a coherent form. Air pressure and velocity are the basic observational parameters recorded. Likewise, precipitation is directly measured at the surface as well as assessed in the tropical atmosphere with use of satellites (the TRMM mission).

Since vertical velocities are small compared to horizontal velocities, they cannot be derived directly from observations. It is in this sense that the term $\mathbf{u} \cdot \nabla p$ is observable with a good accuracy, while the term $\rho w g$ responsible for the gravitational power of precipitation is not. This latter term can only be derived from observations using additional assumptions. Because of this difference, we estimate $W_K$ with less uncertainty than $W_P$.

Comment 5 of Referee 4 [doi:10.5194/acp-2016-203-RC4]: *5. Section 4. I don't really understand why this decomposition is useful. Indeed, a well known consequence of making the hydrostatic approximation is to filter out the contribution of the vertical velocity to the kinetic energy. As a result, the evolution equation for the kinetic energy becomes*

$$\rho \frac{D}{Dt}\frac{u^2}{2} + u \cdot \nabla p = \rho F \cdot u \qquad \text{(c3)}$$

*so that in equilibrium*

$$\int_V u \cdot \nabla p \, dV = Friction, \qquad \text{(c4)}$$

*which shows that only what the authors call the kinetic energy power (the conversion between kinetic energy and available potential energy) becomes relevant to understand*

*how the atmospheric circulation is powered. As is also well known, even without the hydrostatic approximation, the budget of gravitational potential energy is zero*

$$\int_V \rho g w \, dV = 0 \qquad \text{(c5)}$$

*where $\rho w$ is the total mass flux, and hence decoupled from the kinetic energy budget. One may if one so desires to separate the total mass flux into gaseous and liquid components, and restrict attention to the former, for which the GPE budget becomes*

$$\frac{d(GPE)}{dt}\bigg|_{gas} = \underbrace{\int_V \rho g w \, dV}_{-SW_P} + GAS\ DESTRUCTION = 0, \qquad \text{(c6)}$$

*where $\rho w$ is now the gaseous mass flux only, GAS DESTRUCTION means GPE sink due to destruction of water vapour mass by condensation, but that does not make it less decoupled from the horizontal kinetic energy budget, where the underlined term is what the authors call the* power of precipitation*, whatever that means. Physically, this term represents primarily a conversion with internal energy, and is not directly related to the kinetic energy of the system, making its usefulness for clarifying the atmospheric power budget dubious. Moreover, it is also well known that for a hydrostatic fluid, it is the total potential energy of the system (i.e., the enthalpy) that matters, given that large variations in gravitational potential energy are compensated by large variations in internal energy, with no impact on kinetic energy. The focus on gravitational potential energy, therefore, is at odds with the common wisdom that GPE is not useful to consider on its own. The claim that GPE variations are somehow connected with kinetic energy production is odd, given that the hydrostatic approximation is unconnected to the vertical velocity field.*

The decomposition of total atmospheric power $W$ into the kinetic power of winds $W_K$ and the gravitational power of precipitation $W_P$ is useful in several ways. First, as we

discussed above in response to Comment 6 of Referee 4, $W_P$ and $W_K$ in re-analyses are characterized by substantially different uncertainties, so it is useful to keep a separate record for them. Second, $W_P$ can be estimated independently from wind velocities using observed precipitation; this information can be used to constrain vertical velocities. Third, since thermodynamics constrains total power $W$ and not kinetic power $W_K$ or $W_P$ separately, it is necessary to clearly differentiate between $W$, $W_K$ and $W_P$ from a theoretical viewpoint. Distinguishing these components can help avoid confusions when comparing results from different studies (see also above our reply to Comment 2 of Referee 4). For example, given the modern concern about renewable energy resources it is necessary to understand that the so-called "wind power" (Marvel et al., 2013) as well as the river hydropower (which is part of $W_P$) are not the total power of the atmosphere.

We also note that in the presence of condensate the vertical distribution of gaseous air is not hydrostatic; the condensate loading term describes the generation of kinetic energy of the vertical air motions and is not zero. Furthermore, the integral of the right-hand part of the referee's equation (c3) is not zero in the presence of phase transitions, so Eq. (c4) does not hold. This is discussed in detail in the revised section 3 (see doi:10.5194/acp-2016-203-AC4, Eq. 29 on p. C10).

**References**

G. J. Boer and S. Lambert. The energy cycle in atmospheric models. *Climate Dynamics*, 30: 371–390, 2008. doi: 10.1007/s00382-007-0303-4.

J. Huang and M. B. McElroy. A 32-year perspective on the origin of wind energy in a warming climate. *Renewable Energy*, 77:482–492, 2015. doi: 10.1016/j.renene.2014.12.045.

F. Laliberté, J. Zika, L. Mudryk, P. J. Kushner, J. Kjellsson, and K. Döös. Constrained work output of the moist atmospheric heat engine in a warming climate. *Science*, 347:540–543, 2015. doi: 10.1126/science.1257103.

A. M. Makarieva, V. G. Gorshkov, A. V. Nefiodov, D. Sheil, A. D. Nobre, P. Bunyard, and B.-L. Li.

The key physical parameters governing frictional dissipation in a precipitating atmosphere. *J. Atmos. Sci.*, 70:2916–2929, 2013. doi: 10.1175/JAS-D-12-0231.1.

K. Marvel, B. Kravitz, and K. Caldeira. Geophysical limits to global wind power. *Nature Climate Change*, 3:118–121, 2013. doi: 10.1038/nclimate1683.

O. Pauluis. Water vapor and mechanical work: A comparison of Carnot and steam cycles. *J. Atmos. Sci.*, 68:91–102, 2011. doi: 10.1175/2010JAS3530.1.

O. Pauluis and J. Dias. Satellite estimates of precipitation-induced dissipation in the atmosphere. *Science*, 335:953–956, 2012. doi: 10.1126/science.1215869.

O. Pauluis, V. Balaji, and I. M. Held. Frictional dissipation in a precipitating atmosphere. *J. Atmos. Sci.*, 57:989–994, 2000. doi: 10.1175/1520-0469(2000)057<0989:FDIAPA>2.0.CO;2.

Olivier M. Pauluis. The global engine that could. *Science*, 347:475–476, 2015. doi: 10.1126/science.aaa3681.

[Figure]

Figure 1: Long-term mean atmospheric power in MERRA as dependent on temporal resolution: 6-hourly (solid curves), daily (dashed curves) and monthly (dotted curves). (a) total power $W$ (20), (b) kinetic power $W_K$ (21), (c) gravitational power of precipitation $W_P = W - W_K$.

**Fig. 1.**

[Figure]

Figure 2: Long-term mean zonally averaged atmospheric power (daily data for 1981-2015) in the MERRA versus NCAR/NCEP re-analysis as dependent on latitude (red solid curve: MERRA, black dashed curve: NCAR/NCEP). (a) kinetic power $W_K$ (21), (b) total power $W$ (20).

**Fig. 2.**

[Figure]

Figure 3: Atmospheric power within the 41 pressure layers enclosed by the 42 pressure levels in the MERRA dataset. See Appendix B for the full list of pressure levels. The lowest bar of the histograms corresponds to the layer with pressure less than 975 hPa. Sum of the histogram values over all layers gives the global values of the atmospheric power.

**Fig. 3.**

---

## Author Comment (AC7)

**Response to the referees' comments**
**on "Quantifying the global atmospheric power budget"**
**doi:10.5194/acp-2016-203**

**1 Referee 1 [doi:10.5194/acp-2016-203-RC1]**

**1.1 Comment 1**

*Summary: In this study, the authors attempt to examine the gap between the gravitational power of precipitation, which is estimated as the total atmospheric power – kinetic power from the MERRA dataset, and another independent gravitational power of precipitation estimated from the surface precipitation data. While I can see a good merit of this work, I found the paper contains several loopholes that need to be clarified before the manuscript can be accepted for publication. Also, the presentation of this work is somewhat confusing, and can be simplified substantially to make it clearer. My concerns are given as bellow:*

*1. The evaluation of the gravitational power of precipitation (GPP) as presented in Appendix A, which is used to verify the GPP estimated from the MERRA data, contains a significant source of uncertainties as it depends so much on different input parameters as listed in Appendix A. Likewise, the GPP estimated from MERRA also depends strongly on the data resolution, the number of vertical levels, or the numerical approximations. Before trying to explain the discrepancies between GPP obtained from GPCP data and the GPP obtained from the MERRA data, the authors should at least quantify the errors in all of your numbers. While the authors claim that the uncertainty of your estimated GPP from GPCP is 30%, there is no guarantee that the difference between the two GPP estimations will be statistical significance. Afterall, 30% of 1 W $m^{-2}$ is 0.3, and so it could be anything from 0.7-1.3 W $m^{-2}$, which may be comparable to the GPP computed from the MERRA data;*

Response [see also doi:10.5194/acp-2016-203-AC6]:

We agree with the above points and explicitly discuss the uncertainty of $W_P$ in Appendix B, last paragraph. Specifically, we make two notes regarding our conclusion that $W_P$ in MERRA is underestimated. First, as illustrated by the derivation of Eqs. (20)-(22) (see the footnote[1]), $W_P$ *must* depend on data resolution. Indeed, $W_P$ derives from the vertical air velocity and thus describes rainfall associated with air motions at the considered scale.

Meanwhile the theoretical estimate of $W_P$ is based on the total observed rainfall and thus assesses cumulative gravitational power of precipitation at all scales. If $W_P$ derived from MERRA coincided with theoretical $W_P$, that would mean that no rainfall is associated with the air motions at a scale finer than 100 km and six hours. Since the scale of convection is of the order of a few kilometers or less, apparently some rain must remain unresolved by the larger-scale motions. Therefore, the fact that $W_P$ in MERRA is lower than its independent theoretical estimate does not indicate inconsistencies in the database. This note is included in Section 5.1, p. 15.

Second, the theoretical estimate in Appendix B illustrates how the various parameters entering the value of $W_P$ impact its magnitude. The bottomline however is provided by the TRMM-derived estimate of Pauluis and Dias [2012], which is 1.5 W $m^{-2}$ for the area between
* * *
[1]

$$W = -\frac{1}{S}\int_{\mathcal{V}} \mathbf{v} \cdot \nabla p \, d\mathcal{V} \equiv W_K + W_P, \tag{20}$$

$$W_K \equiv -\frac{1}{S}\int_{\mathcal{V}} (\mathbf{u} \cdot \nabla p) d\mathcal{V} + W_c \approx -\frac{1}{S}\int_{\mathcal{V}} \mathbf{u} \cdot \nabla p \, d\mathcal{V}, \quad W_c \equiv -\frac{1}{S}\int_{\mathcal{V}} \rho_c(\mathbf{w} \cdot \mathbf{g}) d\mathcal{V}, \tag{21}$$

$$W_P \equiv -\frac{1}{S}\int_{\mathcal{V}} \rho\mathbf{w} \cdot \mathbf{g} \, d\mathcal{V} = -\frac{1}{S}\int_{\mathcal{V}} gz\dot{\rho} \, d\mathcal{V} = Pg\mathcal{H}_P, \quad P \equiv -\frac{1}{S}\int_{z>0} \dot{\rho} \, d\mathcal{V}. \tag{22}$$

[Figure]

Figure c1: Long-term mean atmospheric power as dependent on temporal resolution: 6-hourly (solid curves), daily (dashed curves) and monthly (dotted curves). (a) total power $W$ (20), (b) kinetic power $W_K$ (21), (c) gravitational power of precipitation $W_P = W - W_K$. Black curves: MERRA; red curves: NCAR/NCEP.

30º N and 30º S. So, global $W_P$ cannot be lower than 0.75 W m$^{-2}$. If it is 0.75 W m$^{-2}$, this means that there is no precipitation at all in the extratropics. However, since extratropical precipitation is significant (2.2 mm day$^{-1}$ versus 3.1 mm day$^{-1}$ in the tropics, see Fig. 5 in our manuscript), it will contribute to the global value of $W_P$. Even we assume that all extratropical rainfall precipitates from $\mathcal{H}_P = 1$ km (which is clearly an underestimate), global $W_P$ will constitute 0.87 W m$^{-2}$. Therefore, the uncertainty of the lower limit of our estimate $W_P = 1$ W m$^{-2}$ is about 10%. This note is included in Appendix B, p. 27.

We also emphasize in the new Section C4 of Appendix C that the second approach to estimating $W_P$ (by extrapolation from the nearest pressure levels to the surface) produces an even lower estimate of $W_P$ suggesting that our conclusion about $W_P$ underestimated in MERRA is robust.

**1.2 Comment 2**

*2. Estimations of the total atmospheric power $W$ and $W_K$ are subject to similar uncertainties as mentioned in my comment # 1 above. At resolution of 1.25 degree and 42 vertical levels, any global estimation of the total integrated energy and kinetic energy contains large variation, let alone the difference between two. Have the authors tried the NCEP reanalysis or ECMWF dataset at different resolutions to see how sensitive your estimations are? As long as we don't have reliable estimation of $W, W_K$, and GPP, explanation for the difference would provide little scientific value.*

Response [see also doi:10.5194/acp-2016-203-AC6]:

We agree with the above comments and extended our analysis to include the NCAR/NCEP daily and monthly data for 1979-2015, as well as MERRA daily and monthly data. This yielded instructive results.

Two major conclusions emerged. First, the new data (Fig. c1) supported our original statement that estimated kinetic power $W_K$ should grow with better resolution until all convective motions are resolved. Our analyses suggest that in this limit $W_K$ should be about 4 W m$^{-2}$. This coincides with our previously published theoretical estimate of condensation-induced air circulation.

Second, we found that, unlike $W_K$, total power $W$ and the gravitational power of precipitation $W_P$ are not consistent across the re-analyses even if the zonal averages of local $W$ are similar (Fig. c2); we explore the reason of these discrepancies and we have now suggested

[Figure]

Figure c2: Long-term mean zonally averaged atmospheric power calculated from daily mean data for 1979-2015 in the MERRA versus NCAR/NCEP re-analysis as dependent on latitude (black solid curve: MERRA, red dashed curve: NCAR/NCEP). (a) $I_K \equiv - \int \mathbf{u} \cdot \nabla p dz$, (b) $I_\omega \equiv - \int omega dz$.

[Figure]

Figure c3: Trends in annual mean $W$, $W_P$ and $W_K$ derived from the 3-hourly instantaneous MERRA data.

how independent estimates of $W_P$ might improve future estimates.

We added three new figures (Fig. c1, Fig. c2 and Fig. c3). The latter figure shows that $W_P$ in MERRA is not correlated with global precipitation, which, according to recent analysis of Kang and Ahn [2015], rises in MERRA, while $W_P$, as we show, declines. We discuss how this may be an artefact of the correction procedures involved during the retrieval of the vertical velocities.

These new analyses are presented in the revised Sections 5.2 and 5.3.

**1.3  Comment 3**

*3. The derivation of the total atmospheric power given by Eq. (7)-(8) is unnecessarily complicated. I can directly obtain Eq. (7) from Eq. (2) by noting simply that $\int p dV/dt = \int p d(\delta x \delta y \delta z)/dt = \int p(\nabla \cdot v) dV$. Not sure why the authors present their argument in such a lengthy and confusing way.* The referee also noted in his general comments that *the presentation of this work is somewhat confusing, and can be simplified substantially to make it clearer.*

Following the suggestion of Referee 1, we unburdened Section 2 of the longer derivation of Eq. (7) from the continuity equation and the ideal gas law and derived the same result immediately from the consideration of the relative change of the air parcel's volume. However, this simpler derivation contains an implicit assumption, which necessitates our original longer derivation that uses the continuity equation and the equation of state for ideal gas (in the

revised text it is moved to the new Appendix A). Specifically, this derivation assumes that any volume change occurs at the expense of the divergence of velocity $\nabla \cdot \mathbf{v}$ defined at an arbitrary scale. Since phase transitions involve gas velocities that are scale-specific, the plausibility of this assumption for this case requires a discussion. It is presented in the revised Section 2.

**1.4    Comment 4**

*4. The authors criticize Laliberte et al. (2015)'s estimation of the integral of $dh/dt$, as they believe that it is not $dh/dt = 0$ but should be $\partial h/\partial t = 0$ for a stationary budget. However, my understanding of Laliberte et al.'s study is that the total derivative that Laliberte et al used is in the context of global integration. So if you define $H = \int h dV$, then $dH/dt = \int \partial h/\partial t dV$, since the total volume is fixed in time. As such, Laliberte et al.'s global stationary approximation is consistent with your local stationary approximation.*

Response [see also doi:10.5194/acp-2016-203-AC5]:

Laliberté et al. [2015] aim to estimate the global mean value of atmospheric power $-\alpha dp/dt$. They cannot therefore follow the above described procedure integrating the first law of thermodynamics first over mass $\mathcal{M}$, $d\mathcal{M} = \rho d\mathcal{V}$, and then taking its derivative over time. This procedure for $-\alpha dp/dt$ would yield $- \int_{\mathcal{V}} \partial p/\partial t d\mathcal{V} = 0$.

Indeed, Laliberté et al. [2015] explicitly define $dh/dt$ as the material derivative of enthalpy [see p. 540, middle column, 7th line from top], not the partial derivative over time. They state that they average the first law of thermodynamics taking the mass-weighted annual and spatial mean of all the terms in the equation, including $dh/dt$ [p. 540, middle column, 7th line from bottom]. They denoted this mean as $\{\cdot\}$. The mass-weighted spatial mean of the material derivative of $h$, which is enthalpy per unit wet air mass, consists in taking its integral over total atmospheric mass and then dividing by the planet surface area. This means that stating that $\{dh/dt\} = 0$ Laliberté et al. [2015] meant $I_h \equiv (1/\mathcal{S}) \int_{\mathcal{M}} dh/dt d\mathcal{M} = 0$ and not $\partial(\int_{\mathcal{M}} h d\mathcal{M})/\partial t = 0$.

We also note that, to support their statement that the expression for total atmospheric power does not contain the enthalpy term, Laliberté et al. [2015] refer to Eq. 4 of Pauluis [2011] [p. 540, right column, 12th line from top]. This link does not recognize that Eq. 4 of Pauluis [2011] [ref. 10 of Laliberté et al. [2015]] refers to atmospheric power defined *per unit dry air mass*. As we note in the revised text (see Eqs. 33 and 34), the material derivative of any variable integrated over *total mass of atmospheric dry air* is zero (because of zero sources or sinks of dry air). In contrast, the material derivative of any variable integrated over *total atmospheric mass* is in the general case not zero, because of the non-zero sources and sinks in the continuity equation. This point, which follows from the previous derivations in the paper, is essential for understanding the atmospheric power budget and also for estimating it.

**1.5    Minor concern**

*The practice of putting a dot on a variable to represent sources/sinks is too confusing, as the dot often denotes time derivation. Equation such $\dot{N} = dN/dt + N(\nabla \cdot v)$ is perplexing. The authors should replace all such dotted source/sink by different symbols to avoid the confusion.*

Response [see also doi:10.5194/acp-2016-203-AC1, footnote 2]:

We agree with this comment, but ultimately chose to retain the dot over sources/sinks to ensure consistency in notations with the study of Laliberté et al. [2015], which we examine in detail.

**2 Referee 2 [doi:10.5194/acp-2016-203-RC2]**

**2.1 Comment 1**

*This manuscript looks at the power budget in the MERRA reanalysis over the last 7 years. It is generally poorly written and way too long for the arguments being made. In its current state, it does not stand up as a contribution worthy of the high standards of publication for ACP. Based on my comments (to be found below), I do not recommend this manuscript for publication at ACP.*

*Key Comments:*

*1. Section 2 is both way too complicated and appears to be wrong. Following Vallis' (2006) notation:*

$$W = \int_{\mathcal{V}} p \frac{d\alpha}{dt} \rho d\mathcal{V} = \int_{\mathcal{V}} p(\partial_t(\rho\alpha) + \nabla \cdot (\rho\alpha\mathbf{v}) - \alpha S_\rho)d\mathcal{V},$$

*where $S_\rho = \partial_t(\rho) + \nabla \cdot (\rho\mathbf{v})$ is the local sources and sinks of mass. Now, $\alpha\rho = 1$ so*

$$W = \int_{\mathcal{V}} p(\nabla \cdot (\mathbf{v}) - \alpha S_\rho)d\mathcal{V} = \int_{\mathcal{V}} (\nabla \cdot (p\mathbf{v}) - \mathbf{v} \cdot \nabla p - \alpha p S_\rho)d\mathcal{V}.$$

*This is the same form as in equation (8). But it depends explicitly on $S_\rho$, contrary to the authors' claim. Why this contradiction? The problem in the authors' derivation comes in part from equation (3). While it is true that $\sum_i d\tilde{N}_i/dt = 0$, it is not true that $\sum_i T_i d\tilde{N}_i/dt = 0$, unless the atmosphere is isothermal. But it is exactly what's used to convert the last term in equation (2) to the last term in equation (4). $\tilde{V} = \tilde{N}/N$ has units of $m^3$ (parcel$^{-1}$). To compute work, however, we need the specific volume with units of $m^3$ (kg$^{-1}$). So we have to introduce a new quantity, the mass per parcel $\tilde{m}$ so that the specific volume is $\tilde{V}/\tilde{m}$. Then the expression for work (equation 4) with the same units as in Vallis (2006) reads:*

$$W = \frac{1}{\mathcal{S}} \int_{\mathcal{V}} p \frac{\tilde{m}}{\tilde{V}} \frac{d(\tilde{V}/\tilde{m})}{dt} d\mathcal{V}.$$

*But the continuity equation (6) also requires fixing. Since, $N$ has units of mol m$^{-3}$ then equation (6) is an equation for mass conservation only if the molar mass $\tilde{m}/\tilde{N}$ is constant. But here the authors are, among other things, concerned about the effect of moisture on the work and moist air, unlike dry air, has an inhomogeneous in molar mass. The continuity equation (6) should then read:*

$$\partial_t((\tilde{m}/\tilde{V})) + \nabla \cdot (\mathbf{v}(\tilde{m}/\tilde{V})) = \dot{\tilde{m}}(N/\tilde{N}) + \tilde{m}/\tilde{N}\dot{N} - (\tilde{m}N/\tilde{N}^2)\dot{\tilde{N}}$$

*where the right hand side is the local sources and sinks of mass. With these fixes, the expression for work will look exactly like in Vallis (2006) and will depend on the sources and sinks of mass.*

Response [see also doi:10.5194/acp-2016-203-AC3]:

In our revised work we consider the entire MERRA data span from 1979 to 2015 at several temporal resolutions; plus additionally NCAR/NCEP data for the same period.

The discrepancy between our Eq. (7) and the referee's derivation results from the incorrect definition of work per unit mass. In the presence of phase transitions it is not $pd\alpha$ as clarified in our revised Section 2, see Eq. 11. Indeed, consider an air parcel of mass $\tilde{m}$ and volume $\tilde{V}$, such that $\alpha \equiv \tilde{V}/\tilde{m}$. Then work of this parcel is $pd\tilde{V}$, while work per unit mass is $(1/\tilde{m})pd\tilde{V} = (1/\tilde{m})pd(\alpha\tilde{m}) = pd\alpha + p\alpha d\tilde{m}/\tilde{m} \neq pd\alpha$. Therefore, $W \neq W_{IV} \equiv (1/\mathcal{S}) \int_{\mathcal{V}} p(d\alpha/dt)\rho d\mathcal{V}$.

We also note that in our derivation we did not assume either $\sum_i d\tilde{N}_i/dt = 0$ or $\sum_i RT d\tilde{N}_i/dt = 0$. This misunderstanding might have arisen because the derivation was presented in a very compact form. The revised more detailed text (new Appendix A) makes it clear that the resulting expression for work does not depend on the temperature term discussed by the referee.

We have striven to present our arguments as succinctly as possible in the revised manuscript. But the need to identify and discuss the various inconsistencies that surround the topic we examine stands in the way of a radical shortening.

**2.2 Comment 2**

*2. Section 3.1. This section is also way too complicated. After the first paragraph, one can jump directly to the top of page 5. Now equation (15) is not wrong per se. However, the Makarieva et al. (2013) analytical derivation is somewhat meaningless when applied to re-analysed data: this can be evaluated directly. This is exactly what I have done for the purpose of this review. Using the 1 hourly vertically integrated budgets provided from the data archive, one can compute the integral $\int_{\mathcal{S}} \overline{h}\overline{\dot{\rho}}d\mathcal{S}$, where the overline indicates vertically integrated fields. In the reanalysis, $\overline{\dot{\rho}} \neq 0$ because of the analysis step. In MERRA, this is provided directly. In the MERRA documentation it is indicated that this $\overline{\dot{\rho}}$ includes both the effect of $E - P$ and adjustments needed to represent the observed surface pressure field accurately. It therefore includes the effect described by the authors. This quantity for 1980-1985 is 0.2 $W/m^2$. Adding the vertical dependence would likely be a second order effect since $E - P$ is mostly driven by horizontal and temporal variability. This simple analysis performed using the output from the MERRA product seems to show that Appendix A is likely to be inaccurate (0.2 is not within 30% of 1.6). In any case, this issue was discussed at length by Trenberth (see his papers in the 1990's) and the proposed solution is to modify the winds so that the continuity equation does not have a source term. I had a hard time finding this but you mention that Laliberte et al (2015) might have done something like this. In this case, I do believe that $\int_{\mathcal{M}} dh/dt d\mathcal{M} = 0$ makes sense since it is an exact derivative.*

Response [see also doi:10.5194/acp-2016-203-AC2]:

We thank Referee 2 for their effort to numerically check our results. However, as we show below, the estimate obtained by Referee 2 appears to result from a misunderstanding (a confusion of the mean of a product $\overline{f_1 \cdot f_2}$ for the product of means $\overline{f_1} \cdot \overline{f_2}$, which is fatal when $\overline{f_1} = 0$). As such, this estimate neither disproves our theoretical result nor justifies the omission of $I_h$ by Laliberté et al. 2015. As we clarify below, we have demonstrated in our work that $I_h$ is not proportional to the vertical integral of the source term $\int \dot{\rho}dz$ and does not vanish when the latter is zero.

We presume that the referee's agreement with our Eq. (15) pertains to the equality

$$I_h \equiv \int_{\mathcal{M}} \frac{dh}{dt} d\mathcal{M} = -\int_{\mathcal{V}} h\dot{\rho}d\mathcal{V} \equiv -A. \tag{c1}$$

The referee proposes to estimate $A$ as

$$A \approx B \equiv \int_{\mathcal{S}} \widehat{h}\widehat{\dot{\rho}}d\mathcal{S}. \tag{c2}$$

suggesting that $\widehat{h}$ and $\widehat{\dot{\rho}}$ are available from the MERRA dataset (we replaced the overline by $\widehat{\phantom{x}}$ in $B$ to preserve the overline for the averages to appear below).

We need first to resolve an inconsistency between the units of our $A$ and the referee's $B$. First, we note that the dot over enthalpy $h$ in $B$ may be a misprint since an *enthalpy source* $\dot{h}$ appears to be an unspecified variable out of context. Next, if following the referee's indication that $\widehat{\phantom{x}}$ in $B$ denotes *vertically integrated fields* we assume that $\widehat{h} \equiv \int h dz$ and $\widehat{\dot{\rho}} \equiv \int \dot{\rho}dz$, then $B$ has the units of [J s$^{-1}$m], while $A$ has the units of [J s$^{-1}$]. So expression $B$ needs some "fix" before it could be compared with $A$.

Keeping $\widehat{\dot{\rho}} \equiv \int \dot{\rho}dz$, the only way we can see to remedy $B$ is to assume that $\widehat{h} \equiv \int h dz / \int dz$, units [J kg$^{-1}$] is the mean enthalpy in the air column (not the vertically integrated enthalpy [J kg$^{-1}$ m]). In this case the units of $A$ and $B$ coincide and what the referee proposes reads

$$A \approx \int_{\mathcal{S}} \left( \frac{\int h dz}{\int dz} \int \dot{\rho}dz \right) d\mathcal{S}. \tag{c3}$$

Noting that $d\mathcal{V} = dzd\mathcal{S}$, this implies the following replacement in $A$

$$\int h\dot{\rho}dz \approx \frac{\int hdz}{\int dz}\int \dot{\rho}dz. \tag{c4}$$

By dividing both parts of (c4) by $\int dz$ we find that (c4) relates the columnar mean of $h\dot{\rho}$ to the product of columnar means of $h$ and $\dot{\rho}$. The two expressions are not equivalent, since, as is well-known:

$$\overline{h\dot{\rho}} = \overline{h}\cdot\overline{\dot{\rho}} + \overline{(h-\overline{h})(\dot{\rho}-\overline{\dot{\rho}})}, \tag{c5}$$

where $\overline{X} \equiv \int Xdz/\int dz$. The second term in the right-hand part of (c5) represents the covariance of the two variables. Indeed, we know that the enthalpy and the rate of phase transitions in the atmosphere are spatially correlated: $h$ is higher at the surface where evaporation occurs and $\dot{\rho} > 0$ and lower in the upper atmosphere where condensation occurs and $\dot{\rho} < 0$. Therefore, $\overline{(h-\overline{h})(\dot{\rho}-\overline{\dot{\rho}})}$ in (c5) is not zero.

When, as proposed by the referee, $\int \dot{\rho}dz \to 0$ and $\overline{\dot{\rho}} \to 0$, the first term in (c5) disappears. The relative error of estimating $\overline{h\dot{\rho}} \neq 0$ by $\overline{h}\cdot\overline{\dot{\rho}}$ tends to infinity. For this reason $B$ carries no information about the real value of $A$ and, hence, $I_h$ (c1).

Note also that since the enthalpy of an ideal gas is defined to the accuracy of an arbitrary constant, the absolute magnitude of $\overline{h}\cdot\overline{\dot{\rho}}$ for $\overline{\dot{\rho}} \neq 0$ does not have any physical meaning as it explicitly depends on that constant. The second term in the right-hand part of (c5) is constant-invariant.

In our work we have estimated $I_h$ assuming that evaporation and condensation are localized at, respectively, the surface $z = 0$ and the mean condensation height $z = H_P$. This approximation allows one to explicitly specify $\dot{\rho}$ via the Dirac delta function

$$\dot{\rho} = E(x,y)\delta(z) - P(x,y)\delta(z-H_p), \quad \int \dot{\rho}dz = E(x,y) - P(x,y), \tag{c6}$$

from which $I_h$ can be explicitly evaluated.

Putting $E(x,y) = P(x,y)$ in Eq. (15), such that $\int \dot{\rho}dz = E(x,y) - P(x,y) = 0$, one obtains from our Eq. (15) that the integral $I_h$ is proportional not to the (zero) difference between evaporation and precipitation, but, as one might have expected, to the intensity of the water cycle, i.e. to $E(x,y) = P(x,y)$ multiplied by the difference in air enthalpy between $z = 0$ and $z = H_P$. (This clarification is added to the revised manuscript, see Section 4.). Since no global observational data exist on the local values of $\dot{\rho}$, our theoretical estimate is currently the only available estimate of $I_h$ (c1).

In revised Section 5.2 we discuss the correction procedure proposed by Trenberth [1991]. Strictly speaking, this procedure does not modify the winds such that the continuity equation does not have sources or sinks. Rather, to achieve mass conservation this correction does take the non-zero sources and sinks into account, but only in the vertically integrated form, since local values of $\dot{\rho}$ are unknown. We suggest that this procedure might be responsible for the physically unreasonable seasonal cycle and multiyear trend of $W_P$ in MERRA, whereby $W_P$ is uncorrelated with precipitation.

**2.3   Comment 3**

*3. Computing the work from MERRA data. As mentioned before, the MERRA product has many vertically integrated budget variables that allow one to quantify each one of the term in the energy equation. For this review, I've looked at the kinetic energy generation 1980-1985 and the yearly average gives 3.40-3.48 $W/m^2$ for the integral of $\omega\alpha$ and 3.6-3.8 $W/m^2$ when including the kinetic energy generation from the analysis step. The kinetic energy generation is balanced by damping from the numerical dissipation, the dynamical remapping and the physically parametrized frictional dissipation. This means that the estimates provided in section 5.1 are substantial underestimates.*

Response [see also doi:10.5194/acp-2016-203-AC6]:

We agree with the referee that our estimates of $W$ are substantial underestimates of the real atmospheric power; indeed, it is our major point. We do not consider the kinetic energy generation from the analysis step. (Neither did Laliberté et al. [2015]). We explicitly address how the global atmospheric power can be estimated using the re-analysis pressure and velocity at their face value, based on our Eqs. (20)-(22). In our revised manuscript we present an analysis of the entire period 1979-2015. We note that for $W_K$ our results practically coincide with those of Huang and McElroy [2015], who reported $W_K = 2.46$ W m$^{-2}$ for 1979-2010. Our calculations for the same period give $W_K = 2.45$ W m$^{-2}$.

Our annual estimates of $W$ for 1980-1985 range from 3.20 to 3.28 W m$^{-2}$. This is 6% smaller than the referee's. The discrepancy may stem from two sources. First, the dataset with the vertically integrated $\omega\alpha$ derives from the 1-hourly surface dataset (presumably MAT1NXINT [tavg1_2d_int_Nx]), while our estimate derives from the 3-hourly dataset (MAI3CPASM [inst3_3d_asm_Cp]). As we have shown that $W$ increases with finer temporal resolution, see Fig. c1a, this may explain the 6% discrepancy. Second, the discrepancy may stem from a difference in the boundary condition for $\omega$ at the surface.

It is not explicitly indicated in MAT1NXINT how the integration was performed. We have described in detail how we treated the surface layer to make our analysis tractable and comparable to other studies. Furthermore, we investigated the impact of the surface boundary condition on our analysis for each variable and showed that the associated uncertainty is about 6%.

For $W$ and $W_K$ we estimated the value of $\omega$ and $\mathbf{u} \cdot \nabla p$ at the surface in two ways (see Appendix C in our revised manuscript). One is to assume that air velocity at the surface is zero, $\mathbf{v} = 0$, another is to linearly extrapolate $\omega$ and $\mathbf{u} \cdot \nabla p$ from the nearest pressure level to the surface. Our increased attention to the boundary layer is justified by the fact that horizontal velocity experiences significant non-uniform changes along the vertical. In the limit of an infinitely precise vertical resolution the two approaches should give the same value. In the real atmosphere they produce somewhat different results.

Specifically, the extrapolated $W_K$ turns out to be higher than $W_K$ calculated assuming $\mathbf{v} = 0$. This has to do with the vertical profile of $W_K$ shown in Fig. c4. Kinetic energy generation grows with increasing pressure in the lower atmosphere. Extrapolation of this dependence to the surface yields a positive surface value for kinetic energy generation. Thus, $W_K$ obtained from this extrapolation is higher than when we assume that $\mathbf{v} = 0$, such that no kinetic energy is generated at $z = 0$

In contrast, the estimate of total power $W$ is smaller when extrapolated than when assuming zero velocity at the surface. This has to do with a different distribution of pressure velocity over pressure levels, Fig. c4. Here the lowest layer between 975 hPa and the surface makes a large negative contribution to the total $W$. This is because the air predominantly descends in the regions of higher surface pressure. Therefore with one and the same $\omega$ at 975 hPa, the layer where the air descends and surface pressure is about, say, 1020 hPa is thicker than where the air ascends and surface pressure is about 1000 hPa. Since $W$ is proportional to $-\omega$, in the result the net contribution of the lower layer to global $W$ is negative.

The difference between the two estimates for $W$ and for $W_K$ is about 10%. The difference between $W_P$ values obtained by the two means is greater. $W_P$ obtained by interpolation is considerably smaller than $W_P$ obtained assuming that zero velocity at the surface. This suggests that our conclusion about $W_P$ being underestimated in MERRA is robust.

**2.4   Comment 4**

*4. In section 5.1, I do not see the use for $W_1$. And why not use $\omega_s = \partial_t p_s + \mathbf{v}_s \cdot \nabla_H p_s$, with $\nabla_H$ being the horizontal gradient? The $p_s$ and $\mathbf{v}_s$ are both available and this is the right expression. Maybe that could fix their underestimate of $W$.*

[Figure]

Figure c4: This figure was added to the revised text, see the new subsection C3 in Appendix C. Atmospheric power within the 41 pressure layers enclosed by the 42 pressure levels in the MERRA dataset MAI3CPASM in 1979-2015. Each bar of the histogram contains the contribution from the corresponding pressure layer $(p_i, p_{i+1})$, where $i$ is pressure level number, plus the contribution from layer $(p_s, p_i)$ if $p_i \leq p_s$ in the considered cell is the pressure level nearest to the surface. For example, the lowest bar of the histograms corresponds to the layer with pressure less than 975 hPa (i.e. the layer from $p_1 = 1000$ hPa to $p_2 = 975$ hPa plus the layer from $p_s$ to $p_1$). Sum of the histogram values over all layers gives the global values of $W$ and $W_K$. Subscripts 1 and 2 refer to the two ways of estimating $W$ and $W_K$, see Table 2 for details.

Response [see also doi:10.5194/acp-2016-203-AC6]:

The use of $W_1$ has been discussed in our reply to Comment 3 of Referee 2. $W_1$ and $W_{K1}$ were used to investigate the uncertainty associated with insufficient data resolution in the boundary layer.

One cannot use $\omega_s = \partial_t p_s + \mathbf{v}_s \cdot \nabla_H p_s$, because the horizontal gradient of surface pressure $\nabla_H p_s$ only exists if the surface is horizontal (i.e. has invariant geopotential height). Since the geopotential height of the real surface varies, surface pressure is much more affected by this variability in the vertical plane than by any effects in the horizontal plane, which prevents the use of $p_s$ for a reliable determination of $\omega_s$.

Furthermore, since the term $\mathbf{v}_s \cdot \nabla_H p_s$ is present in the surface values of both $\omega$ and $\mathbf{u} \cdot \nabla p$, even if this term were added, this would not change the difference between the global $W$ and $W_K$.

To simplify the presentation, in the revised text we everywhere use $W_2$ and $W_{K2}$ and discuss $W_1$ and $W_{K1}$ only in a separate Section C3 in Appendix C.

**2.5 Comment 5**

*5. The way I see it, there are approximately three manuscripts in this study. The first one, sections 2 and 3 as well as Appendix A, consist mostly of derivations that are either flawed or mostly useless for this study. The second paper is more akin to a white paper and comprises*

*sections 6.1 and 6.2. Now, sections 1, 4, 5 and the very beginning of section 6 as well as Appendix B and C are self-contained and describe an original treatment of reanalysis data. Appendix C could be moved up after section 4. If they wish to submit their results to another publication, I would recommend that the authors focus on these sections and perform their analysis on the whole of MERRA (1979-2015).*

Response:

We thank the referee for this kind word about our data analyses. We followed their recommendation to analyze the whole of MERRA (1979-2015) in the revised text, see Section 5. However, as we argued in our response to Comments 1 and 2 of the referee, we disagree that Sections 2 and 3 (now 2 and 4) are not relevant to these data analyses. We find that the literature on our topic, the definition and estimates of global atmospheric power, lacks clarity (see, for example, our Response to Comment 2 of Referee 4 below). Before setting out to analyze numerical data, it is essential to define on clear physical grounds what we are going to measure and constrain. In the revised text we overview the various formulations for global atmospheric power and show why and how they differ and which can be applied to a moist atmosphere (see Introduction, especially Eqs. (1)-(4), and Sections 2 and 3).

We appreciate the referee's mentioning of Appendix C (now D), as we believe that it does indeed contain some interesting results. In particular, in Fig. 10 in the revised manuscript (see Fig. c5) we show that the integral of pressure tendency $\Psi \equiv \frac{1}{\mathcal{S}} \int_{\mathcal{V}} \frac{\partial p}{\partial t} d\mathcal{V}$ is comparable to global atmospheric power $W$ on a seasonal scale and that the formulation of the continuity equation with use of hydrostatic equilibrium prevent a consistent account of this term. If we formally add $\Psi$ to $\Omega \equiv -\frac{1}{\mathcal{S}} \int_{\mathcal{V}} \omega d\mathcal{V}$ (which should give $W$) we obtain an unreasonable result that total atmospheric power $W$ during certain months is smaller than kinetic power $W_K$.

It is also notable that $\Psi$ can be estimated using a very simple formula for a periodically warming and cooling hydrostatic atmosphere from the observed rate of global temperature change.

**2.6 Comment 6**

*6. Finally, I'm not sure the following sentence is logically true: "The fact that $W_{Kc}$ is likewise higher than our MERRA-derived kinetic power, testifies in favor of the theoretical estimate". All it means is that $W_{Kc}$ is potentially a right upper bound. The only way to check whether it is the right upper bound would be to either verify if it holds on other Earth-like planets or using simulations with increasing resolution and seeing that it describes the scaling. As I said before, the last two sections of this manuscript are really too speculative in their current form and they are dragging down the original results described in sections 5.*

Response [see also doi:10.5194/acp-2016-203-AC6]:

We explained in our original manuscript in what sense we consider our results supportive of our theoretical estimate $W_{Kc}$: since we expect that kinetic power $W_K$ should grow with better resolution, it is a good news for $W_{Kc}$ that is is higher than $W_K$ observed at current resolution. If, instead, $W_K$ were higher than $W_{Kc}$, that would testify against $W_{Kc}$.

In our revised manuscript we attempted to estimate how $W_K$ changes with temporal resolution by analyzing additionally daily and monthly mean MERRA and NCAR/NCEP data for $W_K$ (see Section 5.3 and Fig. c1 above). These results indicate that $W_{Kc}$ does indeed represent a plausible upper limit for the kinetic power of convective motions resolved at the scale of about 1 hour. But we do agree with the referee that further analyses are needed to improve reliability of this result.

Section 6, using the results obtained in the previous sections, shows that condensation-driven circulation corresponds to a Carnot cycle with a temperature difference $\Delta T$ coinciding with the mean temperature difference between evaporating and condensing water vapor. We believe that this new result is quite specific.

This work evolved from a short technical comment that we made on the work of Laliberté et al.(2015) in February 2015. This comment and the review we received from Science is

[Figure]

Figure c5: Time series (30-day running mean of daily values for the year 2010) of (a) the global integral of the pressure tendency $\Psi$, the omega integral $\Omega$ and kinetic power $W_K$ (21); global mean surface temperature (b), global mean surface pressure (c) and global mean geopotential height at $p_T = 0.1$ hPa (d). This pressure level moves with vertical velocity $w_T$ of about 300 m in half a year, $w_T \sim 2 \times 10^{-5}$ m s$^{-1}$, which corresponds to $I_T \sim p_T w_T \sim 10^{-4}$ W m$^{-2} \ll W$. Ticks on the horizontal axes correspond to the 15th day of each month.

available from http://www.bioticregulation.ru/ab.php?id=he. In particular, one referee of this short comment refuted our suggestion that air circulation on Earth can be powered by condensation by noting that the models of a dry atmosphere display the same atmospheric power as does the real atmosphere – hence no need for alternative drivers. Assessments of our work by other anonymous colleagues showed that this idea is common. Thus, in Section 6 we explain why models of dry atmospheres cannot indicate whether or not global atmospheric circulation is condensation-driven.

**3 Referee 3 [doi:10.5194/acp-2016-203-RC3]**

**3.1 Comment 1**

*The manuscript is poorly written and requires substantial improvement before publication. The authors misrepresent part of their results as a new analysis, while they have been previously discussed in the literature.*
    *Main comments:*
    *1. Appropriation in the main result:*
    *The manuscript states pretty explicitly that the main contribution here is*
    *"Starting from the definition of mechanical work for an ideal gas, we present a novel*

*derivation linking global wind power to measurable atmospheric parameters. The result-ing expression distinguishes three components: the kinetic power associated with horizontal motion, the kinetic power associated with vertical motion and the gravitational power of precipitation."*

*as it is stated in the abstract. This claim is repeated on multiple occasions. I assume that this specifically refer to the equation (20-22), which the authors claim that "Equations Eqs. (20)-(22) and their derivation have not been previously published."*

*These equations are presented in Pauluis etal. (2000) (See equations (2), (4), (8) and (10). See also equations (4) and equation (6) of Pauluis and Held (JAS, 2002)). It is very troublesome that the authors fail to mention that equations (20-22) are presented in Pauluis etal. (2000) despite the fact that this pa*

*The appropriation is not limited to the equations, but extends to some of the arguments presented. For instance, the authors relate the claim*

*"The meaning is that hydrometeors perform work at the expense of their potential energy. To acquire this energy, a corresponding amount of water vapor must be raised by air parcels. We can also see that WP does not depend on the interaction between the air and the falling hydrometeors. This term would be present in the atmospheric power budget even if hydrometeors were experiencing free fall and did not interact with the air at all (such that no frictional dissipation on hydrometeors occurred)."*

*This points is made previously ( and more clearly) in Pauluis etal. JAS (2000, p. 991):*

*"The dissipation by precipitation can be thought as proceeding in two steps. First, water is lifted by the atmospheric circulation, increasing its potential energy. Then, during precip-itation, the potential energy of condensed water is transferred to the ambient air where it is dissipated by molecular viscosity in the microscopic shear zone around the hydrometeors."*

*To put it bluntly, the authors are presenting as their own an analysis that was done by others, and in doing so, are misleading their reader.*

Response [see also doi:10.5194/acp-2016-203-AC4]:

Presumably there is some misunderstanding involved so we have revised the text clarifying how our results relate to previous work. In particular, we now show that Eqs. 20-22 could not *in principle* be formulated by Pauluis et al. 2000, because their basic assumptions are not consistent with either Eqs. 20-22 or with Eq. 4 of Pauluis and Held 2002. We acknowledge the value in making this claim clear and explicit as it is precisely because Eqs. 20-22 were not published previously that the global gravitational power of precipitation $W_P$ could also not be estimated from re-analyses until now.

We revised the text having added a separate "Section 3.3 Our results compared to Pauluis et al. 2000". Right below Eqs. (20)-(22) we explain why in our view these equations are original. Furthermore, we also explicitly refer the readers to Section 3.3 where these results are compared with Pauluis et al. 2000 by noting: "Equations (20)-(22) and their derivation have not been previously published (see the next section)." Readers can judge our claims for themselves. Reference to Pauluis et al. 2000 is also made already in the revised Introduction: "In Section 3 we discuss how global atmospheric power can be represented as a sum of three distinct physical components. Two components dominate in the atmosphere of Earth: the kinetic power of the wind generated by horizontal pressure gradients and the gravitational power of precipitation generated by the ascending air. We compare our results with the previous assessments of the atmospheric power budget by Pauluis et al. [2000].".

As a separate point, we note that Eqs. (20)-(22) make it clear that $W_P$ can be estimated from the data on air velocity and pressure gradient with no information required about moist processes. As can easily be verified by examining the texts in question, this message is absent from the works cited by the referee (or indeed in any previous publications of which we are aware). To facilitate this comparison we list the equations mentioned by the referee below together with our Eqs. 20-22 from the submitted manuscript.

Pauluis et al (2000), Eqs. (2), (4), (8) and (10), respectively:

$$W_p = \int_\Omega g\rho_c v_{\mathrm{T}}, \qquad (c7)$$

$$W_p = \int_\Omega g\rho_t w, \qquad (c8)$$

$$W_D = \int \bar{\rho} g w \left[ \frac{\Theta'}{\overline{\Theta}} + \left( \frac{R_v}{R_d} - 1 \right) \frac{\rho_v}{\bar{\rho}} - \frac{\rho_c}{\bar{\rho}} \right], \qquad (c9)$$

$$W_{\mathrm{tot}} = \int w g \left[ \bar{\rho} \frac{\Theta'}{\overline{\Theta}} + \rho_v \frac{R_v}{R_d} \right], \qquad (c10)$$

where $\rho_t = \rho_c + \rho_v$.

Pauluis and Held (2002), Eqs. (4) and (6), respectively:

$$W = \int_\Omega p\partial_i V_i, \qquad (c11)$$

$$D_p = \int_\Omega g\rho_c V_T = \int_\Omega \rho q_t g w, \qquad (c12)$$

where $V_i$ is the $i$th component of the velocity, $\partial_i = \partial/\partial x_i$ is the partial derivative in the $i$ direction, $\rho_c$ is the mass of falling hydrometeors per unit volume, $q_t$ is mass of total water per unit mass of moist air, $V_T$ is the terminal velocity of the falling hydrometeors, and $w$ is the vertical velocity of the air.

Equations (20)-(22):

$$W = -\frac{1}{\mathcal{S}} \int_\mathcal{V} \mathbf{v} \cdot \nabla p d\mathcal{V} \equiv W_K + W_P, \qquad (c13)$$

$$W_K \equiv -\frac{1}{\mathcal{S}} \int_\mathcal{V} (\mathbf{u} \cdot \nabla p) d\mathcal{V} + W_c \approx -\frac{1}{\mathcal{S}} \int_\mathcal{V} \mathbf{u} \cdot \nabla p d\mathcal{V}, \quad W_c \equiv -\frac{1}{\mathcal{S}} \int_\mathcal{V} \rho_c (\mathbf{w} \cdot \mathbf{g}) d\mathcal{V}, \qquad (c14)$$

$$W_P \equiv -\frac{1}{\mathcal{S}} \int_\mathcal{V} \rho \mathbf{w} \cdot \mathbf{g} d\mathcal{V} = -\frac{1}{\mathcal{S}} \int_\mathcal{V} g z \dot{\rho} d\mathcal{V} = P g \mathcal{H}_P, \quad P \equiv -\frac{1}{\mathcal{S}} \int_{z>0} \dot{\rho} d\mathcal{V}. \qquad (c15)$$

Note that $\rho = \rho_d + \rho_v \neq \rho q_t$; $\mathbf{v} = \mathbf{u} + \mathbf{w}$ is air velocity (horizontal and vertical).

**3.2 Comment 2**

*2. Discussion of Laliberte etal. (2015)*

*The discussion of Laliberte etal. (2015) is very esoteric and does not pertain much to the rest of the discussion. Section 3.2 is a very minor point. It is fairly well-known that the integral of dp/dt is only equal to the work performed for a steady system, an assumption that is clearly stated in Laliberte etal.*

Response [see also doi:10.5194/acp-2016-203-AC5]:

We agree with the referee that Laliberte et al. assume a steady atmosphere. Section 3.2 did not mention Laliberte et al. and did not question their steady state assumption. This section drew attention to the $\partial p/\partial t$ term and made a reference to Appendix C (now D) where it is shown that this term may be considerable on a seasonal scale thus influencing estimates of global atmospheric power, see Fig. 10a in the revised manuscript or Fig. c5a above. As discussed later in the paper (see revised Section 5.2), this fact can account for the discrepancy between the seasonal changes of global mean precipitation $P$ and $W_P$ derived from mean atmospheric $dp/dt$. To shorten the presentation, we removed Section 3.2 from the revised paper, as all the necessary information is contained in Appendix D.

The referee continues: *As for section 3.1, there are several problems with the authors analysis. First, it should be clearly stated that the global integral of dh/dt is indeed zero in the absence of mass source and sink in the continuity equation.*

Response:

We see no problem here, as this statement immediately follows from the obtained expression for $I_h$. We have included the suggested statement in the revised text, see line 28 on p. 11.

The referee continues: *First, it should be clearly stated that the global integral of dh/dt is indeed zero in the absence of mass source and sink in the continuity equation. This is the assumption made in Laliberte etal. It is also the continuity equation used in the MERRA Reanalysis. Hence, the authors should explicitly acknowledge that the claim that the integral of dh/dt is indeed correct within the assumptions made in the MERRA Reanalysis.*

Response:

The absence of mass source and sink in the continuity equation is equivalent to the absence of a water cycle. Laliberté et al. [2015] focus was on thermodynamic aspects of the atmospheric water cycle. They could not and did not assume the absence of mass source and sink in the continuity equation.

Specifically, on p. 2 in their Supplementary Materials, Laliberté et al. [2015] state: "In the atmosphere, the moist entropy $s$ and the specific humidity $q_T$ satisfy $\partial_t s + v \cdot \nabla s = \dot{s}$ and $\partial_t q_T + v \cdot \nabla q_T = \dot{q_T}$, where $\dot{s}$ and $\dot{q_T}$ are their respective sources and sinks." (Note that the latter equation is equivalent to $dq_T/dt = \dot{q_T}$.)

To make it clear that this statement is incompatible with the assumption of "absent sources and sinks in the continuity equation", we consider the continuity equation for air as a whole

$$\frac{\partial \rho}{\partial t} + \nabla \cdot (\rho \mathbf{v}) = \dot{\rho} \tag{c16}$$

together with the continuity equation for water vapor

$$\frac{\partial \rho_v}{\partial t} + \nabla \cdot (\rho_v \mathbf{v}) = \dot{\rho}. \tag{c17}$$

Noting that $q_T \equiv \rho_v/\rho$ (Laliberté et al. [2015] neglect the tiny condensate content) we find from Eqs. (c16) and (c17) that

$$\dot{q_T} = \frac{\dot{\rho}}{\rho}\left(1 - \frac{\rho_v}{\rho}\right). \tag{c18}$$

Thus, if Laliberté et al. [2015] had assumed $\dot{\rho} = 0$, they would have omitted not only the enthalpy term in their first law of thermodynamics but also the term proportional to $dq_T/dt = \dot{q_T}$. The latter term was the focus of their analysis though. Thus, the referee's suggestion that Laliberté et al. [2015] assumed $\dot{\rho} = 0$ is not valid.

Neither is this assumption made in the MERRA database. What can be assumed in the MERRA database and could also be assumed by Laliberté et al. [2015] (although we see no grounds for such an assumption), is that the vertically integrated continuity equation has negligible sources or sinks, that is $\int \dot{\rho} dz \approx 0$. However, as we discussed in detail in a previous comment [doi:10.5194/acp-2016-203-AC2], this relationship does make $I_h$ equal to zero. As we discuss in revised Section 5.2, the barotropic correction to wind velocities used by Laliberté et al. [2015] is not equivalent to setting the sources and sinks equal to zero.

The referee continues: *Second, it is perfectly valid to question the impact of mass source and sink on the framework of Laliberté et al., but this should be done clearly. In particular, The Bernoulli equation is an equality with 4 different terms. Changing the mass conservation does not only affect the global integral of dh/dt, but also that of ds/dt and dq/dt. The authors here assume -without proof- that the change in the enthalpy integral would be reflected solely in the work output.*

If the referee's assumption about absent sources and sinks in the analysis of Laliberté et al. [2015] were correct, we would agree with this statement. For example, if Laliberte et al. defined $h$ as enthalpy per unit dry air mass, then, as shown in our revised manuscript, the integral of $dh/dt$ over *total dry air mass* would be zero. The other terms in the first law of thermodynamics would look different, too, if taken per dry air mass.

However, Laliberté et al. [2015] defined $h$ as enthalpy per unit wet mass and, as is clear from their approach, integrated it over the entire mass of the atmosphere in the presence of mass sources and sinks. In this case the integral of $dh/dt$ is not zero and its omission is not justified.

The referee continues: *The broader issue here is that the discussion of section 3.1. and 3.2. is presented without context and incomplete. It could only be understood by very few potential readers. It makes the paper unnecessarily confusing and should be removed.*

The work of Laliberte et al. 2015 is published in a journal aimed at a broad readership. Their account is clear: the authors present the first law of thermodynamics and set out to integrate it over atmospheric mass. All the terms in the corresponding equation are explicitly defined. Then they state that the global integral of one of the terms is zero [p. 540, right column, 3rd line from top]. We evaluate this integral and show that it is not zero and that its omission significantly impacts the paper's quantitative conclusions.

If we submitted our present manuscript without discussing Laliberte et al., a referee would rightly advise us to acquaint ourselves with the current literature and address the discrepancy between our results and those of Laliberté et al. [2015] (who analyzed the same MERRA database). We thus believe that our analysis of Laliberte et al. 2015 is an essential part of our study and have striven to present it as clearly as possible in the revised manuscript.

**3.3 Comment 3**

*3. Overal structure:*

*The paper is poorly constructed. It is mainly three separate studies. Sections 2-4 attempt a theoretical discussion of the issues that mostly reprise previous work. It is unnecessarily confusing. Section 5 is the main 'new' result. The computation done are fairly routine, and the result in line with what we know. The inability of the authors to produce a consistent figure for Wp is distressing and should be better addressed in the revision. Section 6 is a lengthy disgression which is mostly a repeat of the authors previous work.*

Response [see also doi:10.5194/acp-2016-203-AC6]:

We believe that the revised paper is now much clearer and presents a coherent theme. We underline that the entire literature on this subject is somewhat confusing and it is the need to identify and examine the inconsistencies in other studies that leads to difficulties. Our revision is attentive to these difficulties (e.g. the different formulations of $W$).

We note that published approaches to $W_P$ suffer important inconsistencies. Pauluis et al. [2000] estimated, on theoretical grounds, that tropical $W_P$ (between 30N and 30S) should be between 2 and 4 W m$^{-2}$. Pauluis and Dias [2012] analyzed TRMM data to conclude that tropical $W_P$ is, rather, 1.8 W m$^{-2}$. Makarieva et al. [2013] likewise on theoretical grounds, suggested that Pauluis et al. [2000] overestimated tropical $W_P$ by around one-hundred per-cent. Their results led Pauluis and Dias to revise their calculations and publish a revised TRMM-based estimate of 1.5 W m$^{-2}$ for the tropics as a corrigendum to their 2012 work. On the other hand, Makarieva et al. [2013] suggested that global $W_P$ should be around 0.8 W m$^{-2}$; in our present work we show that the true value is around 1 W m$^{-2}$ and we address the associated uncertainties.

As we discussed in our reply to Comment 2 of Referee 1, we show in the revision that the inconsistency in the estimates of $W_P$ as well as of total power $W$ is an inherent property of the re-analyses. We disagree that this is already known, since we find no estimates of global $W_P$ from re-analyses or otherwise. This is indeed surprising given recent emphasis on the thermodynamic aspects of the water cycle [see, e.g., Pauluis, 2015]. We hope that our revised work brings greater clarity to this matter.

In particular, our results suggest that if Laliberté et al. [2015] used NCAR/NCEP rather than MERRA data for their analysis, they would have obtained a negative value for total atmospheric power (and, hence, $W_P$). Key to their result is the procedure of zeroing pressure

velocity at the surface between the modelling steps. If a different procedure were used, the results would be different as well.

**3.4    Recommendation**

*My recommendation here would be to simplify section 2 and 4, drop section 3 and expand on section 5. Section 6 could be clarified as well.*

Response:

We revised Sections 2 and 3 (former 4) to present a coherent overview of the available formulations for $W$, $W_K$ and $W_P$ and their physical meaning. We followed the recommendation of the referee to expand on section 5 having included more extensive analyses not only of MERRA but also of NCAR/NCEP for monthly, daily and 3-hourly resolution for 1979-2015 with four new figures. We would be willing to clarify Section 6 but since the referee provided no guidelines we just double-checked our messages for consistency.

Regarding our analysis of Laliberté et al. [2015] (former Section 3, now Section 4), we showed in the revised text that the omission of $I_h$ stems from the same reasoning that led to Comments 1 and 3 of Referees 2 and 4 concerning the definition of work. The reason is a misinterpretation of $dh/dt$ (or $d\alpha/dt$, where $\alpha$ is mass-specific volume) as the change per unit time of, respectively, enthalpy and volume per unit mass of a material element (air parcel). This is not correct in the presence of phase transitions, because the parcel's mass is not constant. The revised text clarifies this issue and should reduce future confusion.

**4    Referee 4 [doi:10.5194/acp-2016-203-RC4]**

**4.1    Comment 1**

*The main aim of this paper is to clarify the atmospheric power budget by seeking to exploit the divergent character of the gaseous mass flux in order to identify those terms in the power budget that can be related explicitly to the condensation/evaporation rates. The paper makes some valid point (Sections 2 and 3), such as pointing out that a term neglected in a recent study by Laliberte et al. (2015) is not only different from zero but too large to be really negligible, but the solution proposed does not seem valid. As to section 4, which claims to revisit the current understanding of the atmospheric power budget, it merely consists in some manipulation of the equations for a hydrostatic atmosphere that arguably sheds no light on the problem. The final section is too speculative. I don't think the paper makes a meaningful contribution to the understanding of atmospheric energetics, and I therefore cannot recommend publication.*

*Main comments 1. Abstract and elsewhere. I believe that the authors abuse the word power, which is used generically for all terms that enter the energy budget, such as in: Kinetic power associated with horizontal motion, the kinetic power associated with vertical motion, and the gravitational power of precipitation. In discussions of ocean and atmospheric energetics, it is more usual to restrict the term 'power' to the particular energy conversion responsible for supplying external energy to the system considered, and to be explicit as to what kind of energy conversions the other term represent. For instance, the term $u \cdot \nabla p$ is as far as a I can judge a conversion between available potential energy and kinetic energy, which is considerably more informative that 'kinetic power', and the authors should similarly clarify the physical meaning of the other terms.*

Response:

In the revised Section 1 we list many expressions used by various researchers to refer to global atmospheric power. Currently there is no consistency in terminology. This situation may reflect some confusion, which, as we show in our work, surrounds the definition and estimate of the power of atmospheric circulation in a moist atmosphere. In our work we

chose the word "power" because it is work per unit time, while our focus is on estimating the work output of the atmospheric circulation – in agreement with the thermodynamic definition of work. We discuss the physical meaning of all the terms of the atmospheric power budget in the revised Section 3 right below Eqs. (20)-(22) on p. 8.

**4.2 Comment 2**

it 2. The water cycle is generally regarded as making the atmospheric heat engine less efficient as the result of part of the solar forcing being expanded in lifting water vapour against the gravity field, part of which is then removed through precipitation, leaving only the residual to power the atmospheric circulation, an idea proposed by Pauluis and reprised in Laliberte et al. (2015). It seems that this should be discussed.

Response [see also doi:10.5194/acp-2016-203-AC6]:

The referee's account of the work of Laliberté et al. [2015] appears to be a misunderstanding. There are three relevant quantities: the power of a Carnot cycle $W_C$, the kinetic atmospheric power $W_K$ and the total atmospheric power $W$. The focus of Pauluis et al. [2000] was indeed to show that $W_K$ is lower than $W$ because, using the referee's words, solar power is "lifting water vapour against the gravity field, part of which is then removed through precipitation, leaving only the residual to power the atmospheric circulation". However, Pauluis [2011] advanced a different statement: that total power $W$ is lower than Carnot power $W_C$ because of the irreversible processes like water vapor diffusion. Laliberté et al. [2015] were likewise concerned about why $W$ is smaller than $W_C$ and did not assess the gravitational power of precipitation.

This misunderstanding might have stemmed from the comment of Pauluis [2015] on the work of Laliberté et al. [2015], where the two statements, $W_K < W$ and $W < W_C$, became mixed. To provide some context, an ideal atmospheric Carnot cycle consuming heat flux $F = 100$ W m$^{-2}$ at surface temperature $T_{in} = 300$ K and releasing heat at $T_{out} = T_{in} - \Delta T_C$ with $\Delta T_C = 30$ K, would generate kinetic energy at a rate of $W_C = F(\Delta T_C/T_{in}) = 10$ W m$^{-2}$. Laliberté *et al.* (2015) estimated total atmospheric power $W$ at around 4 W m$^{-2}$. Comparing their result with $W_C$, Pauluis [2015] noted that "estimates for the rate of kinetic energy production by atmospheric motions are about half this figure". Here confusion has apparently arisen between total atmospheric power $W$ and kinetic power $W_K$ (because Laliberté et al. [2015] assessed only $W$ but not $W_K$, the latter being about 2.5 W m$^{-2}$, i.e. a quarter rather than half of $W_C$). Indeed, Pauluis [2015] continued that "the difference is very likely due to Earth's hydrological cycle, which reduces the production of kinetic energy in two ways", one of which is the gravitational power of precipitation $W_P$ and the other is the irreversible diffusion processes. However, from our Eqs. (20)-(22), $W_P$ reduces $W_K$ compared to $W$ but it does not reduce $W$ compared to $W_C$, since $W_K + W_P = W < W_C$.

**4.3 Comment 3**

*3. Remarks on the methodology. Physically, the atmospheric energy budget is best understood by introducing some kind of available enthalpy ape $= h(\eta, q_t, p) - h_r(\eta, q_t)$, where $h$ is the moist specific enthalpy, $\eta$ is some suitable definition of moist specific entropy, and $q_t$ the total specific humidity, $p$ is pressure, where $h_r(\eta, q_t)$ representing the part of the total enthalpy that is not available for adiabatic conversions into kinetic energy, so that*

$$dh = (T - T_r)d\eta + (\mu - \mu_r)dq_t + \alpha dp$$

*As a result, it is possible to express the total power term as*

$$\int_V p\frac{D\alpha}{Dt}\rho dV = \underbrace{\int_V \frac{D(p\alpha)}{Dt}\rho dV}_{=0} - \int_V \alpha\frac{Dp}{Dt}\rho dV = \int_V \frac{T - T_r}{T}\dot{q}dm + \int_V (\mu - \mu_r)\frac{Dq_t}{Dt}dm$$

*where $\dot{q}$ represents diabatic heating terms by all manner of conduction of radiation. This neglects the integral of $dh/dt$, but this term could be retained if desired. The passage from the first term to the second term requires $\nabla(\rho v) = 0$, and $\rho v$ to the total mass flux, in order to be able to claim that the integral of $D(p\alpha)/Dt$ vanishes, so the authors should clarify this point, as well as boundary conditions assumed by the different velocities entering the definition of $v$. In any case, the above formalism is usually what constitutes the starting point for linking the atmospheric power budget to a Carnot-like theory and for constraining the atmospheric power budget to solar heating, sensible heat fluxes, and condensation/evaporation process. The approach proposed by the authors seem to be quite unrelated to this standard view.*

Response [see also doi:10.5194/acp-2016-203-AC3]:

The referee uses the same incorrect expression for work as Referee 2 in their Comment 1, with the same resulting discrepancies from our derivation. Total power is not equal to $W_{IV} \equiv (1/\mathcal{S}) \int_{\mathcal{V}} p(d\alpha/dt)\rho d\mathcal{V}$. This is clarified in the revised Section 2, see Eq. 11. Moreover, since $\nabla \cdot (\rho\mathbf{v}) = \dot{\rho} \neq 0$, the second equation of the referee contradicts the first one.

We note that our four referees appear to disagree on how the correct expression for atmospheric power $W$ should look like. Referee 1 (and implicitly Referee 3) agree with our Eq. (7), which shows that $W$ does not explicitly depend on the rate of phase transitions. Meanwhile, Referees 2 and 4 opine, respectively, that our results either *appear to be wrong* or are *unrelated to the standard view* suggesting two derivations of their own. However, as we have discussed, both derivations assume that work per unit mass is equal to $pd\alpha$, which is not a valid assumption in the presence of phase transitions. The resulting expressions contradict not only our Eq. (7) but also the identical Eq. (4) of Pauluis and Held (2002) endorsed by Referee 3. We hope that our revised text clarifies this topic.

**4.4    Comment 4**

*4. Sections 2 and 3. The whole point of the exercise of this exercise seems to establish that the term $\int_V dh/dt\rho dV$ assumed to be zero in Laliberte et al. is actually not zero, and that it is too large to be neglected. I agree with this statement, but the result obtained by the authors seems unphysical. The simplest way to show that the above term is not zero is through using using standard integration by parts*

$$\int_V \frac{dh}{dt}\rho dV = \int_V \nabla \cdot (\rho h v)dV - \int_V h\nabla \cdot (\rho v)dV = \int_{\partial V} \rho h v \cdot n dS - \int_V h\nabla \cdot (\rho v)dV$$

*How to estimate this term depends on how the velocity $v$, the density $\rho$ and enthalpy $h$ are defined. If $v$ is the fully barycentric velocity, and $\rho$ the full density, then mass conservation imposes $\nabla \cdot (\rho v) = 0$, and the term is controlled by boundary fluxes of enthalpy and is equal to the difference between the enthalpy evaporated minus the enthalpy precipitated. If $\rho v$ is the mass flux of the gaseous component of moist air, then how to estimate this term is more complicated, since $\nabla \cdot (\rho v) \neq 0$. Physically, the term $h\nabla(\rho v)$ is unphysical, since condensation or evaporation converts water vapour enthalpy $h_v$ into liquid water enthalpy $h_l$ and conversely, so should only involve the difference $h_v - h_l = L$, where $L$ is latent heat, it should not involve the dry air enthalpy; the formula $h\nabla(\rho v)$ involves the dry air enthalpy, however, which is part of the definition of $h$.*

*Physically, the result should not involve the dry air enthalpy, and should also be independent of the different constants entering the definition of the three forms of enthalpy, which the authors have not shown.*

Response [see also doi:10.5194/acp-2016-203-AC5]:

As was stated in our manuscript (see Eq. 5 on p. 3) and is perhaps better emphasized in our revision (first paragraph in Section 2, p. 3 and lines 16-18 on p. 4), velocity $v$ is the velocity of the gaseous component of moist air (i.e. of the substance that actually performs work). Enthalpy $h$ is defined per unit mass of wet air (i.e. dry air mass plus water vapor mass). There is thus nothing unphysical in the resulting expression for the integral of $dh/dt$

over total mass of dry air and water vapor depending on parameters of *both* dry air and water vapor.

In the revised Section 4 we explain the physical meaning of this result (see Eqs. 33 and 34). The integral of $dh/dt$ over mass is not zero simply because it does not represent changes of enthalpy per unit mass of a material element.

**4.5  Comment 5**

*5. Section 4. I don't really understand why this decomposition is useful. Indeed, a well known consequence of making the hydrostatic approximation is to filter out the contribution of the vertical velocity to the kinetic energy. As a result, the evolution equation for the kinetic energy becomes*

$$\rho \frac{D}{Dt} \frac{u^2}{2} + u \cdot \nabla p = \rho F \cdot u \tag{c19}$$

*so that in equilibrium*

$$\int_V u \cdot \nabla p \, dV = Friction, \tag{c20}$$

*which shows that only what the authors call the kinetic energy power (the conversion between kinetic energy and available potential energy) becomes relevant to understand how the atmospheric circulation is powered. As is also well known, even without the hydrostatic approximation, the budget of gravitational potential energy is zero*

$$\int_V \rho g w \, dV = 0 \tag{c21}$$

*where $\rho w$ is the total mass flux, and hence decoupled from the kinetic energy budget. One may if one so desires to separate the total mass flux into gaseous and liquid components, and restrict attention to the former, for which the GPE budget becomes*

$$\frac{d(GPE)}{dt}\Big|_{gas} = \underbrace{\int_V \rho g w \, dV}_{-SW_P} + GAS\ DESTRUCTION = 0, \tag{c22}$$

*where $\rho w$ is now the gaseous mass flux only, GAS DESTRUCTION means GPE sink due to destruction of water vapour mass by condensation, but that does not make it less decoupled from the horizontal kinetic energy budget, where the underlined term is what the authors call the* power of precipitation, *whatever that means. Physically, this term represents primarily a conversion with internal energy, and is not directly related to the kinetic energy of the system, making its usefulness for clarifying the atmospheric power budget dubious. Moreover, it is also well known that for a hydrostatic fluid, it is the total potential energy of the system (i.e., the enthalpy) that matters, given that large variations in gravitational potential energy are compensated by large variations in internal energy, with no impact on kinetic energy. The focus on gravitational potential energy, therefore, is at odds with the common wisdom that GPE is not useful to consider on its own. The claim that GPE variations are somehow connected with kinetic energy production is odd, given that the hydrostatic approximation is unconnected to the vertical velocity field.*

Response [see also doi:10.5194/acp-2016-203-AC6]:

The decomposition of total atmospheric power $W$ into the kinetic power of winds $W_K$ and the gravitational power of precipitation $W_P$ is useful in several ways. First, as we discuss below in response to Comment 6 of Referee 4 (see also revised Section 5.1, last paragraph on p. 16), $W_P$ and $W_K$ in re-analyses are characterized by substantially different uncertainties, so it is useful to keep a separate record for them. Second, $W_P$ can be estimated independently from wind velocities using observed precipitation; this information can be used to constrain vertical velocities. Third, since thermodynamics constrains total power $W$ and not kinetic

power $W_K$ or $W_P$ separately, it is necessary to clearly differentiate between $W$, $W_K$ and $W_P$ from a theoretical viewpoint. Distinguishing these components can help avoid confusions when comparing results from different studies (see also above our reply to Comment 2 of Referee 4). For example, given the modern concern about renewable energy resources it is necessary to understand that the so-called "wind power" [Marvel et al., 2013] as well as the river hydropower (which is part of $W_P$) are not the total power of the atmosphere.

We also note that in the presence of condensate the vertical distribution of gaseous air is not hydrostatic; the condensate loading term describes the generation of kinetic energy of the vertical air motions and is not zero. Furthermore, the integral of the left-hand part of the referee's equation (c19) is not zero in the presence of phase transitions, so Eq. (c20) does not hold. This is discussed in detail in the revised section 3, see p. 8 and Eq. 29 on p. 10.

**4.6   Comment 6**

*6. On a last note, I have a hard time accepting that the term $u \cdot \nabla p$ is something observable, given that the only way to estimate this term can only be done by means of a numerical model; likewise for the internal condensation/precipitation terms.*

Response [see also doi:10.5194/acp-2016-203-AC6]:

In the meteorological literature it is common to refer to the re-analyses data as to *observations* using which models outputs could be verified – see, for example, the study of Boer and Lambert [2008] devoted to the atmospheric energy cycle. This is because the re-analyses aim to systematize available observations of air pressure, velocity, temperature, humidity etc. in a coherent form. Air pressure and velocity are the basic observational parameters recorded. Likewise, precipitation is directly measured at the surface as well as assessed in the tropical atmosphere with use of satellites (the TRMM mission).

Since vertical velocities are small compared to horizontal velocities, they cannot be derived directly from observations. It is in this sense that the term $\mathbf{u} \cdot \nabla p$ is observable with a good accuracy, while the term $\rho w g$ responsible for the gravitational power of precipitation is not. This latter term can only be derived from observations using additional assumptions. Because of this difference, we estimate $W_K$ with less uncertainty than $W_P$. These uncertainties are estimated in the revised Section 5.1, see the last paragraph on p. 16.

**5   Summary of revisions**

Since the original manuscript has undergone a lot of changes, including some restructuring (Section 3 became Section 4 and vice versa), we do not provide a marked PDF with all the changes to avoid confusion. We list the changes made to the manuscript below.

1. Section 1 was revised to include an overview of various formulations of $W$ in the literature.

2. Section 2 was revised following the recommendation of Referee 1 to use an alternative derivation of $W$. Our original derivation became new Appendix A.

3. Section 3 (former Section 4) was revised by adding new subsections 3.1 (discussing the boundary condition for velocity) and 3.3 discussing previous work by Pauluis et al. The text of Section 3.2 remained relatively intact.

4. Section 4 (former Section 3) containing analysis of Laliberté et al. [2015] was shortened following the recommendation of Referee 2. Additionally, it was explained how the omission of the enthalpy integral is related to the incorrect definition $W_{IV}$ for $W$ in a moist atmosphere.

5. Section 5 was considerably revised following the recommendations of Referees 1, 2 and 3 to include new analyses. Three new figures were added.

6. Section 6 was slightly shortened and double-checked for clarity.

7. New Appendix A contains the derivation of $W$ for ideal gas.

8. Appendix B (former A) remains largely intact, but we added one paragraph on p. 27 to discuss the uncertainties of our precipitation-based $W_P$ estimate – as recommended by Referee 1.

9. Appendix C (former B) remains largely intact, but we added a new subsection C3 with a new figure explaining the impact of boundary values of $\omega$ and $\mathbf{u} \cdot \nabla p$ on the resulting estimates of $W$, $W_K$ and $W_P$.

10. Appendix D remained intact.

11. The abstract was modified to reflect the revisions made.

**References**

G. J. Boer and S. Lambert. The energy cycle in atmospheric models. *Climate Dynamics*, 30:371–390, 2008. doi: 10.1007/s00382-007-0303-4.

J. Huang and M. B. McElroy. A 32-year perspective on the origin of wind energy in a warming climate. *Renewable Energy*, 77:482–492, 2015. doi: 10.1016/j.renene.2014.12.045.

S. Kang and J. Ahn. Global energy and water balances in the latest reanalyses. *Asia-Pacific J. Atmos. Sci.*, 51:293–302, 2015. doi: 10.1007/s13143-015-0079-0.

F. Laliberté, J. Zika, L. Mudryk, P. J. Kushner, J. Kjellsson, and K. Döös. Constrained work output of the moist atmospheric heat engine in a warming climate. *Science*, 347:540–543, 2015. doi: 10.1126/science.1257103.

A. M. Makarieva, V. G. Gorshkov, A. V. Nefiodov, D. Sheil, A. D. Nobre, P. Bunyard, and B.-L. Li. The key physical parameters governing frictional dissipation in a precipitating atmosphere. *J. Atmos. Sci.*, 70:2916–2929, 2013. doi: 10.1175/JAS-D-12-0231.1.

K. Marvel, B. Kravitz, and K. Caldeira. Geophysical limits to global wind power. *Nature Climate Change*, 3:118–121, 2013. doi: 10.1038/nclimate1683.

O. Pauluis. Water vapor and mechanical work: A comparison of Carnot and steam cycles. *J. Atmos. Sci.*, 68:91–102, 2011. doi: 10.1175/2010JAS3530.1.

O. Pauluis and J. Dias. Satellite estimates of precipitation-induced dissipation in the atmosphere. *Science*, 335:953–956, 2012. doi: 10.1126/science.1215869.

O. Pauluis, V. Balaji, and I. M. Held. Frictional dissipation in a precipitating atmosphere. *J. Atmos. Sci.*, 57:989–994, 2000. doi: 10.1175/1520-0469(2000)057<0989:FDIAPA>2.0.CO;2.

Olivier M. Pauluis. The global engine that could. *Science*, 347:475–476, 2015. doi: 10.1126/science.aaa3681.

Kevin E. Trenberth. Climate diagnostics from global analyses: Conservation of mass in ecmwf analyses. *Journal of Climate*, 4(7):707–722, 1991. doi: 10.1175/1520-0442(1991)004<0707:CDFGAC>2.0.CO;2.